# Hybridization breaks species barriers in long-term coevolution of a cyanobacterial population

Gabriel Birzu[1,2]*, Harihara Subrahmaniam Muralidharan[3], Danielle Goudeau[4], Rex R Malmstrom[4], Daniel S Fisher[1]*†, Devaki Bhaya[5]*†

[1]Department of Applied Physics, Stanford University, Stanford, United States; [2]Department of Physics, University of Florida, Gainesville, United States; [3]Department of Computer Science, University of Maryland, College Park, United States; [4]DOE Joint Genome Institute, Lawrence Berkeley National Laboratory, Berkeley, United States; [5]Department of Plant Biology, Carnegie Institution for Science, Washington, DC, United States

*For correspondence:
gbirzu@ufl.edu (GB);
dsfisher@stanford.edu (DSF);
dbhaya@carnegiescience.edu (DB)

†These authors contributed equally to this work

Competing interest: The authors declare that no competing interests exist.

## eLife Assessment

This study provides **important** insights into bacterial genome evolution by analyzing single-cell genome sequences of cyanobacteria from Yellowstone hot springs. Using **compelling** evidence, the authors demonstrate that both homologous recombination within species and frequent hybridization across species are major drivers of genome diversification. Despite the challenges that are inherent to sparse and fragmented single-cell data, the analyses are thorough, carefully controlled, and supported by multiple complementary approaches, making the conclusions highly robust. This work represents a significant advance in our understanding of microbial evolution in natural environments.

**Abstract** Bacterial species often undergo rampant recombination yet maintain cohesive genomic identity. Ecological differences can generate recombination barriers between species and sustain genomic clusters in the short term. But can these forces prevent genomic mixing during long-term coevolution? Cyanobacteria in Yellowstone hot springs comprise several diverse species that have coevolved for hundreds of thousands of years, providing a rare natural experiment. By analyzing more than 300 single-cell genomes, we show that despite each species forming a distinct genomic cluster, much of the diversity within species is the result of hybridization driven by selection, which has mixed their ancestral genotypes. This widespread mixing is contrary to the prevailing view that ecological barriers can maintain cohesive bacterial species and highlights the importance of hybridization as a source of genomic diversity.

## Introduction

The origin, persistence, and extinction of species in obligate sexual populations is a fundamental problem in evolutionary theory (*Coyne and Orr, 2004*). But in bacteria, long thought to be primarily asexual, our understanding is still very limited (*Fraser et al., 2009*; *Rocha, 2018*). While it has long been known that bacteria can acquire non-essential genes from distantly related strains (*Rivera et al., 1998*), in recent years, studies across a wide range of bacteria have found that homologous recombination of core genes is also ubiquitous (*Didelot and Maiden, 2010*; *Yahara et al., 2016*; *Rosen et al., 2015*; *Garud et al., 2019*; *Sakoparnig et al., 2021*). How does the interplay between selection acting both asexually and on transferred elements, and barriers to recombination from geographical

**eLife digest** Bacteria are among the most successful organisms on Earth. These small, single-cell microbes inhabit nearly every environment on the planet and play essential roles in health, disease, ecology and industry. Their rapid cell division and ability to take up DNA from the environment allow them to adapt quickly to new challenges. After uptake, bacteria foreign DNA into genomes through a process known as recombination. This enables them to acquire beneficial genetic variants, such as antibiotic resistance genes, and spread them through the population.

Although the molecular mechanisms responsible for bacterial recombination have been extensively studied, its evolutionary consequences remain poorly understood. Unlike sexually reproducing organisms, in which the entire genome is recombined every generation, bacteria exchange only a small fraction – typically less than 1% of their genome. In theory, recombination could reduce genetic differences within a population, making them more similar. Alternatively, if genetic divergence between bacterial strains continued to accumulate despite recombination, those strains could eventually evolve into distinct species.

Previous research on thermophilic cyanobacteria from Yellowstone National Park revealed evidence of frequent DNA exchange between these microbes. This recombination produced genomes essentially containing – even in their conserved core – a random assortment of gene variants from across the population. Building on this work, Birzu et al. investigated whether such recombination affects the cohesiveness of different species or whether DNA exchange occurs primarily within a species. Answering this question is an important step toward developing quantitative models of evolution in natural microbial populations.

Birzu et al. quantified the impact of recombination between species – referred to as hybridization – on genetic diversity within species. To do so, they developed a method to identify hybrid DNA segments transferred from other species. When the donor species is known, hybrid segments can be identified simply by comparing genomes, a method typically used to identify Neanderthal DNA in human genomes. However, this approach may fail when donor genomes are unavailable or extinct.

To overcome this limitation, Birzu et al. exploited the fact that genetic distances between species are typically much larger than those *within* species. Hybridization would thus generate long sequences of highly correlated mutations within the recipient population. Using this idea, Birzu et al. identified hybrid segments and showed that hybridization accounted for up to 95% of the genetic diversity within one of the species.

These results suggest that – rather than diverging – the Yellowstone cyanobacterial population is being homogenized by recombination, leading to a gradual blending between different species. These results have broader implications for understanding microbial evolution in contexts such as the emergence of new pathogens or the adaptation of marine microbes to climate change. They suggest that microbial species may function less as a small number of cohesive strains, and more as quasi-sexual populations with continual DNA exchange within and between species. Extending the approach from this paper to other microbial populations and developing new methods to quantify the impact of hybridization will help clarify which of these evolutionary scenarios is most common in nature.

separation, mechanistic effects, and epistatic interactions lead to formation and maintenance of cohesive species?

One explanation is that ecological specialization provides barriers to recombination (*Kashtan et al., 2014*; *Arevalo et al., 2019*). This view is supported by evidence that selective sweeps of individual genes through parts of a population can lead to distinct ecological subtypes, or ecotypes, even in highly recombining bacterial populations (*Croucher et al., 2011*; *Shapiro et al., 2012*; *Bao et al., 2016*). But it also raises an important open question: if species coexist and recombine with each other, do genetic differences continue to increase over evolutionary time, or does recombination gradually break down barriers between species?

Answering this question is challenging because data on the underlying evolutionary dynamics is lacking. The time scales of direct observations and laboratory experiments are too short, while those corresponding to phylogenetic relationships between extant species are too long. And since spatial

dynamics over these intermediate time scales are unknown in most bacteria (*Hanson et al., 2012*; *Garud et al., 2019*), it is difficult to control for changes in local community structure during evolution. Therefore, microbial communities that coevolved for long time periods can act as natural experiments for studying bacterial speciation. The cyanobacterial populations found in hot springs from Yellowstone National Park form a natural incubator in which one such natural experiment is ongoing. These communities have two key features that allow us to identify and quantify the effects of coevolution on intermediate time scales.

First, like other thermophilic communities, these populations are geographically isolated (*Papke et al., 2003*; *Whitaker et al., 2003*). To date, the two most abundant species, *Synechococcus* A and B, have not been found outside of North America, suggesting that their global mixing time is much longer than for most other bacteria (*Papke et al., 2003*). Comparison of two isolate genomes from these species, OS-A and OS-B', suggest a divergence time considerably longer than the formation of the Yellowstone caldera around 640,000 years ago (*Rosen et al., 2018*). This would imply their ancestors colonized the caldera independently.

Second, these populations have considerable sub-species diversity, suggesting ongoing evolutionary processes generate and maintain it. Analysis of variants of 16S rRNA and marker genes was initially interpreted as evidence of asexual ecotypes occupying distinct niches (*Ward et al., 2006*; *Ward et al., 1998*). However, deep amplicon data from a diverse, but largely OS-B'-like sample revealed extensive recombination (*Rosen et al., 2015*). Genetic exchange with other genomic clusters—which we refer to as hybridization—was also evident, particularly with relatives of OS-A. To date, evidence of hybridization in bacteria has mainly been from comparisons between reference genomes from large genomic databases (*Smillie et al., 2011*; *Diop et al., 2022*), with only a handful of studies investigating hybridization in natural environments (*Tyson et al., 2004*; *Lo et al., 2007*; *Sheppard et al., 2008*). The primary focus of the current paper is to determine the impact of hybridization on the long-term coevolution of a natural microbial community.

Characterizing hybridization requires long-distance linkage between variants that cannot be obtained from traditional metagenomics. Strain isolates are often not representative of the population diversity (*Browne et al., 2016*; *Zheng et al., 2022*), while metagenome assembly destroys linkage. Recruiting metagenomic reads to reference genomes can, in some cases, be used to infer linkage

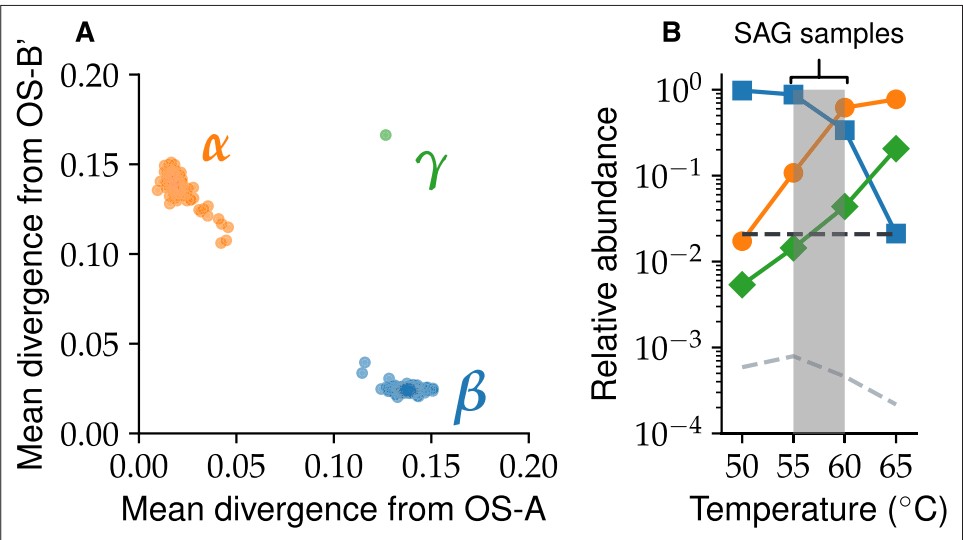

**Figure 1.** *Synechococcus* population comprises three genomic clusters. (**A**) The mean divergence across homologous genes from the two reference genomes is shown for every genome in the data. Each dot represents a single genome and is colored according to its assigned species. (**B**) The relative abundances of all three species across samples from different temperatures within Mushroom Spring. Abundances were estimated based on read recruitment from the metagenome to core genes from the γ genome, as well as representative genomes from the α and β populations. Dashed lines show the minimum detection frequencies from the single-cell (black) and metagenome (gray) data, respectively (see Appendix 3). Shaded gray region shows temperature range of the eight single-cell samples.

(*Garud et al., 2019*), but the high diversity of *Synechococcus* prevents us from using such approaches. To circumvent these problems, we used single-cell genomics (*Rinke et al., 2013*; *Zheng et al., 2022*) to obtain over 300 genomes from the Mushroom and Octopus Springs, which contain two of the most well-studied communities from the Yellowstone caldera (*Ward et al., 2012*). This is to our knowledge the first large sample of genomes from hot spring cyanobacterial populations *without* the compositional bias due to strain isolation.

## Results

### Population comprises three distinct highly recombining genomic clusters

Previous analysis of 16S amplicons identified two main clusters similar to OS-A and OS-B', but also several others at frequencies below 1% (*Rosen et al., 2018*). But whether clusters within 16S amplicons correspond to clusters at the whole-genome level in such a rapidly recombining population is not clear (*Rosen et al., 2015*; *Miller and Carvey, 2019*). Therefore, we first compared each single-amplified genome (SAG) to *Synechococcus* OS-A and OS-B'. Almost all SAGs (330 out of 331) comprised two clusters with an average nucleotide divergence of ~2% from one of the reference genomes and ~15% from the other. Note that the divergence between clusters is well above the 5% threshold that is commonly used to distinguish different species (*Nayfach et al., 2016*; *Jain et al., 2018*; *Garud et al., 2019*). We labeled the cluster closer to OS-A by $\alpha$ (128 SAGs) and the other by $\beta$ (202 SAGs), to distinguish them from the reference genomes. Surprisingly, one SAG (approximately 70% complete) was ~15% diverged from both OS-A and OS-B' and formed a separate cluster which we labeled $\gamma$ (*Figure 1A*). Clustering methods based on different metrics gave identical results (*Appendix 2—figures 2 and 3*).

To investigate the extent that these clusters characterize the distribution of genomes throughout the two springs, we used single-cell genomes as references and recruited reads from a large collection of metagenome samples to determine the abundance of $\alpha$, $\beta$, and, particularly, $\gamma$ across different environments (see Appendix 3). We found that 95% of reads were part of a main cloud of sequences with ≤5% nucleotide divergence from one of the three main clusters. We used the fraction of reads within each main cloud to infer the abundances of each cluster in the metagenomic samples. The abundance of $\gamma$ was highly correlated with $\alpha$ (*Appendix 3—figures 3 and 4*) and increased rapidly with temperature in the Mushroom Spring samples (*Figure 1B*). One interpretation of this pattern is that $\alpha$, $\beta$, and $\gamma$ represent distinct species adapted to different temperatures. Alternatively, genotypes within

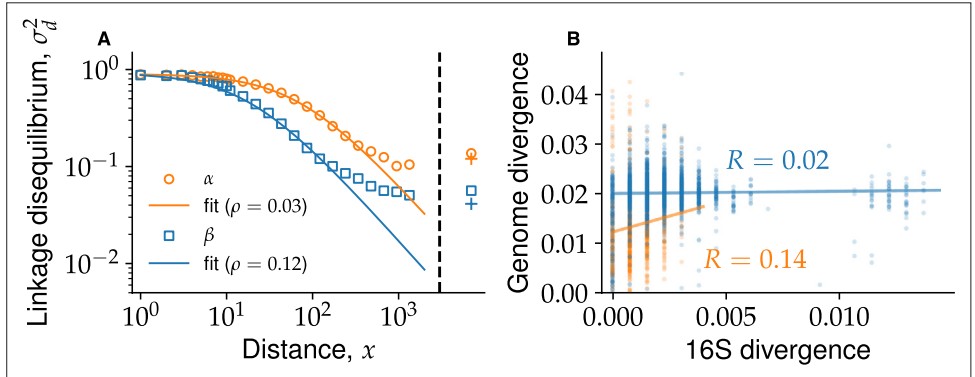

**Figure 2.** Genetic linkage analysis shows extensive recombination within $\alpha$ and $\beta$. (**A**) Linkage disequilibrium $\sigma_d^2$ as a function of separation between SNPs $x$ averaged across all core genes from $\alpha$ (orange circles) and $\beta$ (blue squares). Genome-wide estimates from SNPs in different genes are shown to the right of the black dashed line using the same symbols. Fully unlinked controls for the genome-wide linkage are shown by crosses of corresponding colors. Dashed lines show fit to linkage decay curves in an infinite sample using *Equation 6* for $\alpha$ (orange) and $\beta$ (blue) separately. (**B**) Scatter plot of 16S and genome-wide divergences between all pairs of genomes within the same species. Solid lines show linear regressions indicating very weak Pearson correlation coefficient, *R*, within each species. This demonstrates that 16S divergence is a poor proxy of whole-genome divergence.

the same cluster could be adapted to different temperatures or other environmental conditions. The relative abundances of the clusters would then reflect the complex eco-evolutionary processes that maintain the diversity in the population.

Previous analysis using a much more limited dataset proposed that $\beta$ formed a quasisexual population, in which rapid recombination led to random assortment of alleles within the cluster (*Rosen et al., 2015*). Motivated by these results, we investigated the impact of recombination on the diversity within the two main clusters.

We quantified recombination rates in $\alpha$ and $\beta$ using a standard measure of linkage disequilibrium, $\sigma_d^2$. Given a pair of sites with alleles a/A and b/B, $\sigma_d^2$ is defined as

$$\sigma_d^2 = \frac{\langle (f_{ab} - f_a f_b) \rangle}{\langle f_a(1 - f_a)f_b(1 - f_b) \rangle}, \tag{1}$$

where $\langle \cdot \rangle$ is the average across pairs of sites. In asexual populations with conventional demographic drift, $\sigma_d^2 = 5/11$ independently of the location of the sites. Recombination events affecting one but not both sites can unlink mutations, leading to a decrease in $\sigma_d^2$. Because sites further apart have more opportunities to become unlinked, $\sigma_d^2$ decreases with the distance between sites.

In $\beta$, the linkage disequilibrium $\sigma_d^2$ decreased by around a factor of 10 over distances of ~1 kbp (*Figure 2A*). This agreed quantitatively with earlier results based on deep amplicon sequencing from *Rosen et al., 2015*, which showed a decrease by a factor of ~1 over distances of up to 300 bp, using a different but related measure of linkage, conventionally denoted as $\langle r^2 \rangle$. Over genome-wide length scales, linkage was small ($\sigma_d^2 = 0.06$) and close to a fully unlinked control ($\sigma_d^2 = 0.04$). We found similar results for $\alpha$, but with a quantitatively slower decrease in linkage by a factor of ~8 over ~1 kbp. Nevertheless, genome-wide linkage values were still close to the fully unlinked control ($\sigma_d^2 = 0.14$ for

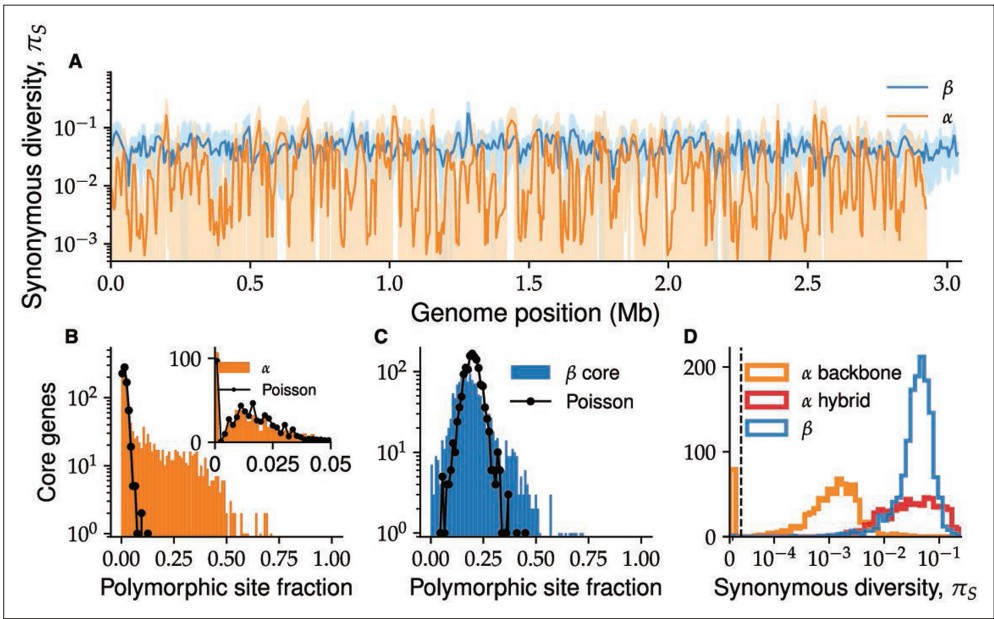

**Figure 3.** $\alpha$ and $\beta$ have distinct patterns of within-species diversity. (**A**) The synonymous diversity $\pi_S$ within $\alpha$ (orange) and $\beta$ (blue) at all core genes ordered according to their respective reference genomes. The solid line shows the value of $\pi_S$, with lighter colors showing one standard deviation across different pairs around the mean. Both lines were smoothed by averaging over a sliding window of five genes with a step size of two genes. (**B**) Distribution of the fraction of polymorphic fourfold degenerate (4D) sites for each $\alpha$ core gene (blue histogram) is shown together with Poisson fit based on the number of genes without any polymorphic sites (black line and dots). The inset shows zoomed-in region of low-diversity genes (<5% polymorphic 4D sites) together with the same fit. (**C**) The same as (**B**) using data from $\beta$. The Poisson distribution is fitted using the mean of the distribution, which provides a better fit than the number of genes without polymorphic sites (data not shown). (**D**) Histogram of synonymous diversities $\pi_S$ within genes for $\beta$ (blue), $\alpha$ low-diversity backbone (orange), and $\alpha$ high-diversity (hybrid) genes (red). Note the overlap between the diversity in the $\beta$ core genome and the hybrid $\alpha$ genes.

the data vs $\sigma_d^2 = 0.12$ for the control). As a consequence of rapid recombination, we found that diversity in marker genes, such as 16S rRNA, was poor predictors of whole-genome diversity (*Figure 2B*). Further details and analyses can be found in Appendices 4 and 8. Appendix 8 also provides estimates of the recombination parameters for each species and compares these to estimates obtained from previously proposed methods (*Didelot and Wilson, 2015*; *Lin and Kussell, 2019*). Altogether, these results are consistent with each cluster representing a quasisexual population and imply that phylogenies inferred from whole genomes do not reflect the evolutionary history of the population.

While linkage decay in $\alpha$ and $\beta$ was similar, we found diversity patterns at the gene level were qualitatively different (*Figure 3A*). The typical diversity at synonymous sites in $\beta$ was $\pi_S \approx 0.04$, with ~2× variation across genes (*Figure 3A*), consistent with previous results (*Rosen et al., 2018*). In contrast, $\pi_S$ in $\alpha$ varied wildly by over two orders of magnitude (*Figure 3A*), ranging from $\pi_S \approx 0.2$ in some genes to others that did not have a single mutation. Note that because here we only used synonymous sites, these variations cannot be explained by differences in the strength of purifying selection or the accumulation of non-synonymous mutations during rapid adaptation (*Neher, 2022*). Importantly, the low-diversity segments of the $\alpha$ genome spanned multiple samples and springs and are thus unlikely to be the result of a very recent clonal expansion as is often seen in human commensal and pathogenic bacteria (*Didelot and Wilson, 2015*; *Garud et al., 2019*; *Sakoparnig et al., 2021*; *Weimann et al., 2024*).

Beyond the low-diversity genes in $\alpha$, we found genomes in both species containing segments very similar or identical to homologous regions from the other species. We performed extensive checks to verify that these segments were not assembly artifacts or cross-contamination between reaction wells during whole-genome amplification (Appendix 10). Such segments could represent accessory genes transferred between species, as has been widely reported in both *Synechococcus* and other species (*Bhaya et al., 2007*; *Smillie et al., 2011*). Alternatively, they could be the result of homologous recombination with other species within the core genome. We refer to the later process as *hybridization* throughout. The rest of our analysis aims to quantify the extent of hybridization and its impact on the evolution of the population. We began by identifying the core genes shared by $\alpha$ and $\beta$.

## Gene-level analysis reveals distinct patterns of hybridization between species

We defined core genes by constructing groups of orthologous genes (orthogroups) and identifying those that were likely present in all genomes. Specifically, we clustered gene sequences in several steps using standard methods based on similarity (Appendix 1). We used phylogenetic distances to identify and separate distant paralogs within the same orthogroup. To assign clusters corresponding to each species, we analyzed the abundance distributions of orthogroups in $\alpha$ and $\beta$ separately. Because of the incomplete nature of our genomes (average coverage ≈0.3, with a range of ≈0.05–0.95; see Appendix 1), the assignment has to be done statistically. Distributions for both species were bimodal, with a peak near zero corresponding to flexible genes and another at high abundances corresponding to core genes. We used a binomial fit to determine single-copy genes that were likely present in all genomes and found 1825 core $\alpha$ and 1737 core $\beta$ orthogroups. Of these, 1448 orthogroups were shared by both $\alpha$ and $\beta$ and make up the *Synechococcus core genes*. We focus the rest of our analysis on these core genes.

We first performed a quantitative comparison of the synonymous diversity between $\alpha$ and $\beta$ across the core genes. We hypothesized that the low-diversity segments of the $\alpha$ genome were the result of mutation accumulation since their common ancestor. If mutation rates across genes are similar, we expect the fraction of polymorphic sites $f_p$ for each gene to follow a Poisson distribution. Empirically, we found that for $f_p < 0.05$, the data closely matched the Poisson expectation with a single fit parameter (*Figure 3B*). This group, which we call the *ancestral backbone of* $\alpha$, contained around half of the core genes (675 out of 1448). Note that this result is distinct from the distribution of divergences *between* species, which was shown previously to also follow a Poisson distribution (*Rosen et al., 2018*). The remaining half (773 out of 1448) had up to ten times higher fraction of segregating sites and were inconsistent with the backbone polymorphism rate. As our subsequent analysis shows, the high diversity in these genes is the result of hybridization (see *Figures 4–7* and subsequent discussion). We therefore refer to this group as the *hybrid* $\alpha$ genes. In $\beta$, the distribution of $f_p$ had a single peak, but a simple Poisson process was a poor fit to the data (*Figure 3C*). This suggests a substantial

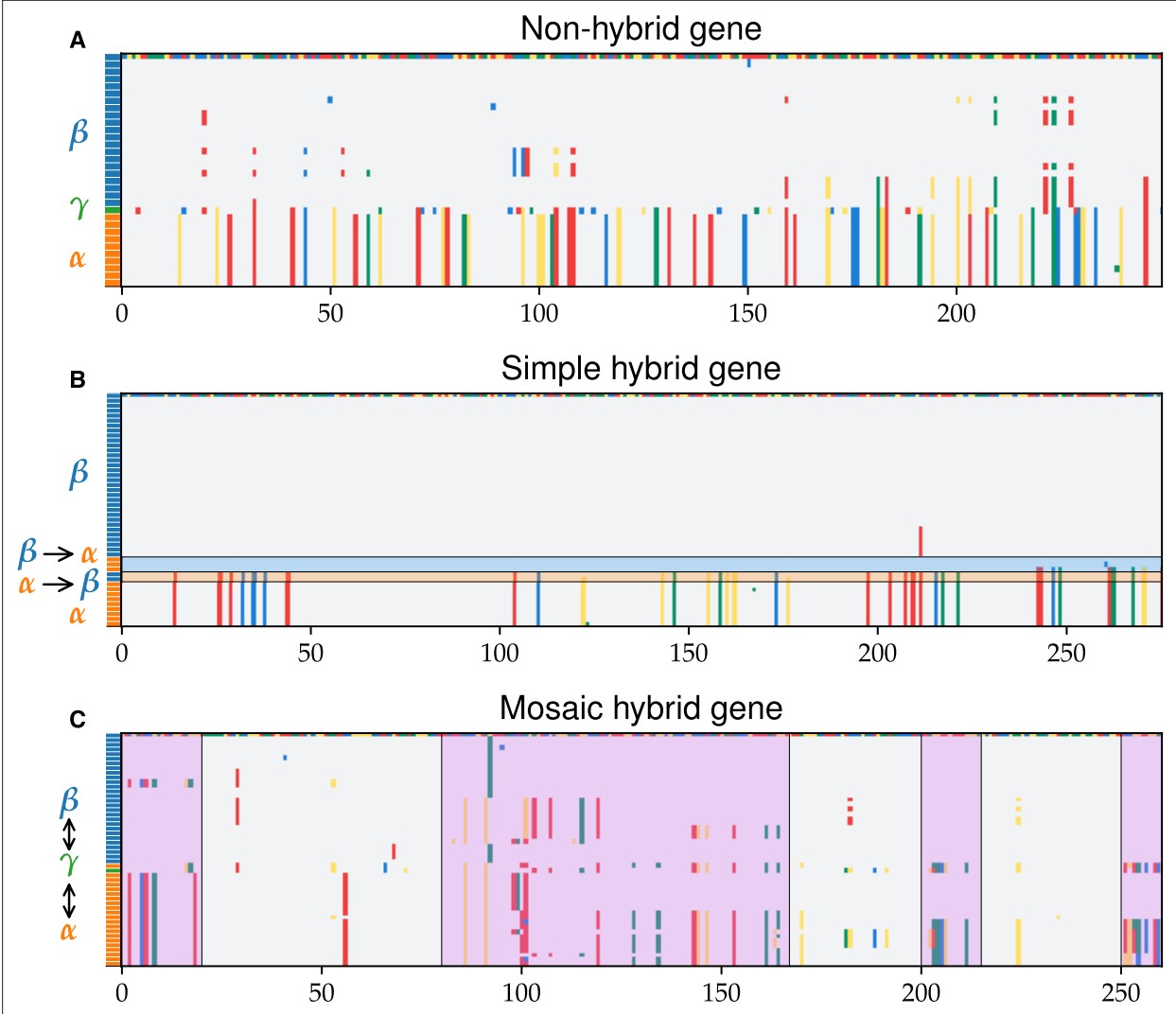

**Figure 4.** Alignments reveal distinct patterns of hybridization across different genes. Three core gene alignments, each chosen to illustrate a distinct pattern of hybridization, are shown. Rows represent different cells and columns different positions along the gene. One sequence was chosen randomly as a reference (top row), with nucleotides shown in different colors. The species assigned to each cell are indicated by the colored rectangles on the left, using the same color scheme as the species clusters ($\alpha$ shown in orange, $\beta$ in blue, and $\gamma$ in green). Rows were sorted hierarchically based on pairwise distances. Nucleotides are shown in gray if they are identical to the reference genome and in a different color if different. Species clusters were assigned as described in the main text and Appendix 1. (**A**) Segment of an alignment containing three distinct species clusters of a non-hybrid gene. (**B**) Segment of an alignment containing two distinct clusters ($\alpha$ and $\beta$) and six different simple hybrids. The direction of the transfer was inferred as described in the main text and is indicated on the left. Transfers from $\alpha$ into $\beta$ and $\beta$ into $\alpha$ are highlighted in orange and blue, respectively. Note that the presence of more than one type of transfer is rare and was chosen here for illustration. (**C**) Segment of an alignment without distinct species clusters chosen to illustrate mosaic hybrid genes. Regions with clear hybrid blocks are highlighted in purple. Note the extensive hybridization on short length scales, despite the fact that sequences from the same species cluster with each other based on pairwise distances.

fraction of the diversity is not simply due to mutations but is introduced through recombination with other strains or species. Consistent with this interpretation, we found the distribution of synonymous diversities in the hybrid $\alpha$ overlapped that of $\beta$, and both were two orders of magnitude higher than in the $\alpha$ backbone (*Figure 3D*). These results suggest that hybridization had a large impact on the genetic diversity of both $\alpha$ and $\beta$. We therefore developed a method to identify hybrid segments directly and test this hypothesis.

We used hierarchical clustering to identify and characterize hybridization patterns between species. If hybridization is common, as in our case, inferences based on phylogenetic trees would not provide reliable estimates of pairwise divergences. We therefore used pairwise nucleotide distances directly. The distribution of pairwise distances was bimodal, with a clear gap around $d = 0.075$, consistent with

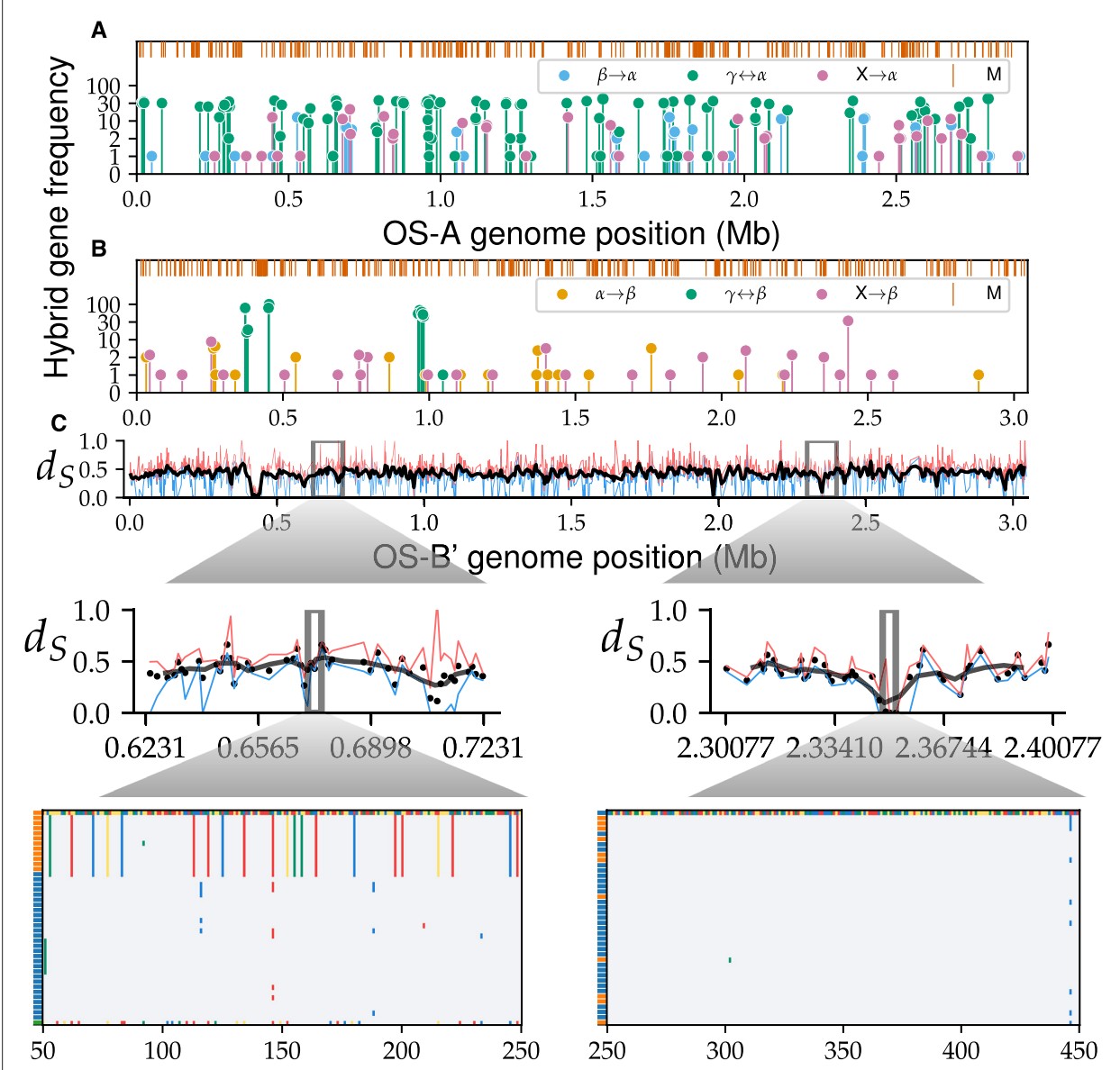

**Figure 5.** Transfers of whole-gene alleles between species are common. The frequency of hybrid gene alleles along the α (**A**) and β (**B**) genomes. Different colors show the donor species, with unknown species labeled $X$ and shown in magenta. The orange lines on top show mosaic loci without distinct gene clusters. Note that the x-axis in (**B**) is shared with (**C**). (**C**) The mean synonymous divergence between α and β at homologous loci, ordered according to OS-B'. Smoothed line over a sliding window of five genes with a step size of two genes is shown in black. Maximum and minimum divergences at each locus are shown in red and blue, respectively. (Insets, middle) A zoomed-in region from a typical genomic region (left) and a divergence trough (right), with dots representing individual genes. (Insets, bottom) Alignment segments of individual genes, subsampled to 40 sequences for clarity. The first sequence is arbitrarily chosen as the reference with each nucleotide represented by a different color. Other sites are colored gray if same as reference or using nucleotide colors if mutated. Left-colored rectangles show the species of each genome.

our comparisons to the reference genomes (**Figure 1**). Based on this observation, we defined species clusters for each gene using a simple cutoff at $d_c = 0.075$ (**Appendix 1—figure 2**) together with additional constraints to ensure that clusters were well separated and consistent with asexual divergence from a common ancestor (Appendix 1).

Detailed analysis of species clusters revealed evidence of hybridization in roughly half of the genome. We found two distinct patterns of hybridization. Around 75% of core genes (1078 out of 1448) had two to three clusters of sizes roughly proportional to abundances of α, β, and γ in the samples (**Appendix 5—figure 1**). The majority of these genes (844 out of 1078) had clusters, each

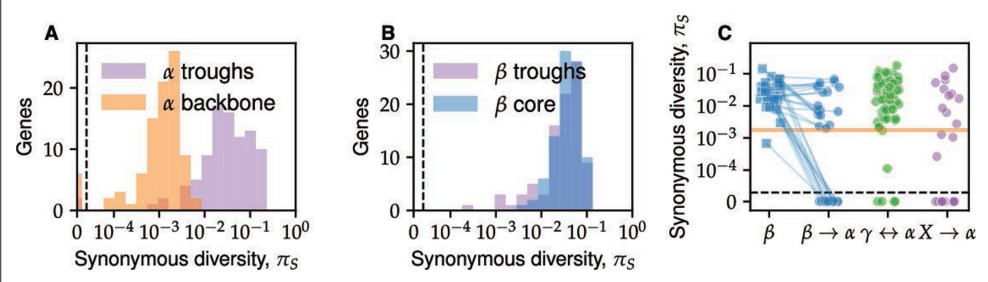

**Figure 6.** Diversity within hybrid genes and divergence troughs reveals mixture of soft and hard sweeps. (**A**) The distribution of synonymous diversities in α genomes in the divergence troughs (purple histogram) is compared to a random sample of genes from the α backbone of the same size (orange histogram). The orange line shows the bootstrapped distribution of the α backbone. (**B**) Same as (**A**), but comparing divergence troughs in β genomes with the β core genome. (**C**) Each circle shows the synonymous diversity of a non-singleton simple hybrid locus (see *Figure 5AB*) in α. The donor species are shown in different colors as labeled on the x-axis. For transfers between α and β, the diversity within β is shown as squares of the same color as a control, with sequences from the same locus connected by a line. The orange horizontal lines show the synonymous diversity in the low-diversity α backbone. Transfers above this line likely contain multiple independent events.

of which contained sequences from a single species (*Figure 4A*), as expected if they were asexually diverged. We call these *non-hybrid loci*. A significant minority (234 out of 1078) contained a mixture of sequences from different species (*Figure 4B*), representing recent transfers of whole genes between species. We call these *simple hybrid loci*. The remaining 25% of core genes (370 out of 1448) formed either a single cluster or had one very large cluster containing most sequences (*Figure 4C*). As we show below, these large clusters result from complex patterns of hybridization *within single genes*. We thus call these *mosaic hybrid loci*. In around a quarter of these loci, we found evidence of complete or near-complete gene sweeps, but in most, the divergence between α and was typical. Almost half (297 out of 773) of high-diversity α genes were mosaic hybrids. To better understand the dynamics of hybridization in the population, we analyzed both hybridization patterns in detail, starting with the simple hybrid loci.

## Simple hybrid genes are common and reveal distinct patterns of hybridization across species

We used a simple set of heuristic criteria to assign the transfer direction of simple hybrids. If species evolved by asexual divergence from a common ancestor, we expect each orthogroup to contain two to three clusters of sizes roughly proportional to the species abundances. For example, based on the average coverage and the proportion of β in our samples (*Figure 1A*), we expect the largest cluster to contain ~80 sequences from β genomes. In some cases (45 out of 1063), we also find a minority of sequences from α genomes in this β cluster, which we infer to be the result of a hybridization event from β into α (*Figure 4B*). An alternative explanation would be a transfer from a third species that underwent a full selective sweep through β and only a partial sweep through α. This scenario would imply that hybridization is even more common than we infer, so our method is conservative. We infer transfers from α into β in a similar way (*Figure 4B*). When the third cluster is present and contains γ, we assign all other sequences in the cluster to hybridization with γ. Because we only have a single γ sequence, these clusters are difficult to classify. We cannot distinguish between transfer *from* and *to* γ, and it is possible that some simple hybrid loci involving γ may actually be mosaic hybrids. When the third cluster is present but does not contain γ or in cases where we find a fourth cluster, we assign those sequences to hybridization with an unknown species X. Using this procedure, we systematically determine the frequencies of different hybridization events in α and β.

We found simple hybrids were present in 5–10% of core genes in both species (*Figure 5A, B*). Specifically, we found 185 simple hybrid loci in α and 68 in β. Note that we conservatively excluded contigs that were fully hybrid from this analysis, so our numbers likely underestimate hybridization rates in the population (Appendix 5).

In both species, simple hybrids were scattered along the genome and contained alleles from different donor species (shown in different colors in *Figure 5A, B*), implying multiple hybridization

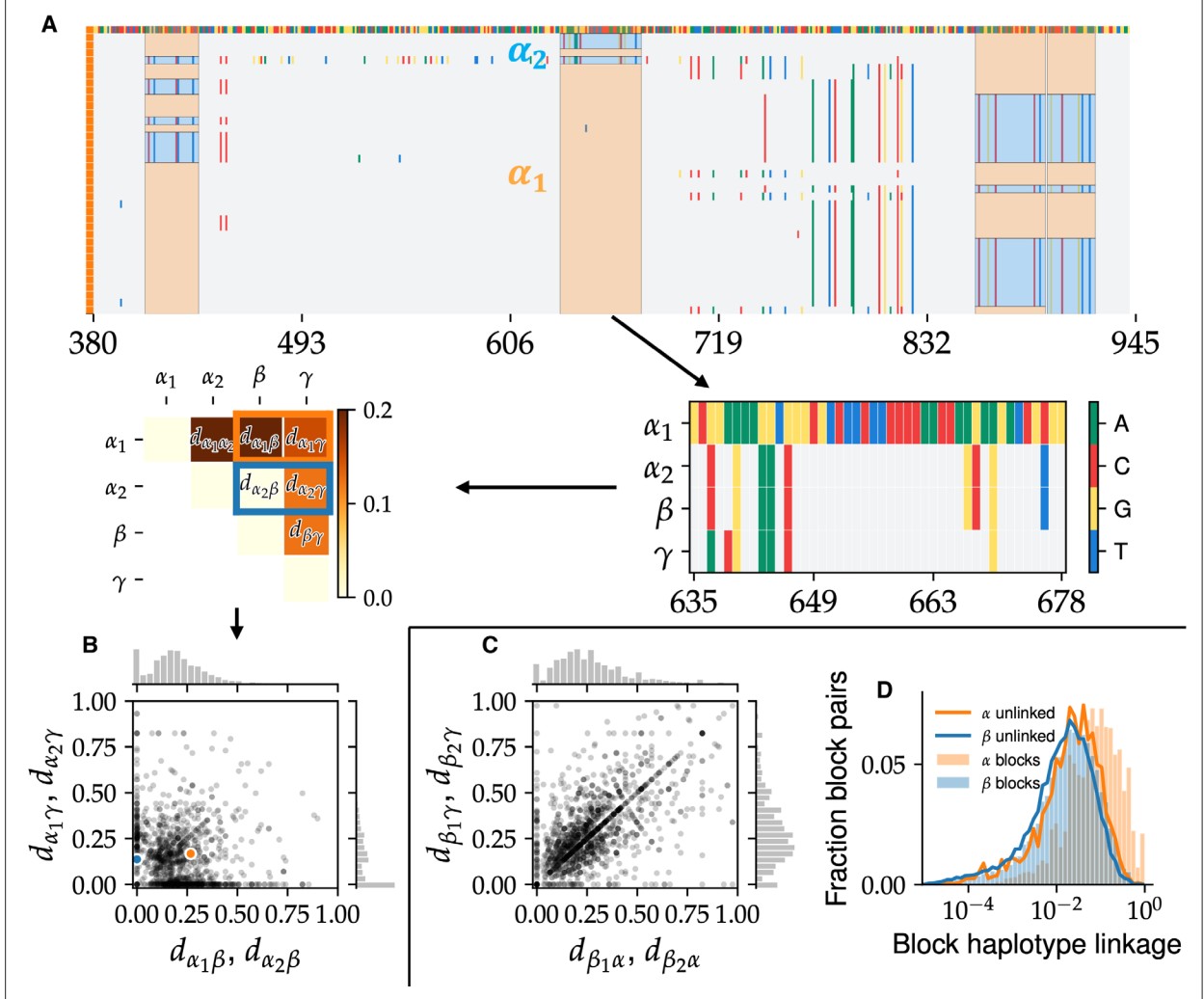

**Figure 7.** Hybridization between species leads to mixing of short DNA segments within single genes. (**A**, top) Alignment segment of $\alpha$ genomes from a representative locus with four SNP blocks. The inferred ancestral $\alpha$ haplotype (labeled $\alpha_1$) is highlighted in orange and the hybrid haplotype (all $\beta$ here; labeled $\alpha_2$) is shown in blue. Note the SNPs in different blocks in different sets of cells. (Middle) Illustration of SNP block analysis pipeline. (Right) The consensus alignment of the two $\alpha$ haplotypes, $\beta$, and $\gamma$ is constructed. (Left) Pairwise divergences between consensus sequences are calculated. Note that $\alpha_2$ is identical to $\beta$ here. (**B**) The joint distribution of divergences of $\alpha$ SNP block haplotypes from the other species for all SNP blocks. Each point represents a haplotype with coordinates showing the divergence from $\beta$ (x-axis) and $\gamma$ (y-axis). The orange and blue dots show the two haplotypes from the example in (**A**). Marginal distributions are shown on the top and right. Note the high density of haplotypes near the axes, indicating transfer from one of the other known species. (**C**) Same as panel (**B**) for $\beta$ block haplotypes. Note the much larger number of haplotypes far from the axes compared to (**B**), indicating transfers of unknown origin. (**D**) Histogram of linkage coefficients between pairs of block haplotypes at different loci for $\alpha$ (orange histogram) and $\beta$ (blue histogram) compared to the expectation under a fully unlinked model (solid orange and blue lines, respectively).

events occurred independently, as expected when recombined segments are much shorter than the genome. Around 80% of $\alpha$ hybrid alleles were present in multiple cells, while in $\beta$ the fraction was lower, at around 50%. The frequencies of simple hybrids were highly variable, ranging from singletons to frequencies close to 100%, consistent with a continual hybridization scenario. Surprisingly, around half of hybrids in $\beta$ and a fifth of those in $\alpha$ were from unknown donors. However, a systematic search over the 34 metagenomic samples did not find evidence of other *Synechococcus* species in either spring (*Appendix 3—figures 5–7*). These sequences could be the result of hybridization with species that were formerly present in the caldera and are now either extinct or at very low abundances. Alternatively, the hybrid sequences could have resulted from hybridization with, as yet, undiscovered species from other springs. But regardless of the exact scenario, overall, these results suggest the current population is descended from multiple cyanobacterial species that hybridized with each other.

## Soft and hard selective gene sweeps drive hybridization of whole genes

Is the genomic mixing between species the result of neutral processes or does positive selection for beneficial hybrids play an important role? Distinguishing between these alternatives is difficult in general, so we instead focused on cases where beneficial hybrid genes swept through the entire host species, that is, full gene sweeps. Full gene sweeps would greatly reduce the synonymous divergence between $\alpha$ and $\beta$, making them much easier to detect. Visually, we observed the synonymous divergence between $\alpha$ and $\beta$ contained several *divergence troughs*, in which the average diversity was well below typical values. We identified the genes contained in these troughs by imposing a simple cutoff $d_{trough} = 0.2$ on the synonymous divergence between $\alpha$ and $\beta$, which we chose close to the inflection point of the synonymous divergence distribution across genes (data not shown).

A systematic search revealed 91 very low divergence loci, spread across more than half a dozen divergence troughs (*Figure 5C*). Comparing alignments of typical genes and divergence through genes revealed a dramatic difference in fixed substitutions between species, which can only be explained by a recent selective sweep. The largest divergence trough contained 27 genes, representing around 25% of divergence trough genes, most of which were part of a nitrogen-fixation pathway. In addition, the nitrogen-fixation pathway is one of the largest genomic segments in which the gene order between $\alpha$ and $\beta$ is conserved (*Bhaya et al., 2007*), suggesting all genes were transferred simultaneously. The most likely explanation is that at least one of the ancestors acquired the ability to fix nitrogen after it colonized the Yellowstone caldera and that this gave it a substantial selective advantage. The genetic diversity within the pathway genes gave further clues about the hybridization process. The diversity within the trough was similar in both species ($\pi_S = 0.052$ in $\alpha$ and $\pi_S = 0.039$ in $\beta$) and was close to the average diversity within $\beta$ ($\pi_S = 0.042$), suggesting the $\beta$ ancestor already had the ability to fix nitrogen when it colonized the caldera. However, the trough diversity was more than an order of magnitude larger than the diversity within the $\alpha$ backbone (*Figure 6A, B*). This large discrepancy strongly suggests $\alpha$ acquired the pathway from $\beta$ through multiple independent transfers after they colonized the Yellowstone caldera.

We performed a similar analysis of diversity within simple hybrid genes to determine the direction and timing of transfers. We focused our analysis on $\alpha$, where we had more data. Presence of hybrid genes at high frequencies suggests the hybridization process may be driven by selection. Two distinct patterns of diversity could result. If a single allele was transferred into $\alpha$ and then underwent a selective sweep, we would see very little diversity among the hybrid alleles. This is known as a hard sweep. Alternatively, if multiple hybrid alleles were transferred into $\alpha$ while undergoing a sweep, the diversity would be comparable to that in the host species. This is the pattern we observed in the divergence troughs and is known as a soft sweep.

We found evidence that both hard and soft sweeps contribute to hybridization in the population (*Figure 6C*). Around 15% of $\alpha$ simple hybrids (31 out of 153 genes) had no diversity at all, regardless of the donor species, consistent with hard sweeps driving their spread. In the remaining 85% (122 out of 153 genes), diversity within $\alpha$ was 1–2 orders of magnitude higher than the $\alpha$ backbone diversity. As a result, the diversity could not be due to mutation accumulation since the common $\alpha$ ancestor, but must have originated from multiple independent transfers—that is, a soft sweep. We further verified the presence of soft sweeps within $\alpha$ by comparing the diversity of individual genes. In most cases, the diversity within $\alpha$ at these loci was similar to $\beta$, with a handful of cases of $\alpha$ having significantly less, consistent with a subset of $\beta$ alleles sweeping through the population simultaneously (*Figure 6C*).

## Recombination within hybrid genes leads to extensive genomic mixing on short genomic length scales

Selective sweeps of individual genes could also lead to mixing of other genomic segments from the donor species into the host population through genetic hitchhiking. If the diversity within the two hybridizing species is low, this process would result in blocks of tightly linked SNPs consisting of primarily two haplotypes, one from the host and another from the donor (*Figure 7A*). Anecdotally, we frequently observed such patterns in mosaic hybrid genes (*Figures 4C and 7A*). We developed a systematic method to search for SNP blocks based on clustering the SNP linkage matrix. Briefly, we search for consecutive SNPs that have perfect linkage (see Appendix 6). We used $\alpha$ hybrid genes to

calibrate our method and chose very conservative parameters to avoid false positives. Consistent with this, only around 17% of SNPs from the $\alpha$ hybrid genes belonged to blocks.

Despite the very conservative thresholds used in our algorithm, we found over one thousand SNP blocks in each species, scattered across hundreds of genes, including in highly conserved ribosomal genes and genes in which distinct species clusters could be assigned (*Figure 7BC*; *Appendix 6— figures 1 and 2*). Most blocks were short, with typical lengths around 70 bp in $\alpha$ and around 30 bp in $\beta$ (*Appendix 6—figure 1*). The divergences between SNP block haplotypes were similar to that between species, consistent with hybridization (*Appendix 6—figure 3*). To confirm their hybrid origin, we mapped the haplotypes from each block to the two other species in our sample. We found 66% (406 out of 605) of SNP blocks within $\alpha$ and 15% of those within $\beta$ (128 out of 776) contained a haplotype identical to consensus of other species (*Figure 7B*). Note that we only considered blocks in which all three species were present for this comparison. In addition, we found a comparable number of blocks that were statistically similar but did not match the other species $\alpha$, $\beta$, and $\gamma$ (*Figure 7B, C*). The most likely explanation is that these blocks resulted from hybridization with other species, similar to the whole-gene hybrids found previously (*Figure 5*).

How much linkage is maintained between blocks? Given the relatively high divergences between species, one might expect epistatic interactions between blocks to constrain the genotypic structure of the population. Conversely, if recombination is dominant, each genome would be a mosaic of segments from the two ancestral species. To answer this question, we calculated the linkage coefficients between haplotypes for each pair of blocks and compared their distribution to a fully unlinked

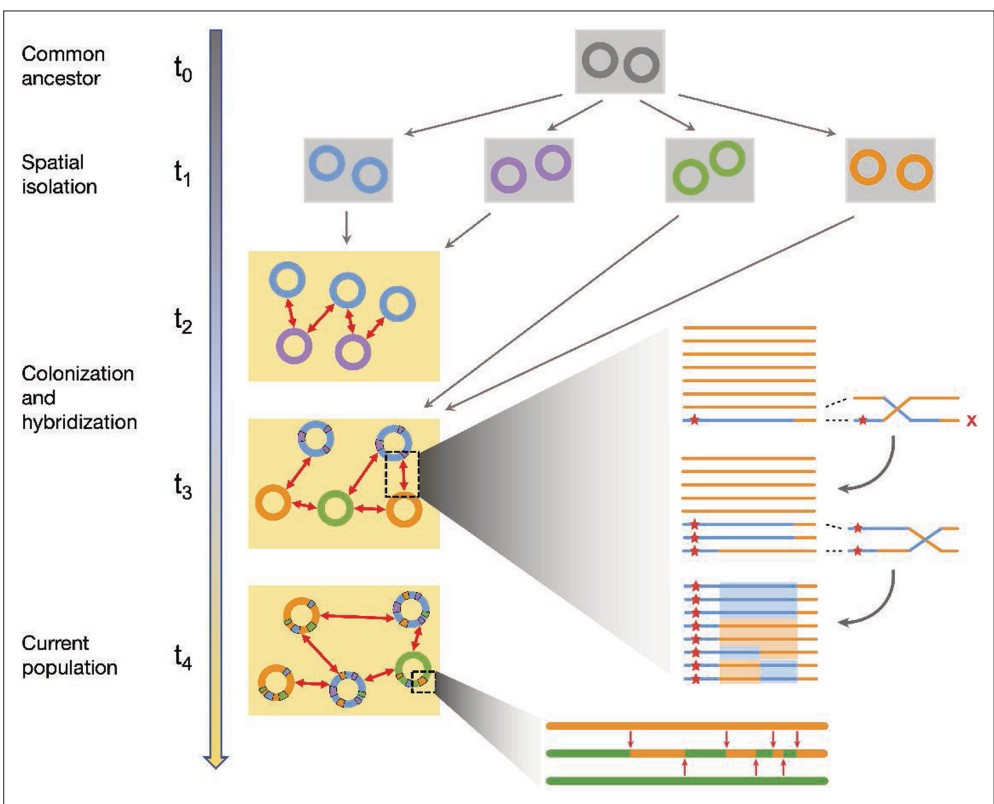

**Figure 8.** Hybridization between previously isolated lineages is the main source of diversity within the population. (Main panel) An illustration of the evolutionary scenario consistent with the main results of this paper. A single common ancestor (gray) for the population ($t_0$) diverged into several species (four shown here) following spatial isolation ($t_1$). The species colonized the Yellowstone caldera, represented by the yellow rectangle, at different times ($t_2 - t_3$) and began hybridizing. This process continued up to the present ($t_4$). (Top inset) Illustration of the mechanism for generating unlinked SNP blocks through recombination between a hybrid segment (blue) containing a beneficial region (red star) and the rest of the host species (orange). Note the four distinct haplotypes highlighted at the end. (Bottom inset) Zoom in of the ancestry of a particular genome (middle) compared with the donor (orange, top) and host (green, bottom) ancestral genomes. After the initial hybridization, recombination with other host genomes (red arrows) creates a mosaic structure across a range of genomic length scales.

control (see Appendix 6 for details). Within the $\beta$ species, the two distributions were almost identical, implying that the genomes are consistent with a random mixture of block haplotypes, as was previously proposed (***Rosen et al., 2015***). Within $\alpha$ the linkage coefficients were statistically larger than those predicted from the unlinked control (***Figure 7D***), but the overall value was still low, with $\left\langle r^2 \right\rangle \approx 0.12$ for data compared to $\left\langle r^2 \right\rangle \approx 0.05$ for unlinked control. These results are consistent with the slower rate of linkage decrease within $\alpha$ compared to $\beta$ discussed previously. We also found similar results for the linkage between hybrid gene alleles (Appendix 4). Thus, rather than finding clonal subpopulations, every one of the ~300 cells in our sample has a unique combination of SNP block haplotypes.

## Discussion

By analyzing over 300 *Synechococcus* single-cell genomes, we show that hybridization plays a major role in shaping their genomic diversity. Our results suggest a simple scenario for the evolutionary history of the population (***Figure 8***; see also Appendix 7). Multiple strains were geographically separated and evolved independently into distinct species from an ancestral population. They then colonized the Yellowstone caldera after its formation around 640,000 years ago and have since been gradually hybridizing. The presence of genes with multiple-sequence clusters and previous analysis of 16S sequences (***Rosen et al., 2018***) suggests at least four independent colonizers, but the current population is dominated by the descendants of only three: the ancestors of $\alpha$, $\beta$, and $\gamma$. The $\beta$ population has been hybridizing with others and recombining with itself long enough that its genomes are a mosaic of different ancestral segments, consistent with a quasisexual population. But $\alpha$ still exhibits a large fraction of low-diversity genomic segments, interlaced with clearly hybridized segments. Molecular clock estimates using the low-diversity regions of the $\alpha$ genomes suggest it emerged around $10^{4-5}$ years ago and on these time scales the population is well-mixed across springs (***Appendix 9—figure 1***). Using the hybridized genomic segments of $\alpha$ as a control, we also estimated the contribution of de novo mutations to the $\beta$ diversity. This analysis suggests that the ancestor of $\beta$ colonized the Yellowstone caldera soon after its formation (Appendix 7).

Diversity within species is often modeled using neutral processes (***Didelot and Wilson, 2015***; ***Rocha, 2018***; ***Chen et al., 2022***). But in this *Synechococcus* population, there is overwhelming evidence for the crucial role played by selection. The divergence troughs resulted from selective sweeps through the whole population that likely overcame mechanistic barriers to recombination, genetic incompatibilities, and ecological differences, evident in the apparent preferences to different temperatures of the three species. These results are consistent with previous analyses showing that recombination events between pathogenic *Streptococcus* species were primarily driven by selection (***Lefébure and Stanhope, 2007***; ***Shapiro et al., 2009***). Moreover, the thousands of blocks composed of tightly linked SNPs we observe are difficult to explain through neutral drift, due to the long time it would take for them to reach observable frequencies. Instead, genetic hitchhiking during selective sweeps is by far the most likely explanation.

Selective sweeps of single genes through subpopulations have been argued to lead to the formation of separate ecotypes, recombination barriers, and eventual speciation (***Croucher et al., 2011***; ***Shapiro et al., 2012***). But we find multiple cases of genes from one species partially sweeping through another, sometimes via several independent transfer events. Thus, gene sweeps do not necessarily lead to ecological specialization, but can do the opposite, acting to homogenize the population. Overall, the population shows abundant evidence of hybridization on a wide range of genomic scales leading to both homogenization between species, through the erosion of genomic clusters, and diversification within species.

Our results suggest several directions for future work. A significant fraction of SNP blocks in $\alpha$ and the majority of those in $\beta$ were transferred from species not present in our samples. But further work is needed to determine the precise number of hybridizing species and whether some may still be present at high abundances in other springs. Metagenome samples from a wider range of springs would provide valuable information for addressing this question (see Appendix 3). Another important question is how do variants reach observable frequencies in the population. Our analysis revealed direct evidence for genetic sweeps at several hundred loci within divergence troughs and simple hybrid loci. But these loci are only a lower bound on the effect of the selection within the population.

A more quantitative analysis of SNP statistics would provide important insights into evolutionary dynamics at these intermediate time scales, which could be informative for understanding bacterial evolution more broadly.

Whether the erosion of genomic clusters we observe would continue indefinitely remains an open question. The hybridization process appears to be very slow: we estimate that the genomic mixing we observe occurred over at least $10^4$ years and likely started soon after $\alpha$ became abundant (see Appendix 7). It is therefore possible that the current population is an evolutionary transient that would eventually lead to a single hybrid population. This scenario would be contrary to the prevailing view that microbial evolution leads to increased specialization (*Shapiro et al., 2012*; *Jain et al., 2018*). Distinguishing between these two scenarios would require a detailed understanding of how the diversity changes during the hybridization process. For example, a better theoretical understanding of how SNP correlations within and between SNP blocks change after an initial transfer could generate testable predictions about single- and multi-site frequency spectra that could be checked in the data. Some statistical features will likely be more sensitive to selective pressures that act at the whole-genome level and could help maintain clusters indefinitely. Similar approaches could shed light on the evolutionary history of other highly recombining bacteria, such as *H. pylori* or *SAR11* (*Vergin et al., 2007*; *Kennemann et al., 2011*). Thus, how extensive hybridization affects, and is affected by, ecology and how the interplay between selection at both gene and genome scales shapes the diversity of long-term coexisting populations are intriguing questions for future work.

The importance of hybridization in eukaryotic evolution is increasingly being recognized, but there have been few investigations in prokaryotes (*Sankararaman et al., 2014*; *Moran et al., 2021*). Metagenomic studies on communities in acid mine drainages revealed several hybrid bacterial and archaeal populations whose genomes were mosaics of hundreds of kbp segments from two different ancestors (*Tyson et al., 2004*; *Lo et al., 2007*). Those results are consistent with very recent hybridization, after the start of mining operations (*Denef and Banfield, 2012*). Our study shows the effects of this process over much longer time scales. At the level of individual genes, evidence for hybridization has been observed among commensal and pathogenic species of *Campylobacter* (*Sheppard et al., 2008*; *Mourkas et al., 2022*). But unlike the thermophilic *Synechococcus*, such host-associated bacteria are globally dispersed and have likely sampled many different environments, on a wide range of time scales, during their evolutionary history. It is thus difficult to identify which evolutionary forces drive their hybridization.

For the Yellowstone *Synechococcus*, a fortuitous combination of spatial separation and later mixing on the same time scales on which speciation and then hybridization can extensively shape—but not completely overwrite—their diversity has enabled us to infer a great deal about the evolutionary history and the effects of full and partial sweeps. This provides a window into processes that shape species-level bacterial diversity. But for most bacterial populations, the effects of these processes are obscured by the broad range—and little knowledge—of the time scales on which different strains have co-occurred and are able to exchange DNA. Investigating whether hybridization plays a major role in other communities will thus require developing a predictive theory that can distinguish between possible evolutionary scenarios.

## Acknowledgements

We thank Benjamin H Good, Alana Papula, and Freddy Bunbury for helpful comments and discussions. GB and DSF gratefully acknowledge support from the Simons Foundation via a Postdoctoral Fellowship Award 730295 to GB and Sabbatical Fellowship to DSF. This work was also supported by NSF Grants PHY-1607606 and PHY-2210386 to DSF. HSM was supported by the National Institutes of Health grant R01-AI-100947 to Mihai Pop. DB acknowledges support from the NSF/BIO-BBSRC collaborative research grant 1921429, NSF MIM grant 2125965, Joint Genome Institute Community Sequencing Project 503441 and 509352, and the Carnegie Institution for Science. Samples processed for single-cell genome amplification were collected using NP Park Permits: YELL-5494 to David Ward (multi-year), YELL-5660 to DB (2007–2008), and YELL 5694 to DB (2007–2009). The work (proposal: https://doi.org/10.46936/10.25585/60001132) conducted by the U.S. Department of Energy Joint Genome Institute (https://ror.org/04xm1d337), a DOE Office of Science User Facility, is supported by the Office of Science of the U.S. Department of Energy operated under Contract No. DE-AC02-05CH11231.

## Additional information

### Funding

| Funder | Grant reference number | Author |
| --- | --- | --- |
| Simons Foundation | 730295 | Gabriel Birzu<br>Daniel S Fisher |
| National Science Foundation | PHY-1607606 | Daniel S Fisher |
| National Science Foundation | PHY-2210386 | Daniel S Fisher |
| National Science Foundation | 1921429 | Devaki Bhaya |
| National Science Foundation | 2125965 | Devaki Bhaya |
| Joint Genome Institute | 503441 | Devaki Bhaya |
| Joint Genome Institute | 509352 | Devaki Bhaya |
| Simons Foundation | Sabbatical Fellowship | Daniel S Fisher |
| Carnegie Institution for Science | | Devaki Bhaya |
| National Institutes of Health | R01-AI-100947 | Harihara Subrahmaniam Muralidharan |
| United States Department of Energy | 10.46936/10.25585/60001132 | Devaki Bhaya |

The funders had no role in study design, data collection, and interpretation, or the decision to submit the work for publication.

### Author contributions

Gabriel Birzu, Conceptualization, Investigation, Methodology, Writing – original draft, Writing – review and editing; Harihara Subrahmaniam Muralidharan, Investigation, Writing – review and editing; Danielle Goudeau, Resources, Data curation; Rex R Malmstrom, Resources, Data curation, Methodology, Writing – review and editing; Daniel S Fisher, Conceptualization, Supervision, Funding acquisition, Methodology, Writing – original draft, Writing – review and editing; Devaki Bhaya, Conceptualization, Funding acquisition, Writing – original draft, Project administration, Writing – review and editing

### Author ORCIDs

Gabriel Birzu https://orcid.org/0000-0003-3561-024X
Rex R Malmstrom https://orcid.org/0000-0002-4758-7369
Daniel S Fisher https://orcid.org/0000-0002-5559-2491
Devaki Bhaya https://orcid.org/0000-0001-7965-4258

Reviewer #1 (Public review): https://doi.org/10.7554/eLife.90849.3.sa1
Reviewer #2 (Public review): https://doi.org/10.7554/eLife.90849.3.sa2
Author response https://doi.org/10.7554/eLife.90849.3.sa3

## Additional files

### Supplementary files

MDAR checklist

### Data availability

All SAG assemblies and metagenome samples used in this study can be found on the JGI website (https://genome.jgi.doe.gov/portal/) under the project ID 503441. The analyzed data and code used to produce the figures in the main text are publicly available at https://github.com/gbirzu/

yellowstone_cyanobacteria_hybridization (copy archived at *Birzu, 2025*) and https://doi.org/10.5281/zenodo.17534465.

The following dataset was generated:

| Author(s) | Year | Dataset title | Dataset URL | Database and Identifier |
|---|---|---|---|---|
| Birzu G | 2025 | Yellowstone Cyanobacteria hybridization | https://doi.org/10.5281/zenodo.17534465 | Zenodo, 10.5281/zenodo.17534464 |

The following dataset was generated:

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

# Appendix 1

## Materials and methods

### Sample collection and sequencing

Cork borers (approximate diameter 0.5–1 mm) were used to collect mat core samples from Octopus Spring (4453401N, 11079781W) and Mushroom Spring (4453861N, 11079791W). Samples were collected in 2005, 2006, 2007, and 2009 under YNP Park Permits (YELL-5494, YELL-5660, YELL 5694). Cores were immediately frozen until later use. Frozen samples were shipped to the Joint Genome Institute (JGI) in 2019, where they were processed and sequenced.

At JGI, for disaggregation of sample, frozen mat cores were cut into pieces using a razor blade on glass over dry ice and aliquoted chunks of approximately 5 mm diameter. Next, 500 µl PBS and 50 µl of 500 mM NaEDTA were added. Samples were vortexed briefly, sonicated for 1 min on benchtop sonicator, and vortexed briefly again. Passed 25 times through a 25-gauge needle and run through a 35-µm strainer cap. Next, 100 µl were placed in 900 µl PBS and stained with SYBR Green at 1X. Samples were run on the Influx with Gate SYBR+ Cells and Gate 670/30 off of 632 autofluorescence. Next, samples were sorted on a BD influx following the protocol outlined in *Rinke et al., 2014*, and single-cell genomes were amplified with WGAX protocol as described in *Stepanauskas et al., 2017*. Certain samples from plate YuBhay.Octo06.RedA.3 received an additional treatment with lysozyme (20 min at room temp in 50 U/µl prior to alkaline lysis) because of difficulty lysing the cells. Otherwise, they followed the standard alkaline lysis described in *Stepanauskas et al., 2017*. In brief, libraries were made with Nextera XT v2 kits and sequenced on an Illumina Novaseq in 2 × 150 bp mode. Reads were assembled into contigs and annotated using the Integrated Microbial Genomes (IMG) platform as described in *Rinke et al., 2013*. All of the data generated for this study is available on the JGI website, under the project ID 503441.

### Data processing

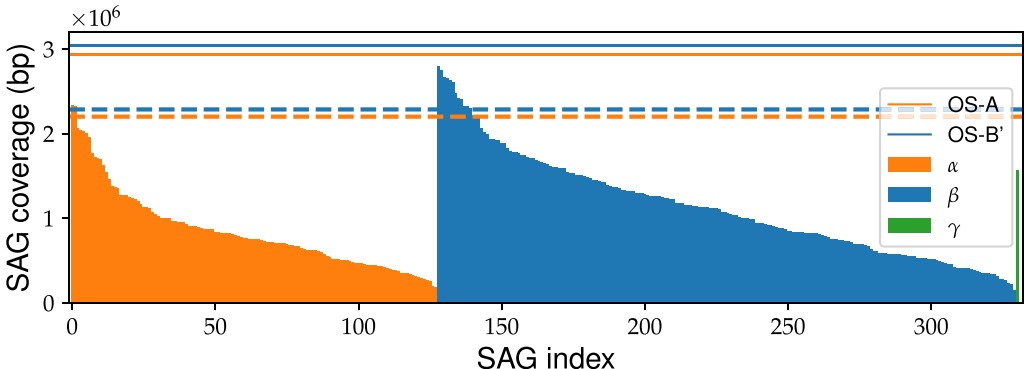

**Appendix 1—figure 1.** The genome coverage across single-amplified genomes (SAGs) is highly variable. The SAG coverage, defined as the total length of contigs, is shown for all 331 SAGs (after quality control). SAGs were grouped by species (see Appendix 2) sorted in descending order of coverage within each species. The solid orange and blue horizontal lines show the genome sizes of OS-A and OS-B'. The dashed orange and blue lines show the threshold used to define high-coverage SAGs. The values of the thresholds are set at 75% of the length of the reference genomes.

In this section, we describe the procedure we used to process the single-amplified genomes (SAGs). Our analysis is based on grouping genes into orthologous gene clusters or *orthogroups*, which are then analyzed separately. This procedure makes it much easier to analyze the fragmented and incomplete genomes resulting from single-cell sequencing (*Appendix 1—figure 1*) compared to multiple-sequence alignment (MSA) of entire genomes, which are sometimes used for analyzing genomes obtained from isolates. An alternative approach is to align the assemblies or even the raw reads to reference genomes. In our case, there are only two reference genomes available, OS-A and OS-B', so aligning to them would likely result in biases in the inferred diversity.

There are several ways of constructing orthogroups: (1) pairwise comparisons or (2) graph methods. Graph methods have the advantage that they produce sets of mutually homologous genes so we decided to use this approach. There are several packages available for identifying homologous gene clusters among samples of different genomes. Most widely used packages, such as OrthoDB (*Kriventseva et al., 2007*) and OrthoFinder (*Emms and Kelly, 2015*), were designed for finding homologs between different eukaryotic species and are inappropriate for studying large bacterial pangenomes (*Ding et al., 2018*). More recently, several groups have developed packages that allow for efficient construction of large bacterial pangenomes (*Page et al., 2015*; *Ding et al., 2018*). However, such packages are primarily aimed at studying large collections of isolate genomes and were not designed to handle samples with low genome coverage, which are common in single-cell sequencing projects such as ours. As a result, we implemented a custom pipeline for processing the single-cell genome assemblies and constructing the pangenome from these sequences. The pipeline is divided into two parts: (1) a pre-processing step where assemblies are filtered for contamination and checked for misassemblies and misannotations, and (2) a gene clustering step, in which clusters of homologous genes are identified and paralogs are separated from true orthologs. The second half of the pipeline closely follows the approach taken in *Ding et al., 2018*. A more detailed outline of the pipeline is given below. For details on the assignment of sequence clusters within orthogroups to different species, see Appendix 5.

## Outline of data processing pipeline

1. Data pre-processing
2. Orthogroup clustering
3. Orthogroup MSA
4. Deep branch splitting
5. Orthogroup species clustering
6. Reference genome mapping

The rest of this section explains how each step in the pipeline was performed. Readers not interested in the details may wish to move on to the next section.

### Data pre-processing

The goal of this step is to remove (1) SAGs from non-*Synechococcus* OS species, (2) SAGs that might have resulted from sequencing multiple cells in the same reaction well, and (3) contigs that might be the result of contamination from external DNA or other sequencing wells. We first removed all SAGs that were positive controls in which multiple cells were amplified in the same reaction well (all SAGs from the third column of the 384-well plate in our case). We then aligned all of the genes in our data to the OS-A and OS-B′ reference genomes using BLAST (alignment parameters: `-evalue 1E-10 -word_size 8 -qcov_hsp_perc 75.0`). We then removed all SAGs with >50% of genes that did not map to either reference genome. The remaining SAGs are assumed to be from *Synechococcus* OS cells. Next, we filtered out all contigs without any hits to OS-A and OS-B′. This was done to avoid external DNA or cross-contamination from affecting the rate of hybridization we observed.

Next, we removed any remaining SAGs that were suspected to have been the result of sequencing multiple cells in the same well. We reasoned that sequencing multiple cells would lead to an increase in the number of duplicated genes in the SAG. We used the probability of sampling a gene with more than one copy in the two reference genomes, both ~10%, as a benchmark and removed all SAGs with a multicopy frequency >20%.

Preliminary analysis of the data following the above filtering procedure revealed the existence of a small number of palindromic contigs, in which the first half of the sequence was mirrored exactly in the second half, that were very likely the result of assembly artifacts. To remove these artifacts, we labeled the genes in each contig based on the hits to the reference genomes and used these to search for palindromic sequences within each contig. We then removed any contig that contained a palindromic subsequence that was either longer than four genes or contained >50% of genes in the contig. Finally, we removed all partial genes from the remaining contigs to simplify the analysis of the multiple-sequence alignments (MSA), which we performed next.

## Orthogroup clustering

The goal of this step is to obtain clusters of orthologous genes or *orthogroups*, which were used in subsequent analysis. To do this, we followed (*Page et al., 2015*; *Ding et al., 2018*) and used a Markov clustering algorithm to find groups of sequences that were similar to each other and well-separated from all other sequences (*Enright, 2002*). The Markov clustering algorithm takes as input a similarity graph, where each node represents a gene sequence and the weight of each edge represents the similarity between the two sequences.

We constructed the gene similarity graph by translating all of the gene sequences in the filtered dataset and performing an all-to-all protein alignment using BLAST (alignment parameters: `-evalue 1E-3 -word_size 3 -qcov_hsp_perc 75.0`). Other approaches either perform a pre-alignment clustering (*Page et al., 2015*) or use faster but less sensitive methods for all-to-all alignments (*Ding et al., 2018*), but we found that this was not necessary in our case. The higher sensitivity provided by BLAST could help distinguish between paralogs within large gene families and minimize gene misassignments, but we did not extensively test how the performance would be affected by using other methods. We used the bitscore from the alignments to define the weight of each edge in the similarity graph. Using the bitscore instead of the $e$-value is a simple way to correct for the minimum $e$-value of $1E-180$ in BLAST and has been shown previously to improve the clustering performance (*Gibbons et al., 2015*).

We used MCL (v. 14-137) to cluster the resulting similarity graph and obtain orthogroups (*Enright, 2002*). We used a relatively small inflation parameter (-I 1.5) in order to cluster genes aggressively and used subsequent steps in the pipeline to split clusters resulting from incorrect cluster assignments or the presence of paralogs (steps 5 and 6). As a first step, we did an additional clustering based on sequence length as follows. We constructed a graph in which all sequences in an orthogroup were represented as nodes and edges were attached between pairs of sequences if the difference between their lengths was less than 30% of their average length. We then identified all connected components in this graph. Because homologous genes in OS-A and OS-B′ sometimes have different lengths, we hypothesized that each orthogroup should contain at most two main clusters. We therefore used the following heuristic algorithm to filter out orthogroups with unusual gene length patterns that could possibly result from overly aggressive clustering. If a single-length cluster was present or the largest cluster contained >80% of sequences, only the sequences from the largest cluster were kept and the rest were removed. If two clusters accounted for >80% of sequences, only the sequences in those two clusters were kept and the rest removed. In all other cases, we removed the orthogroup from our dataset and did not include it in subsequent analysis.

## Orthogroup MSA

The filtered orthogroups from the previous step were aligned using a custom codon-aware alignment algorithm. Briefly, nucleotide sequences were translated into protein sequences and aligned using MAFFT (v. 7.475 with default parameters) (*Katoh and Standley, 2013*). The alignments were then reverse-translated using nucleotide sequences as templates. After alignment, phylogenetic trees for each orthogroup were constructed using FastTree (v. 2.1.11 with parameters `-nj -noml`) (*Price et al., 2010*).

## Deep branch splitting

In this step, we separated gene clusters within the initial orthogroups that were joined by unusually long branches in the phylogenetic tree. Specifically, we implemented the following algorithm. For each orthogroup, we checked the branch length for each node in the phylogenetic tree and grouped any sequences with a branch length greater than a cutoff value $b_c = 0.3$ into separate orthogroups. Each newly created orthogroup was realigned as in step 3, and the procedure was repeated until no new orthogroups could be separated. We used the results from previous analysis of homologous genes in OS-A and OS-B′, which showed that typical divergences were ~0.15 (*Rosen et al., 2018*), as a benchmark to set $b_c$. Based on those results, we expected very few genuine homologs with divergences above $b_c = 0.3$. The clusters that resulted after this step represent the final orthogroups that were used in subsequent analysis.

## Orthogroup species clustering

We next sought to determine which orthogroups could be reliably partitioned into distinct species clusters, which would be expected if the different species asexually diverged from a common ancestor. To do this, we implemented the following algorithm in a custom Python script. For each orthogroup, the alignments were trimmed to remove excess gaps at the edges using a custom script. We then calculated the fraction of sites with different nucleotide between each pair of sequences and hierarchically clustered the sequences using average linkage as implemented in the Python Scipy package (v. 1.9.3) (*Virtanen et al., 2020*). Preliminary clusters were assigned using a distance cutoff $d_c = 0.075$ (*Appendix 1—figure 2*). To ensure that the clusters were well-separated and robust, we implemented an additional check by requiring that the minimum distance between any two sequences from different clusters be at least $c = 1.5$ times greater than the maximum distance between any two sequences within either cluster. Any clusters that did not satisfy this criterion were designated as *mosaic orthogroups*. We validated the robustness of the cluster assignments by $d_c$ and c and found only small changes in the results.

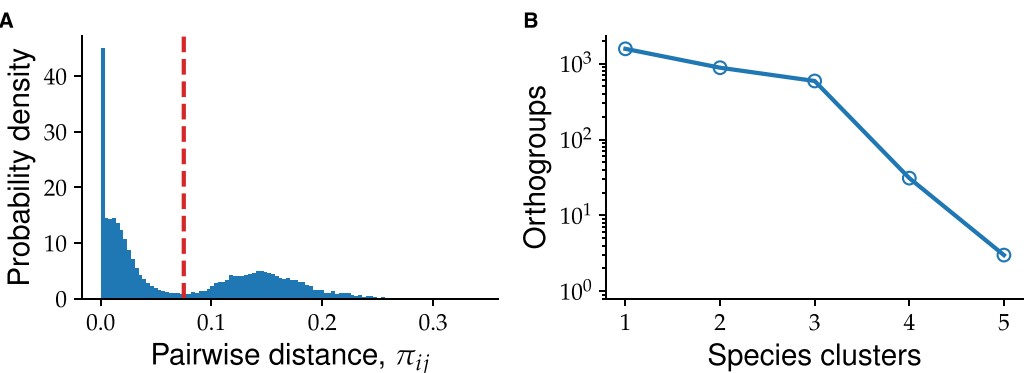

**Appendix 1—figure 2.** Distribution of pairwise nucleotide distances within orthogroups shows clear gap around $\pi_{ij} \approx 0.075$. (**A**) Histogram of nucleotide distances between all pairs of cells across the 1448 core orthogroups. The dashed red line shows the distance cutoff used to define species clusters ($d_c = 0.075$). (**B**) Distribution of orthogroups according to the number of species clusters. Species clusters were generated as described in the text using the distance cutoff shown in (**A**).

Manual examination of the mosaic orthogroup alignments revealed a small fraction with a large number of gaps. Such alignments could be the result of rapidly evolving genes in the presence of hybridization, but may also result from misalignments or the inclusion of non-orthologous sequences during the orthogroup construction. To simplify subsequent analysis, we therefore removed mosaic orthogroups with an unusually high fraction of gaps.

## Mapping to reference

For this step, we use reciprocal best hit (RBH) given by BLAST to map the consensus sequences to the OS-A and OS-B′ reference CDS. Since at this point in the pipeline, most sequences within one OG are less than 10% diverged from each other, we use nucleotide sequences for this step to increase the accuracy of the mapping. The consensus sequence for each OG is constructed by finding the consensus codon at each residue along the CDS after replacing all codons containing gaps by a gap codon. The resulting sequences that have RBH to OS-B′ are mapped onto those respective genes. The genes without RBH to OS-B′ are searched against OS-A and mapped to it if an RBH is found.

## Appendix 2

### SAG species assignment

In this section, we outline the method we used to assign genomes to each species cluster. We also verify the robustness of the species assignment by using three different methods and show that they give identical results.

Comparisons of genomes from bacterial populations often reveal clusters at different length and divergence scales (*Jain et al., 2018*). But the interpretation of such clusters is controversial (*Cohan, 2002*; *Fraser et al., 2009*). There are three main possible interpretations for such patterns. One interpretation is that the clusters represent the bacterial analog of eukaryotic species, which are sexually and ecologically isolated from each other. In practice, a whole-genome divergence cutoff of 5% is frequently used to define bacterial species (*Jain et al., 2018*). An alternative interpretation is that clusters are the result of clonal expansions of particular strains from a much more diverse and possibly more genetically homogeneous population. The third interpretation is that the clusters are the result of uneven sampling from a larger and more diverse population. The unique combination of having a large collection of genomes, with minimal compositional bias, from a geographically isolated population allows us to test these alternatives. We will mainly aim to distinguish between the first two interpretations and the third one by checking whether there are robust genomic clusters in our sample. More discussion on the evidence concerning the first two hypotheses can be found in Appendices 3 and 5.

We used three different methods to check whether the genomes form robust clusters: the phylogenetic tree of the 16S rRNA sequences, average genome-wide divergence from the two reference genomes OS-A and OS-B', and patterns of divergences between triplets formed by the SAG and the two reference sequences at different loci. The phylogenetic tree of the 16S is the simplest and still the most commonly used method to characterize the diversity of natural microbial communities (*Yarza et al., 2014*). Because of its practical importance, we included it for comparison with the other two methods. However, in a highly recombining population like the Yellowstone *Synechococcus*, it is not clear to what extent the 16S divergence accurately reflects the whole-genome diversity. To address this question, we used the divergences across the entire genome as a comparison. The main limitation of using summary statistics such as the average divergence across the entire genome is that it obscures heterogeneities along the genome that could have important ecological and evolutionary implications. There are different ways to address this concern. Here, we used an extension of a method first used in *Rosen et al., 2018*, which provides a coarse description of the variation in the diversity patterns across different loci. We present the results from each of these analyses next.

We first constructed the phylogenetic tree of 16S sequences. Predicted 16S rRNA sequences (117 from 331 filtered SAGs) were extracted and aligned using MAFFT (v7.453) (*Katoh and Standley, 2013*). Two sequences (SAGs MA02M11_3 and MuA02C9) were incomplete and were removed from the analysis. The remaining (115) sequences were hierarchically clustered using the average linkage (UPGMA) method implemented in Scipy. Results are shown in *Appendix 2—figure 3*, with the colored background showing the species assignment using whole-genome divergences (see below) for comparison. We found perfect agreement between the whole-genome assignment and the largest three clusters in the 16S phylogenetic tree.

The average divergence within the $\alpha$ cluster is approximately half that of $\beta$ ($1 \cdot 10^{-3}$ vs $2 \cdot 10^{-3}$). However, a significant fraction of the average divergence within $\beta$ was due to two outliers (MuA1L16 and MusA1J8). After removing the outliers, the average divergence within $\beta$ decreased to $1.7 \cdot 10^{-3}$. Closer investigation of the alignments revealed that both outlier $\beta$ sequences contained a short segment of three SNPs that were identical to all $\alpha$ sequences and different from all other $\beta$ sequences, at positions 1067–1072. In addition, MuA1L16 contained a larger segment of almost 200 bp (positions 390–570) containing 10 SNPs and an insertion that were different from all other $\beta$ and were identical to the consensus $\alpha$. Apart from these segments, the outlier sequences were typical of the $\beta$ species. This pattern is similar to the SNP blocks discussed in the main text and in Appendix 6 and therefore is likely the result of hybridization between $\alpha$ and $\beta$.

We compared the 16S rRNA clusters to whole-genome divergences from the two reference genomes, OS-A and OS-B'. We aligned all of the predicted genes from each SAG to all of the

coding DNA sequences (CDS) from each reference genome using BLAST (v2.13.0+) (**Altschul et al., 1990**). For each SAG, we calculated the divergence to each reference genome ($d = 1 - \text{pident}/100$) and calculated the average over the best hits across all genes. The results for all SAGs with more than 100 hits to both references are shown in **Figure 1A**. Three clear clusters were observed, which we labeled as $\alpha$, $\beta$, and $\gamma$. The species of each SAG was assigned as the label corresponding to the cluster it belonged to in **Figure 5A**.

The previous two metrics do not reflect possible heterogeneity in the patterns of diversity along the genome. To characterize this variation on a gene-by-gene basis, we extended a method that was previously used to analyze amplicon data from the Mushroom Spring in **Rosen et al., 2018**. We refer to this method as *gene triplet analysis*. This analysis exploits the separation of scales between and within species clusters. At most loci and in most cells, we observed previously in order to coarse grain the high-dimensional matrix of pairwise divergences into a small number of discrete patterns. Specifically, we take triplets of sequences from the SAGs, OS-A, OS-B', at loci where all three are homologous and calculate the divergences between all three pairs of sequences. These divergences typically form a triangle as shown in **Appendix 2—figure 1**. Note that it is possible to have triplets of divergences that violate the triangle inequality, which cannot be represented as triangles in a two-dimensional space. In our data, such cases were rare, but this may not be generally true, particularly for sequences that are not clearly separated and which are the result of extensive recombination. Note, however, that the classification into distinct patterns given below can still be done even in these cases. To reduce the dimensionality of the data further, we define a main cloud diameter $d_M = 0.1$ for the diversity within species. We then classify the shape of the triangle into the 5 distinct patterns shown in **Appendix 2—figure 1**, depending on whether each side of the triangle is longer or shorter than $d_M$. For example, denote the divergence triplet by $(d_{XA}, d_{XB}, d_{AB})$, representing the divergences between the SAG sequence and OS-A, the SAG sequence and OS-B', and OS-A and OS-B', respectively. Then the first pattern in **Appendix 2—figure 1A** (labeled 'XA–B') contains loci where $d_{XA} < d_M$, $d_{XB} \geq d_M$, and $d_{AB} < d_M$, the second pattern (labeled 'XAB') contains loci where $d_{XA} < d_M$, $d_{XB} < d_M$, and $d_{AB} < d_M$, and so on. The cutoff of $d_M = 0.1$ is the same as the one used in **Rosen et al., 2018** and provides a good separation between species at a gene by gene basis. The results of our analysis, however, do not depend on the specific value of $d_M$ as discussed at the end.

For each SAG, we assigned all genes that were orthologous to OS-A and OS-B' to one of the five patterns described above. The three rows in **Appendix 2—figure 1** show typical examples of an $\alpha$ (**Appendix 2—figure 1A**), a $\beta$ (**Appendix 2—figure 1B**), and the $\gamma$ (**Appendix 2—figure 1C**) SAG. The majority of genes across each row in **Appendix 2—figure 1** are found in the XA–B, XB–A, and X–A–B patterns, respectively. This is consistent with the species clusters we saw in the genome-wide average divergences from before. However, we also find a significant minority of genes forming other patterns, most notably XAB, representing genes where all three sequences are within the main cloud ($d < d_M$). Some of these genes were contained in the divergence troughs discussed in the main text. In the case of the $\gamma$ SAG, the fact that X–A–B is the dominant pattern rules out the possibility that $\gamma$ is a hybrid resulting from the mixture of $\alpha$ and $\beta$. Such hybrids have been observed after recent clonal expansions in acid mine drainage microbial communities (**Tyson et al., 2004**; **Lo et al., 2007**), but do not appear to be present at substantial frequency in the Yellowstone community. Instead, it appears that $\gamma$ is a distinct species that diverged from the other two around the same time $\alpha$ and $\beta$ diverged from each other.

Using the pattern described above allows us to compare different SAGs to each other and allows us to test the robustness of the species assignment based on the average genome-wide divergence. Specifically, we assigned to each SAG a gene triple fingerprint $f = (f_{XA-B}, f_{XAB}, f_{X-A-B}, f_{X-AB}, f_{XB-A})$, representing the fraction of genes with each pattern (note that $\sum_i f_i = 1$). We then compared the fingerprints of each SAG with the species assignment based on the average divergence from OS-A and OS-B' from **Figure 1A**. We found that the fingerprints formed three clear patterns that exactly matched the assignments based on the average divergence (**Appendix 2—figure 2**). We tested this assignment for different values of $d_M$ ranging from 1% to 20% and found that the fingerprint patterns were robust across the entire range (data not shown). These results show that the classification into distinct species is robust and consistent across the three methods described.

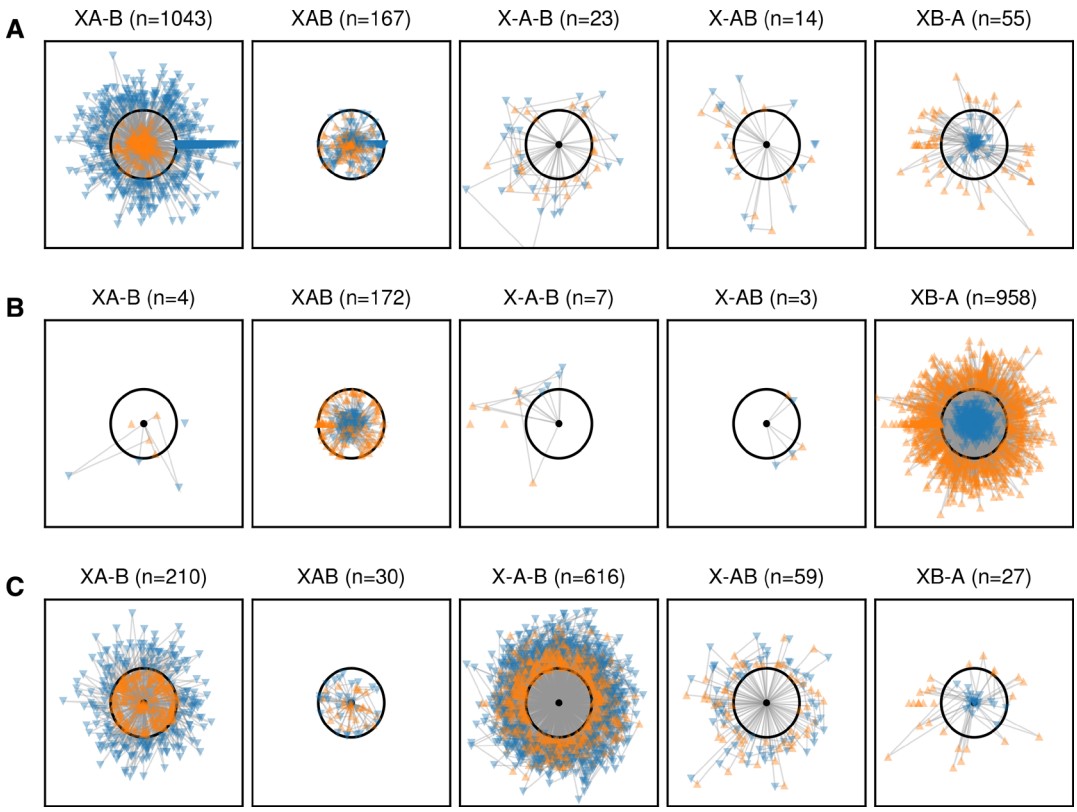

**Appendix 2—figure 1.** Species clusters at the gene level approximately match those at the genome level. Figures show gene-by-gene comparison of three cells to the OS-A and OS-B' reference genomes. The cells shown are representative of the α (**A**; OctA1J9) and β (**B**; MA02H14_2) clusters and the one γ cell (**C**; OcA3L13). Each core gene with homologs in both OS-A and OS-B' was compared to the references and the divergences plotted as a triangle with each side length equal to the divergence between a given pair. The cell sequence is represented by a black dot, with the OS-A and OS-B' sequences represented by the orange upward and blue downward triangles, respectively. The black circle is a 10% radius which was chosen as a cutoff to classify each gene. The different columns show genes in one of the five possible patterns labeled as follows: XA–B (the cell sequence is less than 10% diverged from OS-A and more than 10% diverged from OS-B'), XAB (all three sequences are within 10% of each other), X–A–B (all three sequences are more than 10% diverged from each other), X–AB (the cell is more than 10% diverged from both OS-A and OS-B', but both references are less than 10% diverged from each other), and XB–A (the cell is less than 10% diverged from OS-B' and more than 10% diverged from OS-A).

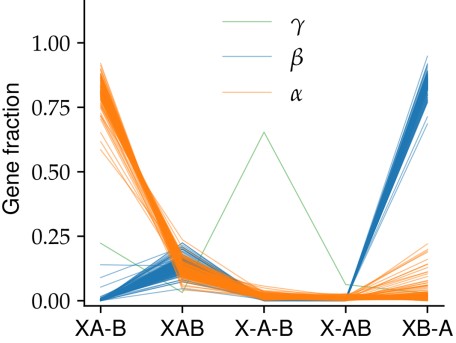

**Appendix 2—figure 2.** The proportion of gene triplet patterns shows three distinct clusters. The proportion of genes in each of the five patterns shown in *Appendix 2—figure 1* is shown for each cell passing our quality control criteria (see Appendix A). Lines represent different cells and are colored according to the species classification based on whole-genome divergences.

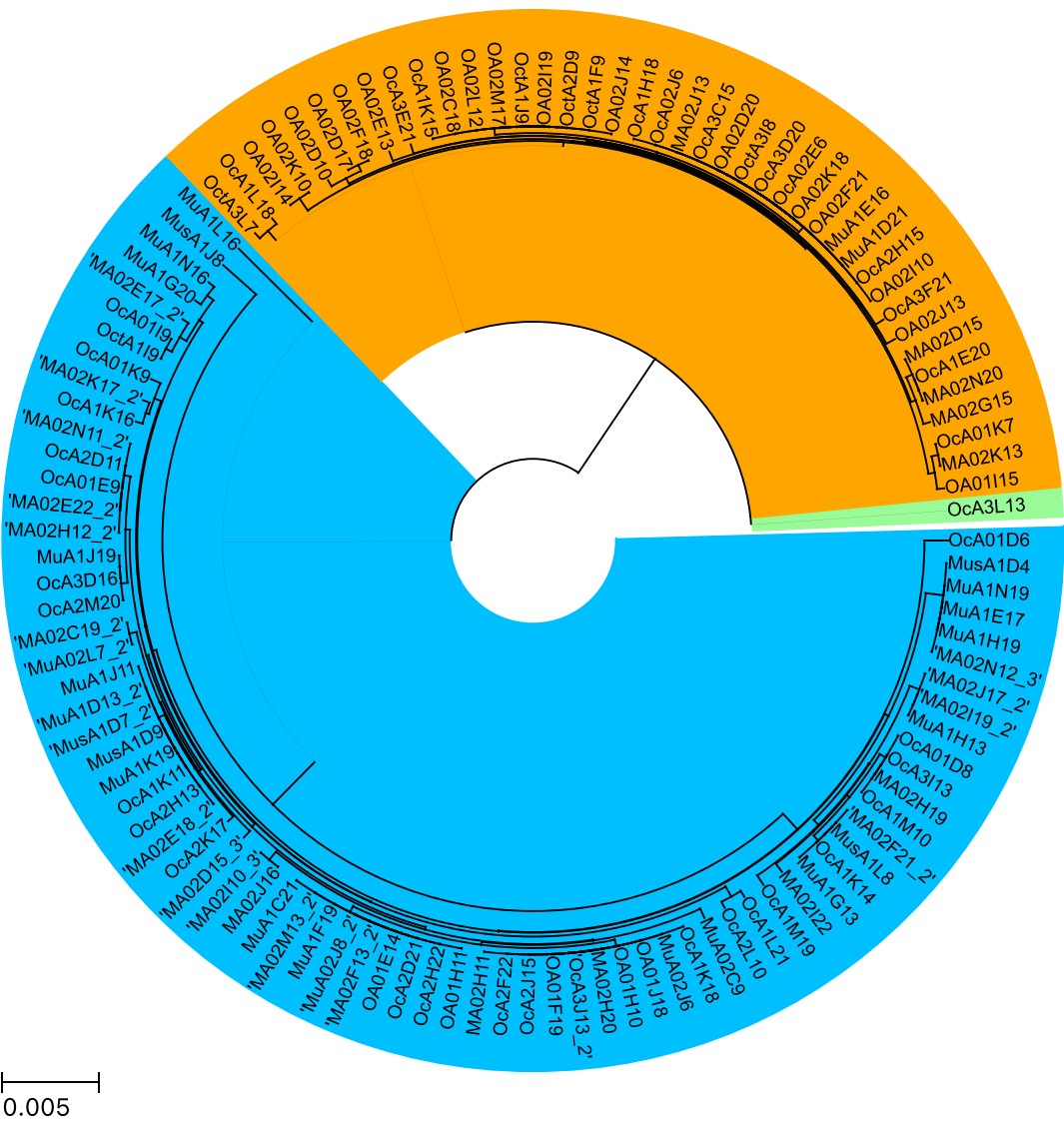

**Appendix 2—figure 3.** Phylogenetic tree inferred from 16S rRNA agrees with whole-genome species classification. The pairwise divergences between all 16S sequences passing quality control were calculated from the multiple sequence alignment performed using MAFFT (*Katoh and Standley, 2013*). The tree was generated using a hierarchical agglomerative clustering algorithm with average linkage (UPGMA) as implemented in SciPy (*Virtanen et al., 2020*).

# Appendix 3

## Metagenome analysis

### Phylogeny of $\gamma$

Here, we present the details of the phylogenetic analysis of the $\gamma$ SAG, OA3L13, which shows that it represents a new uncharacterized species within the Yellowstone cyanobacteria population. To establish its phylogeny, we used a concatenated alignment of marker genes, following **Wu and Eisen, 2008**. Of the 31 marker genes used in **Wu and Eisen, 2008**, 16 were covered in the $\gamma$ SAG (*dnaG, frr, pgk, rplA, rplK, rplL, rplM, rplS, rplT, rpsB, rpsI, rpsJ, rpsK, rpsM, smpB,* and *tsf*) and were used to determine its phylogeny. We used a large collection of cyanobacterial genomes from **Chen et al., 2021** to check for closely related species to $\gamma$. The concatenated alignment was constructed as follows. For each of the 16 marker genes, we extracted the alignment of all species labeled 'Thermal springs' that were within one of the subsections labeled I through V in **Chen et al., 2021** and added the $\gamma$ sequence to the alignment. All '?' characters were replaced with 'X'. The sequences were then realigned using MAFFT (v7.453) (**Katoh and Standley, 2013**) using default parameters and the alignments were concatenated and trimmed using trimal (v1.4.rev22) (**Capella-Gutiérrez et al., 2009**) with default parameters. Finally, the resulting sequences were realigned using MAFFT and the phylogenetic trees were inferred using FastTree (v2.1.11) (**Price et al., 2010**) with default parameters. The resulting tree is shown in **Appendix 3—figure 1**.

To verify that the lack of other closely related genomes to $\gamma$ was not due to the incompleteness of our reference database, we searched for homologs to the 16 $\gamma$ marker genes in the NCBI database. We used the Web version of blast on NCBI with the megablast settings and word size 16. For all 16 marker genes, the top two hits were from OS-A and OS-B', while all other high-scoring hits had less than 85% nucleotide identity. We are also matching sequences to the 16S rRNA using the SILVA database (v138.1) (**Quast et al., 2013**). Using the non-redundant version of the database (SSU Ref NR99) only gave hits to OS-A and OS-B' sequences. Using the full database we did find several sequences that were less than 10 SNPs away from the γ sequence across the entire 16S rRNA (>99.7% identity), including one from a previous study of Yellowstone microbiomes (**Meyer-Dombard et al., 2011**). The corresponding divergence of $3 \cdot 10^{-3}$ is comparable to the typical diversity within species in our SAGs of $1 - 2 \cdot 10^{-3}$ and much less than the typical divergence between species, which is approximately $2 - 4 \cdot 10^{-2}$ (Appendix 2).

**Appendix 3—table 1.** Best hits to OA3L13 16S rRNA in SILVA SSU Ref database.

| SILVA ID | Matches/hit length | Identity | Notes |
|---|---|---|---|
| AFSR01000050 | 1424/1454 | 0.978 | OS and MS metagenome, **Klatt et al., 2011** |
| AY884052 | 1424/1454 | 0.978 | OS-A isolate, **Allewalt et al., 2006** |
| HM448361 | 1382/1386 | 0.997 | **Meyer-Dombard et al., 2011** |
| HM448360 | 1381/1386 | 0.997 | **Meyer-Dombard et al., 2011** |
| KU382141 | 1381/1389 | 0.994 | |
| CP000239 | 1295/1324 | 0.978 | *Synechococcus* OS-A |
| CP000240 | 1261/1324 | 0.952 | *Synechococcus* OS-B' |

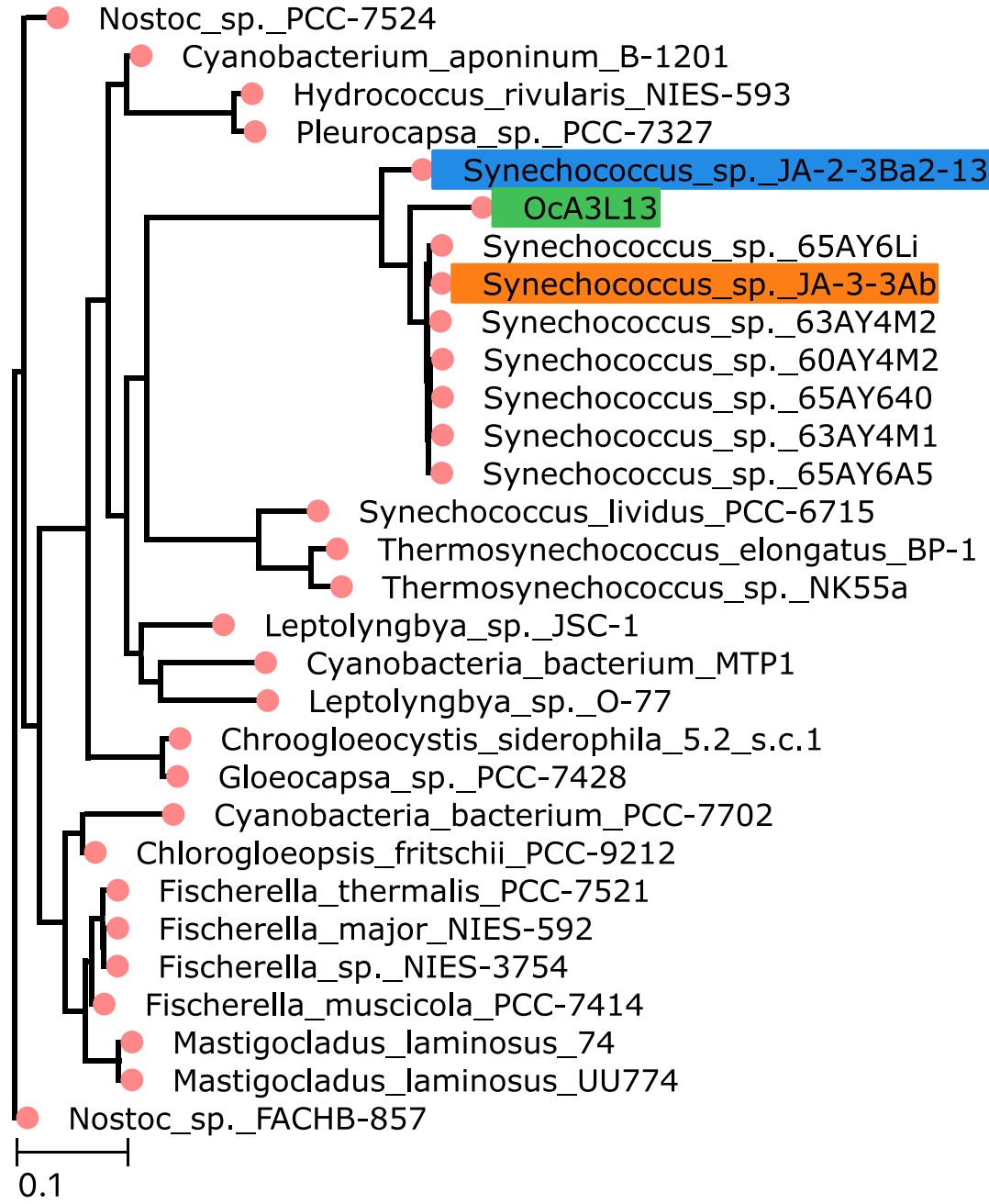

**Appendix 3—figure 1.** $\gamma$ SAG represents a new distinct species within Yellowstone *Synechococcus*. Shows the maximum likelihood tree obtained from an alignment of 16 marker genes present in the $\gamma$ SAG together with all other thermophilic cyanobacteria from *Chen et al., 2021*. Colored labels show the two Yellowstone *Synechococcus*, OS-A (orange) and OS-B' (blue), and the $\gamma$ SAG OA3L13 (green).

Inferred abundances of $\gamma$ from metagenomic samples are consistent with a single $\gamma$ SAG in our data (see Appendix 3). To further verify that no other γ SAGs were accidentally excluded during quality control (see Appendix A), we performed a BLAST search (alignment parameters: `-evalue 1E-10 -word_size 9 -qcov_hsp_perc 75.0`) of the $\gamma$ genes against the contigs from the SAGs that were filtered out. The results were consistent with the majority of the excluded SAGs containing mostly $\alpha$ and $\beta$ genes, with a smaller number of SAGs with divergences around twice as large (**Appendix 3—figure 2**). This confirms that no other $\gamma$ SAGs were present in our data.

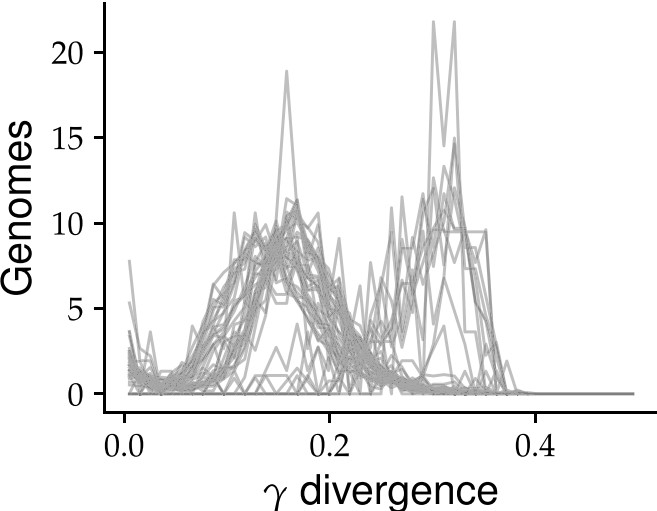

**Appendix 3—figure 2.** Single-amplified genomes (SAGs) excluded during quality control do not include additional γ genomes. The distribution of divergences from the best BLAST hit to the γ SAG (OcA3L13) across genes for each SAG excluded during the quality control process (see Appendix A) is shown. Each line represents the probability density of divergences across genes for one of the 44 SAGs.

## Estimating abundances of $\alpha$, $\beta$, and $\gamma$

To estimate abundances of the three main species $\alpha$, $\beta$, and $\gamma$ in the Octopus and Mushroom Springs, we recruited metagenomic reads from 34 samples taken from both springs to SAGs from each species. We chose the one γ SAG (OA3L13, ≈70% coverage) along with representative SAGs from $\alpha$ (MA02D15, ≈77% coverage) and $\beta$ (OcA2H14, ≈96% coverage) as reference genomes. Hybridization on length scales longer than the typical read size (150 bp in our case) can lead to errors in our estimates of species abundances. To minimize these errors, we only used core genes where three distinct species clusters could be identified and there was no evidence of hybridization in the single-cell data (see Appendices 1 and 5). This resulted in a set of ≈400 genes for each reference. We aligned all the metagenomic reads to these target gene sequences using BowTie2 (v2.3.4, default parameters) (*Langmead and Salzberg, 2012*) and extracted reads aligning to each of the three reference sequences using Samtools (v1.7.0) (*Li et al., 2009*) and bedtools bamtofastq command (v2.26.0) (*Quinlan and Hall, 2010*).

We assigned the recruited reads to each species cluster if their divergence from their respective reference was less than the 'main cloud' radius $d_M$. We determined $d_M$ self-consistently from the data as follows. For each reference, counted the number of reads that had divergences less than $d_M$ from that reference sequence and more than $d_M$ from the other two reference sequences, for $d_M$ from 1% to 15%. If the population consists of species clusters with typical diversity $\pi$, the number of reads within the cloud will increase until $d_M \approx \pi$. As $d_M$ approaches the typical divergence between species $d \approx 15\%$, more reads will start to map to multiple reference sequences, so the number of reads uniquely assigned to each species cluster will decrease. Therefore, the number of reads uniquely assigned to each species cluster should have a maximum as a function of $d_M$. We indeed found this to be the case for all three species (data not shown). Empirically, we found that $d_M = 5\%$ was close to the maximum for all three species, so we used this value throughout our analysis.

We found that the average depth of recruited reads across genes varied within the same sample. However, the variations across samples were highly correlated, which confirmed that most reads aligning to our chosen genes came from genomes that were part of the same species cluster rather than from genes transferred across species (*Appendix 3—figure 3*). We found that the genes that recruited the largest numbers of reads had read depths that were much more similar within the same sample compared to other genes. Interestingly, we found much more variation in depth between genes within the γ cluster compared to $\alpha$ and $\beta$ (*Appendix 3—figure 3*). One likely explanation for this is that γ contains much more diversity compared to either $\alpha$ or $\beta$.

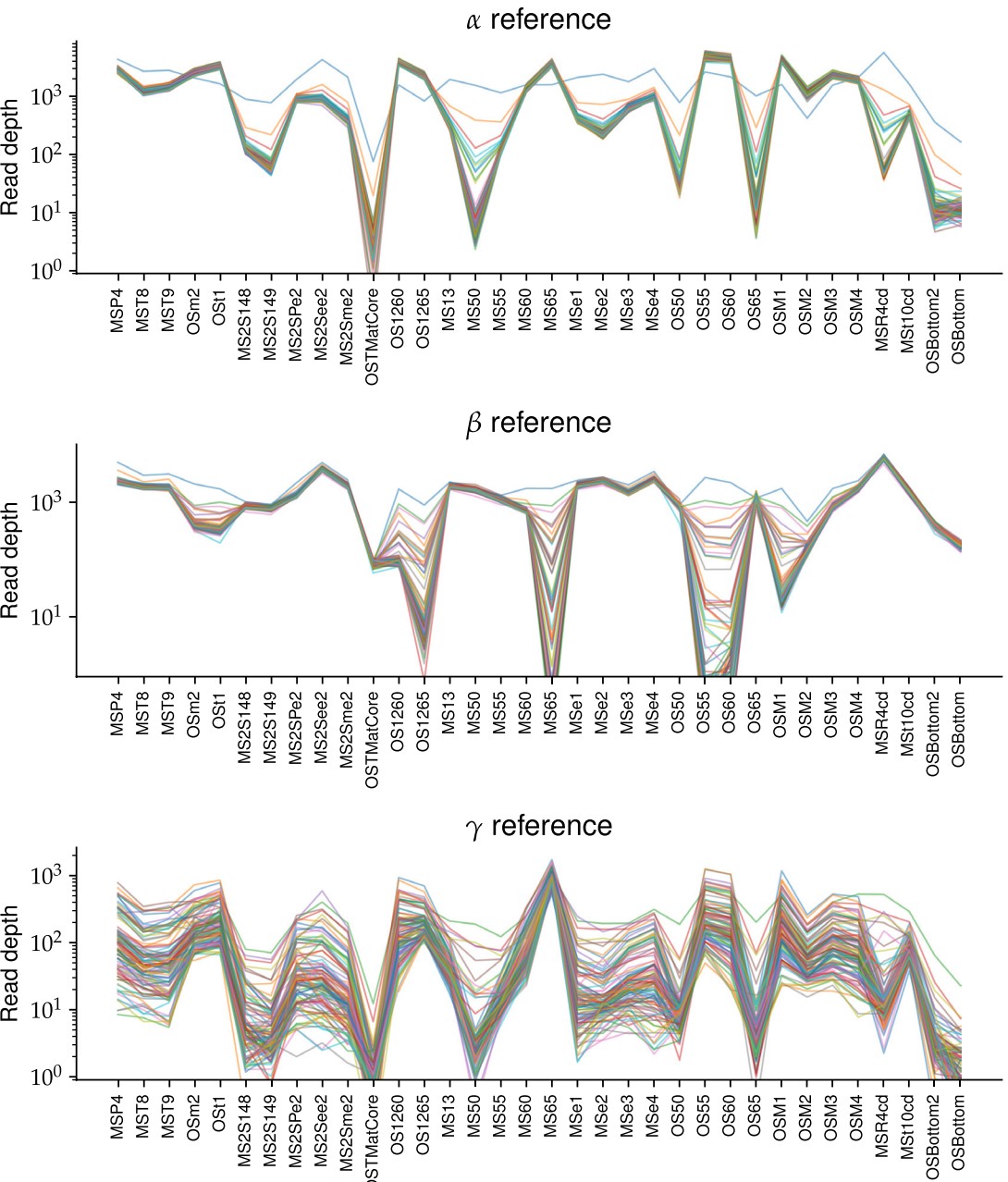

**Appendix 3—figure 3.** The number of reads recruited to different loci from the three main species is highly correlated across samples. The three panels show the average depth of reads recruited to representative single-amplified genomes (SAGs) from α (top), β (middle), and γ (bottom), respectively. Each panel shows 100 genes with the highest average read depth across samples, with each line representing one gene. Only reads that were <5% diverged from the reference genomes were included.

We estimated the abundance of each species cluster by averaging the depth of the 100 genes with the highest average depth across all samples for each cluster. We found that γ was more than an order of magnitude more abundant than our detection threshold given by the metagenomic read depth in 32 out of the 34 metagenome samples (*Appendix 3—figure 4*). Moreover, its abundance in the two samples for which we had single-cell and metagenome data (MSe4 and OSM3) was close to the detection threshold of the single-cell samples, which was around 2%. This explains why we only obtained a single γ cell from the single-cell samples after our quality controls. Finally, we found that the abundance of γ was highly correlated with α across all samples and was only present at

abundances higher than $O(1)\%$ in the Mushroom Spring sample taken from 65°C. This suggests that $\gamma$ is, together with $\alpha$, primarily adapted to growth at higher temperatures.

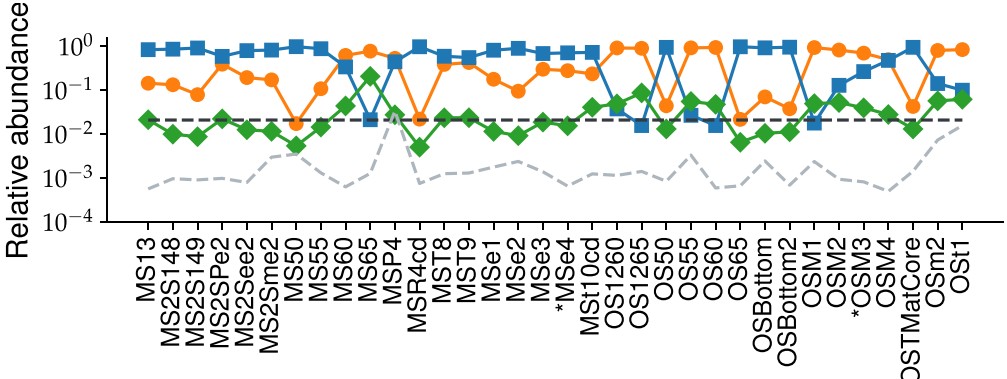

**Appendix 3—figure 4.** Abundance of $\gamma$ is significantly above detection threshold in 32 of 34 metagenomic samples and is highly correlated with $\alpha$. The relative abundances of $\alpha$ (orange circles), $\beta$ (blue squares), and $\gamma$ (green diamonds) across all 34 metageomic samples. The detection threshold for the single cell ($f_t = 1/48 \approx 0.021$, based on the maximum number of single-amplified genomes [SAGs] per sample) and metagenomic samples is shown in black and gray dashed lines, respectively. The asterisks on the x-axis label the two samples (MSe4 and OSM3) with both single-cell and metagenomic data.

## Other *Synechococcus* species

Are other *Synechococcus* species present in Mushroom or Octopus Spring apart from $\alpha$, $\beta$, and $\gamma$? To answer this question, we again used the 34 samples from which we have high-depth metagenomes. We used BLAST with highly sensitive parameter choices to minimize the number of true *Synechococcus* reads not captured. For computational efficiency, we focused on the 100 genes with highest coverage depth in the metagenomes across all three references.

To determine how much of the population is in the $\alpha$–$\beta$–$\gamma$ main cloud, we used the same three SAGs as in the previous subsection as references and recruited reads using BLAST (`-task blastn -evalue 1e-3 -word_size 6 -num_threads 32`). Using these parameters, we could recruit reads belonging to the dominant *Roseiflexus* species in our samples, which is in a different phylum from *Synechococcus*. We were therefore confident that any reads from other potential *Synechococcus* species would be included in our analysis. To limit the analysis to just *Synechococcus* species, we removed all reads that were more than 20% diverged from all of our references from subsequent analysis.

Because of the large amount of data in this analysis, we focus on just the Mushroom Spring samples taken from different temperatures (MS temperature series). We recruited a total of 4,522,833 reads with an average depth per gene of 4980. We calculated the minimum divergence from the three reference sequences for each read and found 94% within 5% divergence of one of them (*Appendix 3—figure 5A*). For comparison, we determined the typical divergences between species by dividing the reference sequences into 150 bp segments (the same size as our reads) and calculating the divergences between all three pairs of species (*Appendix 3—figure 5B*). The middle 95-percentile of the combined distributions extended down to a divergence of around 5% (eight SNPs), consistent with our cutoff for the intraspecies divergences at these loci. These results are consistent with all of the *Synechococcus* reads being from one of the three species and set an upper bound on the relative abundance of any other species at below 4%.

To check how sensitive our detection method is to the presence of other species, we used the Mushroom Spring sample from 65 in which $\gamma$ had a relatively high abundance. To mimic the presence of another species, we compared the previous distributions to the distribution of the minimum divergence from $\alpha$ and $\beta$ only (dashed red line in *Appendix 3—figure 5A*). We found a 19% decrease in cumulative number of reads within 5% from the references in this case, which was very close to the 21% abundance of $\gamma$ we estimated in the previous subsection based on the average depth across

loci. This shows that our method is sensitive to the presence of other species and illustrates the benefits of combining single-cell and metagenome analysis for identifying unknown species.

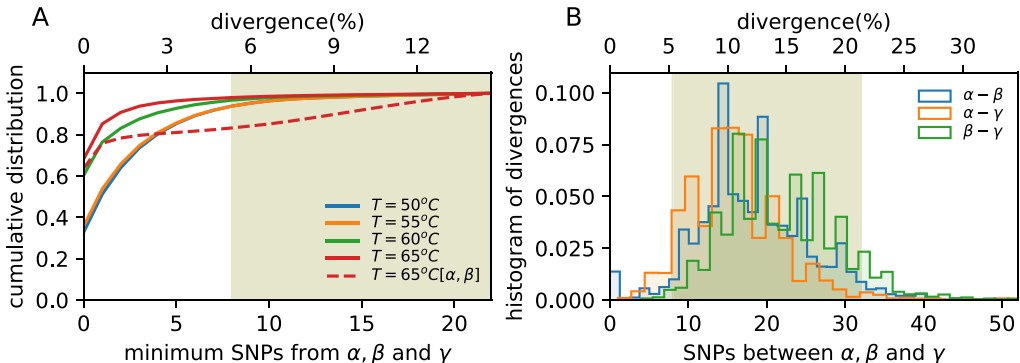

**Appendix 3—figure 5.** A vast majority of *Synechococcus* core gene sequences are within the α–β–γ main clouds. (**A**) Cumulative distribution of the minimum divergence of each read from the reference sequences across all loci used in the left panel. Different colors represent samples taken from different temperatures from Mushroom Spring. Solid lines show minimum divergence from all three references, and dashed lines from α and *β* only. Shaded olive region shows the typical divergences between species shown in the right panel. (**B**) Histogram of pairwise divergences between three representative single-amplified genomes (SAGs) for α, *β*, and γ at the 100 core loci chosen for illustration. Distributions are across all 150 bp segments from each locus. Shaded olive region is the middle 95% percentile for the combined distribution.

To further verify that no other *Synechococcus* species were present in our samples, we closely examined the diversity of 16S sequences. We used only single reads in our analysis instead of trying to assemble full sequences to prevent misinference caused by hybridization. Because the diversity along the 16S sequence is highly heterogeneous, variation between read recruitment locations can dominate the variation between species. We controlled for this by limiting the analysis to reads that aligned within a given region of the 16S. For 150 bp reads, we found that using segment sizes of 200 bp provided the best compromise between increasing depth and minimizing variance due to alignment location. To find the most informative region, we calculated the nucleotide diversity at each site $\mathcal{H}_i = 2\langle f_i(1-f)\rangle$, where $f_i$ is the minor allele frequency at site $i$, and averaged it over all 200 bp segments along the 16S (*Appendix 3—figure 6A*). We found the diversity varied by more than a factor of four, consistent with the presence of conserved and variable regions. The diversity was nearly identical when using all three species as reference genomes, suggesting our recruitment captured all sequences within the main cloud. The highest average diversity was close to 400–600 bp segments, which we chose for further analysis.

We calculated the distribution of divergences from the $\gamma$ allele for all reads recruited from this segment for MS temperature series. We found temperatures of 60°C and below had two distinct peaks close to *Appendix 3—figure 6B* 4% and 8% divergences. For comparison, we calculated the range of divergences from $\alpha$ and $\beta$ for all 150 bp subsegments within the 400–600 bp segment and found these matched with the two peaks we observed. At 65°C, we also found only two peaks, one close to zero and another close to 4%, consistent with our previous estimates showing that this sample mainly consisted of $\alpha$ and $\gamma$ cells.

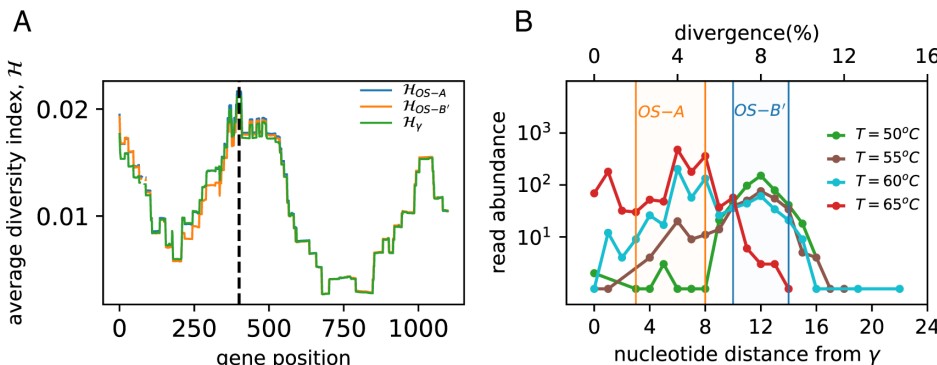

**Appendix 3—figure 6.** Diversity of *Synechococcus* 16S rRNA sequences from Mushroom Spring temperature series samples is entirely contained within the α–β–γ main clouds. (**A**) Average diversity $\mathcal{H}$ (see text) of metagenome reads from all 34 samples mapping to *Synechococcus* along the length of the 16S sequence. Averaging was done over all sites within a 200-bp window starting at the location given by the x-axis. Results for reads recruited using each of the three main species as a reference sequence are shown in different colors. A black dashed line indicates the 400–600 bp segment with the highest diversity which was chosen for subsequent analysis. (**B**) Distribution of nucleotide divergences from the γ allele within the 400–600 bp segment of the 16S for four different samples taken from Mushroom Spring. Shaded regions show the range of divergences of OS-A (orange) and OS-B' (blue) from γ across all 51 windows of 150 bp length, corresponding to the range of expected divergences of reads from α and β.

Do peaks correspond to single-species clusters or are they amalgamations of several distinct species with similar divergences from each other? To answer this question, we constructed a multiple-sequence alignment of all of the 16S reads recruited across the 34 metagenome samples and performed a hierarchical clustering on the pairwise divergence matrix using average linkage. Pairwise divergences were calculated by padding the reads with gaps to ensure all sequences were 200 bp in length and calculating the nucleotide divergence ignoring any gaps. For the MS temperature series, we found 97% of reads were in one of the three clusters associated with α, β, or γ (*Appendix 3—figure 7A–D*). Consistent with our previous results, samples below 65°C had two main clusters close to OS-A and OS-B' (*Appendix 3—figure 7A–C*), while at 65°C they were close to γ and OS-A (*Appendix 3—figure 7D*). Aggregating across all 34 metagenome samples, we found that α and β were the two dominant clusters, with γ being the next largest (*Appendix 3—figure 7E, F*). We also observed a large difference in the overall abundances of α and β across the two springs, but the very large heterogeneity across samples we found previously (*Appendix 3—figure 4*) suggests this is likely due to the particular samples used in this study and may not necessarily be an accurate reflection of the overall abundances in the two springs.

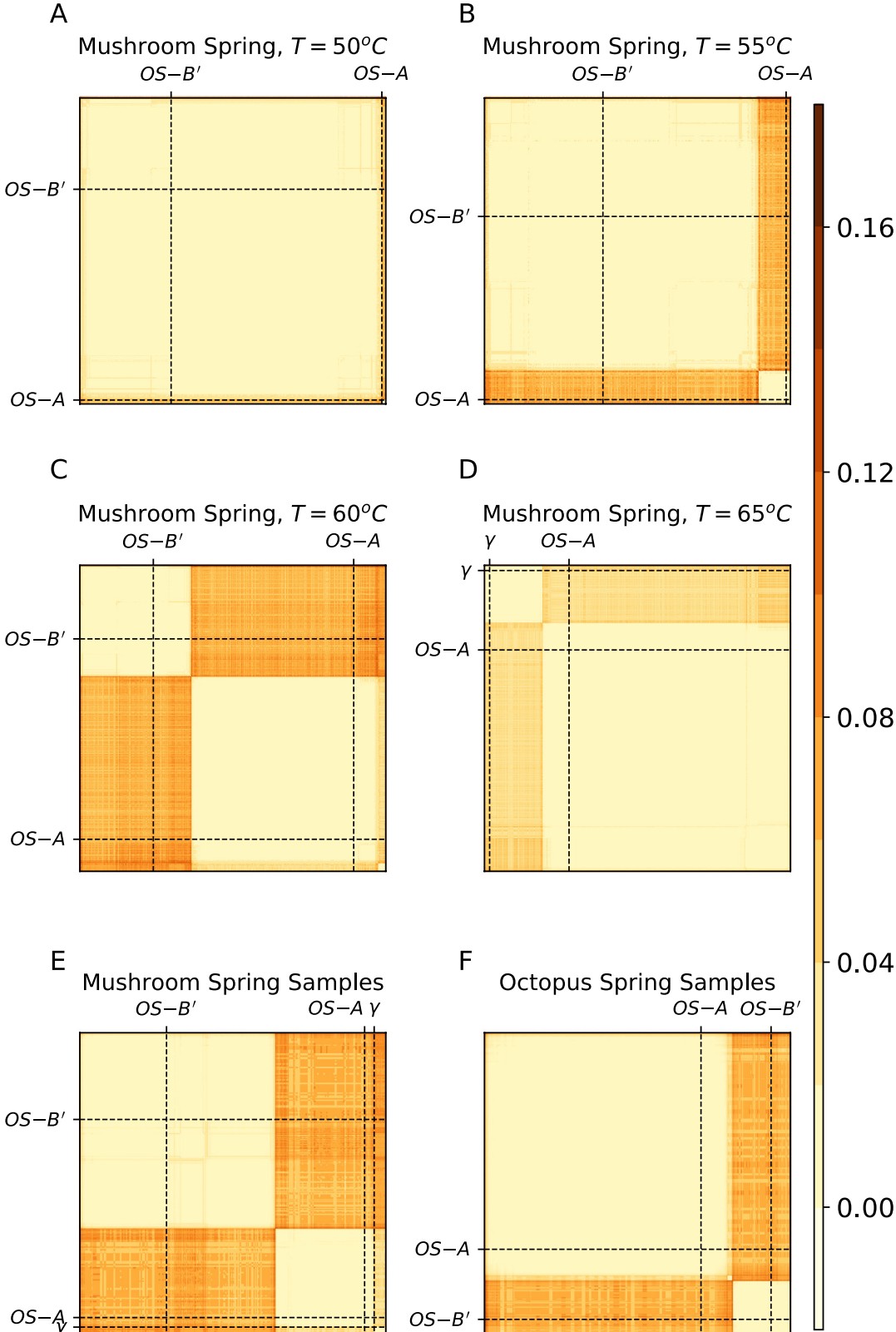

**Appendix 3—figure 7.** All *Synechococcus* 16S rRNA sequences from the metagenome samples belong to one of the three known species clusters. (**A–D**) Each subpanel shows the matrix of pairwise divergences between all reads recruited to the 400–600 bp segment of the γ 16S rRNA sequence for the four samples from the MS temperature *Appendix 3—figure 7 continued on next page*

*Appendix 3—figure 7 continued*
series. Dashed lines highlight the OS-A, OS-B', and $\gamma$ reference sequences in this segment for comparison. (**E, F**) Same as above using a subsample of reads recruited from all 34 metagenome samples divided by spring.

## Appendix 4

### Linkage analysis

In this section, we summarize our analysis of linkage disequilibrium within $\alpha$ and $\beta$. The structure of this section is as follows. We begin by reviewing definitions of two commonly used ways to quantify linkage disequilibrium. We then compare the rate of decay of linkage disequilibrium within and between the two species. We next compare the decrease in linkage in the data to theoretical predictions from the neutral model and show that while the decay is similar to the neutral model prediction, the overall shape of the linkage decay curves cannot be fully explained by the classical neutral model. We then show that within $\alpha$, there is substantial heterogeneity in the rate of linkage decay in low-diversity regions compared to high-diversity ones. Surprisingly, we find that linkage decreases more rapidly in low-divergence regions, suggesting the absence of a clonal frame for $\alpha$ even on short time scales. The final subsection presents a technical validation of our method for processing alignments showing that our results are robust to filtering for outlier gene alleles.

### Statistical measures of linkage disequilibrium

We used two distinct but related methods for quantifying linkage disequilibrium, $\sigma_d^2$ and $r^2$. Both metrics are related to the correlation coefficient between alleles at two loci but differ in their normalization. For a single pair of loci $A$ and $B$ with alleles A/a and B/b, we can define the correlation coefficient between their respective frequencies as

$$r_{AB}^2 = \frac{(f_{AB} - f_A f_B)^2}{f_A(1 - f_A)f_B(1 - f_B)}. \tag{2}$$

Using this expression, we can define a statistical measure of linkage as a function of distance along the genome by averaging over pairs of loci with a given separation $x = |x_A - x_B|$, where $x_A$ and $x_B$ are the positions of the two loci along the genome. The two most common ways to perform the average are

$$\sigma_d^2(x) = \frac{\left\langle (f_{AB} - f_A f_B)^2 \right\rangle_{|x_A - x_B| = x}}{\left\langle f_A(1 - f_A)f_B(1 - f_B) \right\rangle_{|x_A - x_B| = x}}, \tag{3}$$

and

$$r^2(x) = \left\langle \frac{(f_{AB} - f_A f_B)^2}{f_A(1 - f_A)f_B(1 - f_B)} \right\rangle_{|x_A - x_B| = x}. \tag{4}$$

For a neutral panmitic and well-mixed population, both metrics decrease as $\sigma_d^2(x) \sim r^2(x) \sim 1/x$. For $\sigma_d^2$, there also exists an analytic expression (**Ohta and Kimura, 1969**):

$$\sigma_d^2 = \frac{10 + 2NR}{22 + 26NR + 4(NR)^2}, \tag{5}$$

where $N$ is the population size, and $R$ the recombination rate per base pair per generation. When recombination is rare and occurs via crossover $R(x) \approx R_0 x$, and the expression is consistent with the scaling $\sigma_d^2(x) \sim 1/x$ mentioned previously. While numerical and theoretical studies have shown that $r^2$ is more sensitive to low-frequency alleles compared to $\sigma_d^2$ (**Hudson, 1985**; **Song and Song, 2007**), our analysis did not reveal significant differences between two metrics. We therefore focus here on $\sigma_d^2$ which is more commonly used and easier to compare to theoretical predictions.

We calculated $\sigma_d^2(x)$ and $r^2(x)$ for $\alpha$ and $\beta$ separately and together. Both procedures are similar, so we focus on the analysis for all $\alpha$ and $\beta$ sequences. Unless otherwise stated, all analyses were performed on alignments of individual genes and the results were averaged across genes. Using this method, we can easily obtain linkage information over distances of up to around 1 kbp. Over longer distances, the results can be dominated by specific recombination events in a small number of unusually long genes and are therefore less useful for quantifying genome-wide statistical patterns.

We thus perform most of our analysis up to distances of around 1 kbp, except when calculating typical genome-wide linkage as discussed below.

Alignments for each orthogroup were first trimmed to remove segments with excess numbers of gaps. We then determined a consensus sequence by taking the most common codon sequence at each location along the reading frame, excluding gaps. Each nucleotide was labeled 0 if it was a gap, 1 if it was the same as the consensus, and 2 if it was different from the consensus. While this does not distinguish between biallelic and multi-allelic sites, the latter are much less common than the former and should not affect our conclusions. Finally, we calculated the two linkage coefficients using *Equations 3 and 4*. To estimate linkage across distances longer than a single gene, we performed the average across SNPs spanning a pair of genes and then averaged the values across a random sample of 1000 pairs of genes.

The linkage calculations within each species were performed in an analogous way, with the exception of a two-step filtering process of the alignments at the beginning. We describe this procedure in detail at the end of this section.

## Linkage decrease within vs between species

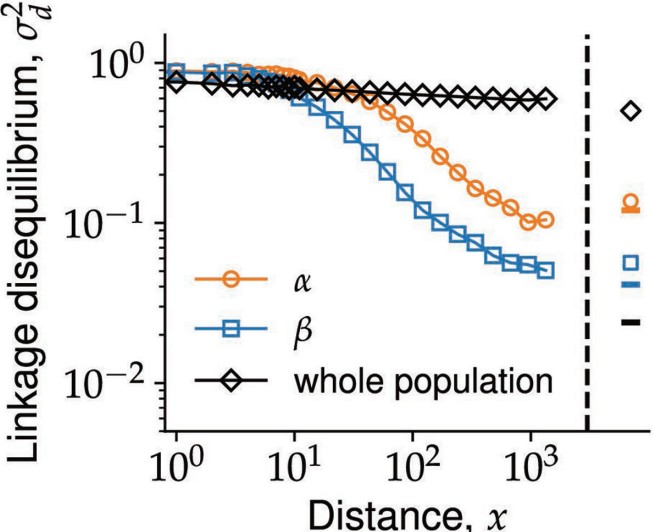

**Appendix 4—figure 1.** Genetic linkage analysis shows extensive recombination within α and β. Linkage disequilibrium $\sigma_d^2$ as a function of separation between SNPs $x$ averaged across all core genes among the α (orange circles), β (blue squares), or all of the cells (black diamonds). Genome-wide estimates from SNPs in different genes are shown to the right of the black dashed line using the same symbols. Fully unlinked controls for the genome-wide linkage are shown as horizontal lines of different colors. The main cloud cutoff for both α and β was set to $c = 0.05$.

Our analysis showed that within both $\alpha$ and $\beta$, linkage quickly decreased by approximately an order of magnitude over a distance of ~1 kbp (orange circles and blue squares in *Appendix 4—figure 1*). However, when considering the population as a whole, we find linkage disequilibrium is maintained across the whole length of the genome (black diamonds in *Appendix 4—figure 1*). Within $\beta$, the linkage at distances of 1 kbp was identical to the genomewide linkage between random pairs of genes. Within $\alpha$, the decrease was slower, but followed a similar pattern. The value of the genome-wide linkage was slightly higher than at 1 kbp, likely because of the reduced sample size at long genomic distances caused by poor genome coverage. Consistent with this, we found that the genomewide-linkage values in both species were close to completely unlinked controls obtained by randomly permuting each column of the alignments ($\sigma_d^2 = 0.14$ vs $\sigma_d^2 = 0.12$ in $\alpha$ and $\sigma_d^2 = 0.06$ vs $\sigma_d^2 = 0.04$ in $\beta$). Overall, these results suggest that the dynamics leading to the reassortment of alleles are similar in the two species and only differ in the length scale over which alleles become unlinked. We discuss this in more detail in the next section.

The fact that both species have similar levels of genomewide linkage despite the sample size of $\alpha$ being less than half of that of $\beta$ shows that our estimates for the genomewide linkage are not

significantly affected by the sample size effect. We further validated this by taking random samples from the $\beta$ main cloud equal to the size of the $\alpha$ main cloud at each locus and recalculating the linkage curves. We found the results for the full $\beta$ main cloud and the subsampled main cloud were almost identical (*Appendix 4—figure 2*). This shows that the sample size does not have a large effect on our results and cannot explain the differences in linkage decay between the two species.

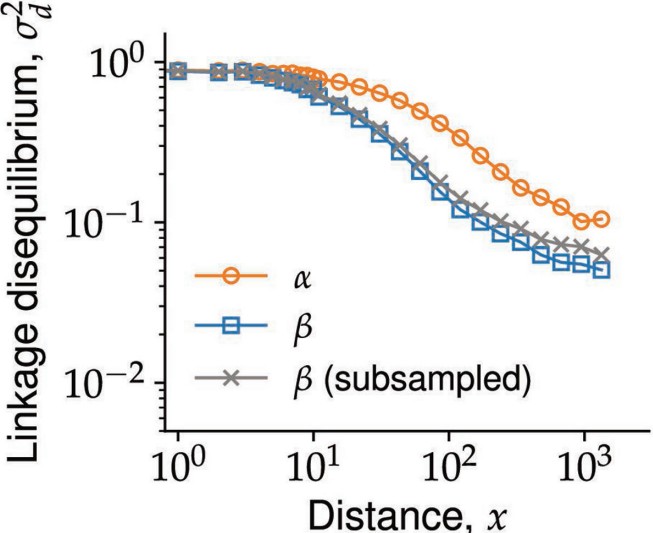

**Appendix 4—figure 2.** Sample size has a small effect on linkage estimates. Linkage disequilibria curves for the whole $\alpha$ and $\beta$ populations shown in *Appendix 4—figure 1* are compared to a random subsample of $\beta$ sequences chosen to match the $\alpha$ sample size at each locus (shown in gray).

The rapid decrease in linkage sets important limits on how much information about the diversity in the population is contained in samples of marker genes. To illustrate this, we performed a linear regression between the whole-genome divergences and the 16S rRNA divergence. Consistent with our previous analysis, we found slightly stronger linkage in $\alpha$ compared to $\beta$, but in both cases, the correlations were weak and the 16S divergences explained less than 10% of the genome-wide divergences (*Appendix 4—figure 1B*). These results show that frequent recombination in the population makes marker genes, such as the 16S rRNA, poor predictors of the genome-wide diversity.

## Comparison to model of neutral drift with recombination

To get a more quantitative understanding of the effects of recombination on the genetic diversity in our population, we compared our previous results to theoretical predictions from population genetics.

At short distances, the prediction from *Equation 5* is that $\sigma_d^2 = 5/11$. However, in our data, the value of $\sigma_d^2$ for nearby sites was very close to one. While the Yellowstone *Synechococcus* have a strong mutational bias toward GC nucleotides (*Rosen et al., 2018*), *Equation 5* is independent of the relative rates of mutations. As a result, we conclude that the neutral model cannot explain the linkage curves we observe.

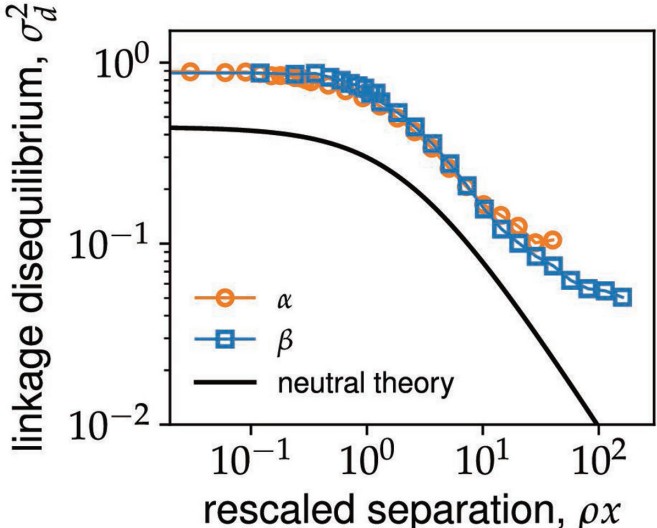

**Appendix 4—figure 3.** Linkage decay curves for α and β collapse onto the same curve after rescaling distances by average recombination rate $\rho$. Shows the linkage decay as a function of distances between sites rescaled by the recombination rates $\rho$ fitted from **Figure 2** (main text). The solid black line shows the theoretical prediction from the neutral model given by **Equation 5**.

While the overall level of $\sigma_d^2$ in our data is inconsistent with the neutral prediction, we can still ask whether the rate of decay with distance has a similar functional form. To do this, we followed **Garud et al., 2019** and fit the linkage curves to the following functional form:

$$\sigma_d^2(x) = A \frac{10 + \rho x}{22 + 13\rho x + (\rho x)^2}, \tag{6}$$

where $\rho = 2NR$ is the population recombination rate. We performed the fit by choosing $A$ such that the above expression matched the value of $\sigma_d^2$ in the data for neighboring sites ($x = 1$). We then fitted the curves by eye by adjusting the value of $\rho$.

We found that this functional form could fit the data very well for both species up to distances of $x \approx 300$ bp (**Figure 2** in main text). For longer distances, the linkage decrease was slightly slower than predicted by **Equation 6**. One possible interpretation of this slowdown is that longer distances have a significant overlap with the distribution of lengths of recombined segments $L_r$, beyond which we expect linkage to reach the genomewide level (**Garud et al., 2019**). This suggests that recombined segments as short as 300 bp might play an important role in the recombination process within the population.

We further validated that the linkage curves have the same functional form by rescaling distances between sites by the recombination rate $\rho$ inferred previously. We found a very good collapse of the linkage curves for the two species when plotted against the rescaled distances (**Appendix 4—figure 3B**). This strongly implies that the dynamics leading to the decay of linkage are the same in both species, with the differences between them being length scale over which sites are effectively linked.

## Linkage on intermediate length scales

Our analysis so far was limited to either distances up to the length of typical genes (~1 kbp) or across the entire genome. Extending our analysis over distances in between these two extremes is difficult due to several technical challenges. The first is the fact that typical coverages for our SAGs are around 30%. Together with a typical contig size of ~10 kbp, this means that our sample size quickly decreases with distance. The second challenge is that at longer distances, gene order might vary across genomes, making it difficult to define a genomic distance at the population level.

## Linkage between multi-SNP alleles

We calculated the linkage disequilibrium between multi-SNP features, such as SNP blocks or hybrid alleles, using **Equation 2** as in the case of biallelic SNPs. In the case of SNP blocks shown in **Figure 7D**

in the main text, we calculate the frequencies for the four haplotypes within SAGs that are covered at both loci. The unlinked controls (represented by solid lines in *Figure 7D*) were obtained by a random permutation of the alleles at each locus independently, while keeping the SAGs that are covered at each locus fixed. This procedure preserves any correlations in genome coverage between nearby loci and provides the most accurate estimate of $r^2_{AB}$ for a completely unlinked genome.

The linkage between hybrid gene alleles was calculated similarly as for the SNP blocks, except for the genotype assignment. At each locus, we labeled the most abundant allele using upper case letters ($A, B, \ldots$) and grouped all of the remaining ones into a single allele labeled by a lower case letter ($a, b, \ldots$). The linkage matrix $r^2_{AB}$ and the unlinked control were calculated the same as previously described. The results for $\alpha$ were $\langle r^2_{AB}\rangle = 0.04 \pm 0.15$ compared to $\langle r^2_{AB}\rangle = 0.02 \pm 0.11$ for the unlinked control. For $\beta$, the results were $\langle r^2_{AB}\rangle = 0.01 \pm 0.09$ compared to $\langle r^2_{AB}\rangle = 0.009 \pm 0.08$ for the unlinked control. Note that the average linkage disequilibrium was close to the unlinked controls in both species, with $\alpha$ having slightly higher linkage. These agree with those from the analysis of SNP blocks shown in the main text.

## Determining the main cloud cutoff for $\alpha$ and $\beta$

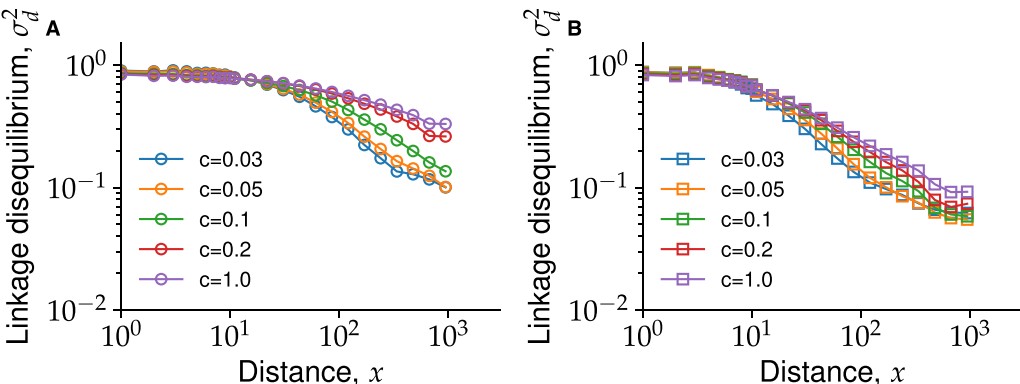

**Appendix 4—figure 4.** The effect of the main cloud diameter $c$ on the rate of linkage decay. The linkage decay within α (A) and β (B) is plotted for different main cloud cutoffs $c$ (see main text). Note the convergence of the curves for $c \leq 0.05$.

For each species, we first selected all of the sequences that belonged to SAGs assigned to that species (see Appendix 2). We then calculated the consensus sequence for this group of sequences, as described above, and removed all sequences with divergences greater than a main cloud cutoff $c$. This is a simple way to remove sequences that are the result of gene-level hybridization (see Appendix 5 and *Rosen et al., 2015* for a similar procedure). We chose the main cloud cutoff for each species by calculating the linkage curves for different values and choosing the largest value for which the curves did not noticeably change (*Appendix 4—figure 4*). Based on this criterion, we use $c = 0.05$ for $\alpha$ and $c = 0.1$ for $\beta$ throughout.

## Appendix 5

### Gene-level hybridization

This section describes the method we used to quantify hybridization at the gene level. Our method uses a set of simple heuristics to assign labels to the species clusters defined in Appendix A based on the composition of species to which the sequences belonged. We then identified hybridization events by searching for discrepancies between the gene and whole-genome labels as explained below.

### Simple hybrid genes

We first assigned labels to the species-level clusters constructed as described in Appendix 1. Based on the gene triplet results presented in Appendix 2, we expect most sequences to be contained in two main clusters, close to OS-A and OS-B', respectively, with at least one sequence forming a separate cluster whenever the orthogroup was covered in the $\gamma$ SAG. Based on this expectation, we considered all of the clusters from each orthogroup in decreasing order of their sizes and labeled them as described in the main text.

We next determined which orthogroups are part of the core genome. We defined a gene as a core gene if it is likely present in all of the genomes in our sample. The average coverage for each SAG in our sample was approximately 30%, so it is not possible to directly determine whether some genes are always present. Instead, we use a statistical measure of the coverage to assign orthogroups to the core genome. Specifically, we used the rank-coverage plot across all orthogroups and SAGs (equivalent to the complementary cumulative distribution) to determine an appropriate cutoff for the core genome. The rank-coverage plot showed three distinct regions. First, there were around a dozen orthogroups that had unusually high abundances, with the number of sequences in some cases being greater than the number of SAGs in our sample. These are likely multiple-copy genes and IS elements and were excluded from further analysis. Then there was a large plateau comprising over 1000 orthogroups with coverages in the range $100-150$, followed by a sharp decrease (on a logarithmic scale) in the coverage. We found that setting a coverage threshold for the presence at 20% of the number of SAGs from each species approximately matched the shoulder between the plateau and the sharp decrease, and so we used this as a criterion for assigning core genes.

The final step in our algorithm is to catalog hybridization events, which we do by simply searching for gene sequences whose labels were different from the whole-genome species assignment. Analysis of these results showed a large number of transfers, mainly from $\beta$ into $\alpha$, which came from contigs that did not contain any genes labeled as the host species. One explanation for the presence of such contigs is that they represent larger genomic segments flanked by repetitive regions (such as IS elements) that are transferred and integrated into the host genome. Long repetitive genomic segments can lead to contig breaks during assembly, which would explain the complete absence of genes belonging to the host cluster. Alternatively, such contigs could be the result of contamination with external DNA. However, the contamination hypothesis does not explain why there seem to be more transfers from $\beta$ into than vice versa. We also did not find any evidence of differences in the number of such transfers between the sample containing only $\alpha$ SAGs (OS2005, see Appendix 10) and samples containing mixtures of and $\beta$. While this evidence suggests these transfers are very likely the result of hybridization, we cannot rule out the possibility of contamination. We therefore excluded all hybridization events on contigs that did not contain any genes labeled as the host genome from the analysis. The results of our analysis are summarized in *Figure 5A, B* in the main text.

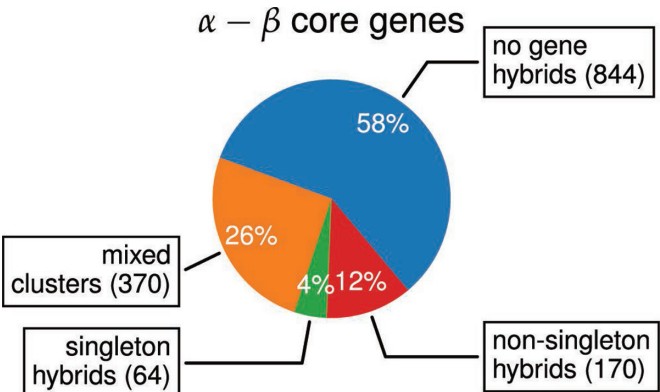

**Appendix 5—figure 1.** The majority of orthogroups contain hybrid sequences. Orthogroups were classified into four mutually exclusive groups depending on the amount of hybridization observed. The majority of mixed-species orthogroups are mosaic hybrids (see main text) and are shown in orange. Orthogroups in which the local sequence clusters matched the whole genome ones were labeled 'no hybrids' and shown in blue, those in which a single cell contained a sequence from a different cluster were labeled 'singleton hybrids' and shown in green, and those in which multiple cells contained sequences from different clusters were labeled 'non-singleton hybrids' and shown in red.

## Appendix 6

### Linkage block detection and analysis

In this section, we describe the method we used to discover SNP blocks and the statistical analyses we performed.

### Processing alignments for SNP block detection

In order to identify linkage blocks originating from hybridization, the raw alignments from the pangenome construction were processed as follows. We divided core OGs into distinct species clusters, as explained in Appendices 1 and 5, and removed any sequences from SAGs assigned to a different species than the cluster. This was done to avoid false positive blocks due to the presence of recently transferred whole-gene alleles from different species. For OGs with mixed-species mosaic clusters, all sequences in the cluster were initially included. Alignments were then trimmed and codons that contained a high frequency of gaps were removed. Finally, we defined a main cloud of sequences for $\alpha$ and $\beta$ comprising sequences within a divergence diameter $d_M = 0.1$ from the consensus for each species and removed all sequences outside it.

### SNP block detection method

We used a linkage-based approach to search for evidence of hybridization within genes. Visual inspection of the alignments in the highly diverse $\alpha$ genes and in most $\beta$ genes revealed segments with high SNP density grouped into two haplotypes in strong linkage disequilibrium. We hypothesized that these SNP blocks were the result of intragenic hybridization, and we developed an algorithm to systematically identify them.

The algorithm can be summarized by the following steps:

1. Read orthogroup alignment
2. Pre-process alignment
    a. Trim alignment edges and remove columns with high fraction of gap codons
    b. For each species, infer consensus sequence and remove sequences outside a predetermined diameter around the consensus
    c. Replace all SNPs with absolute frequencies $n < n_c = 4$ with the consensus for their respective columns
3. For each species $s$, subsample sequences from cells assigned to $s$
4. Calculate the linkage matrix $r_{ij}^2$ for all pairs of SNPs
5. Identify groups of consecutive SNPs with $r_{ij}^2 = 1$, up to a rounding error of $\epsilon = 10^{-5}$, containing at least $l_{min} = 5$ SNPs

The inclusion of step 2c accounts for the possibility of more recent low-frequency SNPs emerging after the initial hybridization took place. Note that typical alignments depths are ~30 for $\alpha$ and ~70 for $\beta$ and are considerably larger than $n_c$.

We applied the above algorithm to the alignments of all of the core genes and obtained a total of 1000 blocks within $\alpha$ and 1483 blocks within $\beta$. Most blocks were short, but those in $\alpha$ were on average more than twice as long as those in $\beta$ ($\langle L \rangle = 68$ in $\alpha$ vs $\langle L \rangle = 29$ in $\beta$; **Appendix 6—figure 1A**). Similarly, $\alpha$ blocks contained more SNPs on average compared to $\beta$, although here the difference was less pronounced ($\langle l \rangle = 11.3$ in $\alpha$ vs $\langle l \rangle = 7.9$ in $\beta$; **Appendix 6—figure 1B**).

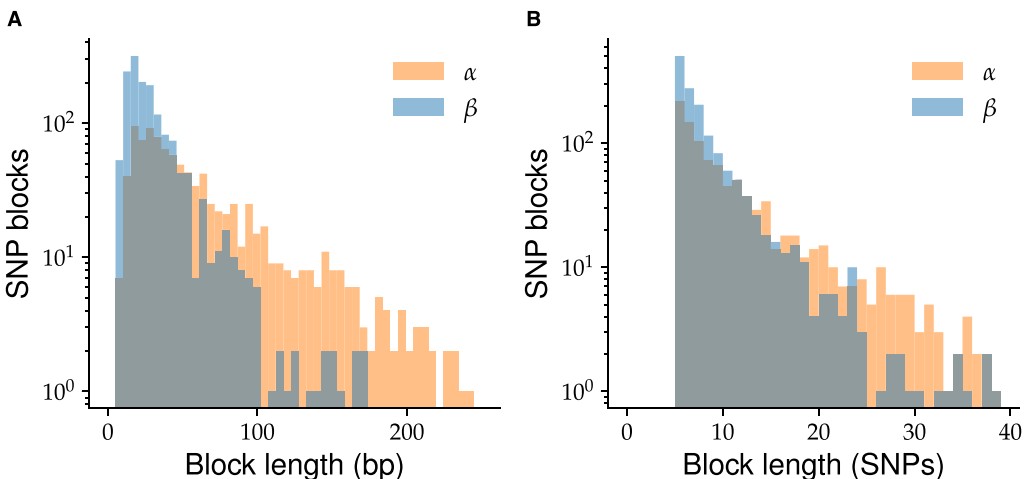

**Appendix 6—figure 1.** Typical blocks are short and contain a handful of SNPs. (**A**) The distribution of lengths of SNP blocks within $\alpha$ (orange) and $\beta$ (blue) genomes, detected using the algorithm described in this section. (**B**) The distribution of the number of SNPs within the same blocks as in (**A**).

To identify hybrid haplotypes, we first determined the consensus sequences for the two block haplotypes, together with the other two species within the block segment. We then labeled any haplotypes whose sequence was identical to one of the other two species as a hybrid segment with the corresponding species as the donor. These results are shown in *Appendix 6—table 1*.

**Appendix 6—table 1.** Number of SNP blocks within $\alpha$ and $\beta$ containing haplotypes that mapped to different donor species in core genome.

| Host species | Block donor species | | | | Total blocks | Genes with blocks | Total gene transfers | Majority mosaic genes | Core genes |
|---|---|---|---|---|---|---|---|---|---|
| | $\alpha$ | $\beta$ | $\gamma$ | $X$ | | | | | |
| $\alpha$ | - | 291 | 347 | 244 | 1000 | 365 | 194 | 370 | 1448 |
| $\beta$ | 310 | - | 125 | 941 | 1483 | 669 | 70 | 370 | 1448 |

To control for other processes that could potentially produce similar patterns of linkage disequilibrium, we compared the distributions of divergences between haplotypes and block lengths for cases where a hybrid haplotype could be identified directly to cases we could not. No obvious difference in the distributions could be seen between the two cases, consistent with our interpretation that the blocks without matches to other species in our sample are also the result of hybridization.

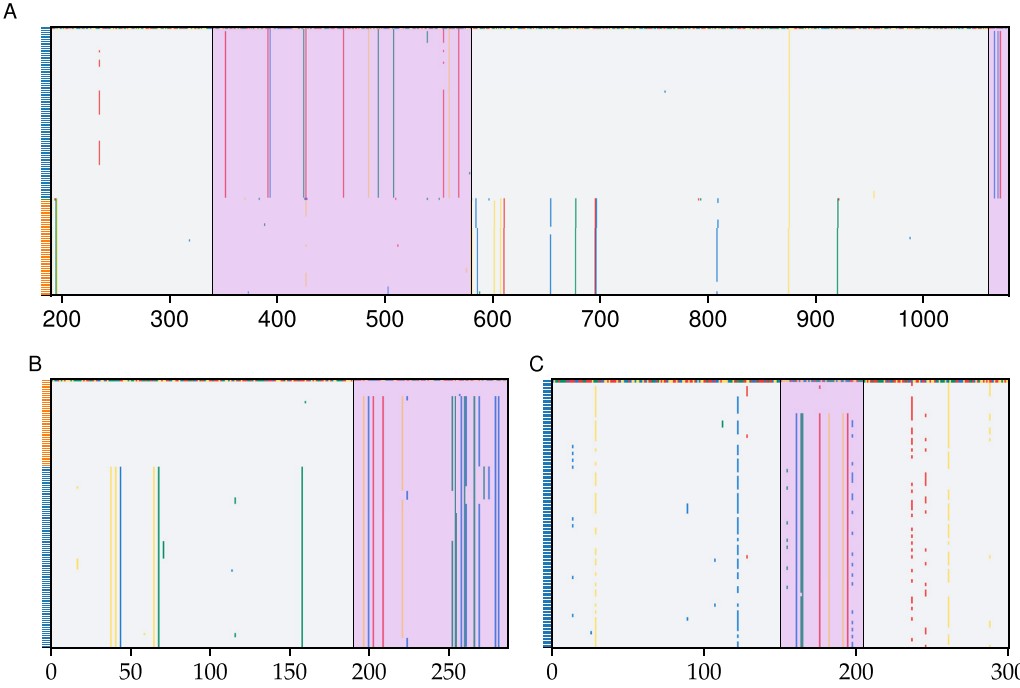

**Appendix 6—figure 2.** Examples of blocks and hybrid sequences within ribosomal genes. Alignments are represented as described previously with blocks highlighted in purple. (**A**) Alignment of 16S sequences with a $\beta$ cell containing a hybrid segment starting at around position 340 and ending around position 580 in the alignment. Note the shorter block shared by the first two $\beta$ sequences at the end of the alignment (around position 1070). (**B**) Alignment of the SSU ribosomal protein rpsN with α cells ordered at the top and $\beta$ cells at the bottom of the alignment. The first $\alpha$ cell was chosen as the reference sequence and is shown in color. Note the large fraction of $\alpha$ cells around 200–300 bp that are similar to the $\beta$ sequences and distinct from the α reference. (**C**) Segment of alignment of $\beta$ sequences at the LSU ribosomal protein rplM. The reference sequence was chosen to illustrate short SNP block in the center of the alignment.

## Haplotype divergences

We further verified that the divergences between block haplotypes were consistent with what we would expect from hybridization. We calculated the fraction of nonsynonymous and synonymous nucleotide substitutions, $p_N$ and $p_S$, respectively, between the consensus sequences of each haplotype (see Appendix 7). We then removed blocks with either $p_N$ or $p_S$ were above a threshold of 0.6. This was done to avoid large errors in estimating the divergence, which we calculated using the Jukes–Cantor correction $d = -(3/4)\ln(1 - 4p/3)$ (*Jukes and Cantor, 1969*). The overall number of such blocks was small (46 out of 1442 in $\alpha$ and 143 out of 2008 in $\beta$) so their removal does not affect our conclusions about the typical blocks. We found that the distributions of both the nonsynonymous and synonymous divergences were similar to those previously reported from comparisons of OS-A and OS-B' (*Rosen et al., 2018*), and consistent with our hybridization hypothesis (*Appendix 6—figure 3*).

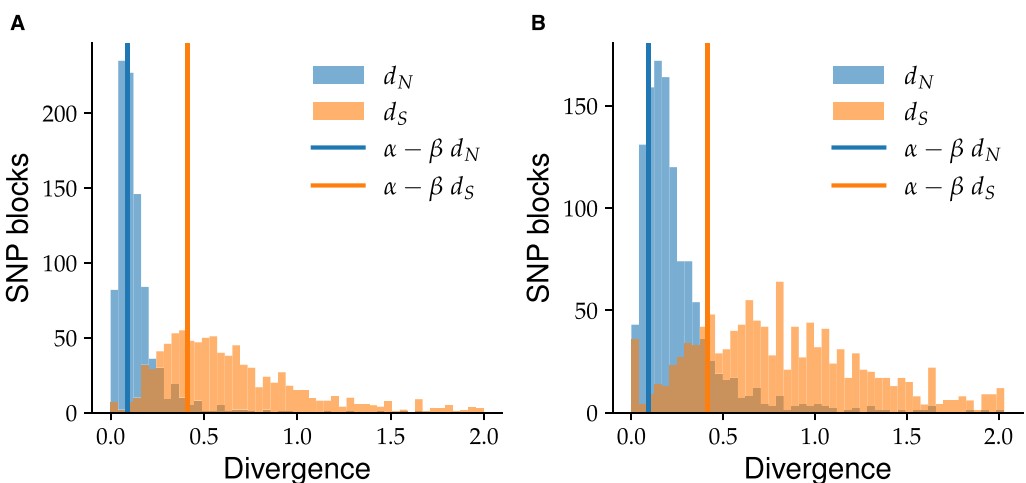

**Appendix 6—figure 3.** Block haplotypes have high synonymous and non-synonymous divergences, consistent with them being generated through hybridization. Panels show histograms of nonsynonymous (blue) and synonymous (orange) divergences between the two haplotypes of SNP blocks from α (**A**) and β (**B**), respectively. Solid lines show the mean nonsynonymous (blue) and synonymous (orange) divergences between α and β.

## Appendix 7

### Genetic diversity analysis

In this section, we present our analysis of the genetic diversity within individual genes in $\alpha$ and $\beta$. In the first subsection, we define the different metrics we used and present the variations in diversity along the genome of both species. In the next four subsections, we compare the diversity of different genes and use the $\alpha$ backbone to roughly infer the time scales of the evolutionary process. In the last subsection, we present a scenario for the evolution of the population.

### Nucleotide diversity along the core genome

We quantified the nucleotide diversity across all genes in both $\alpha$ and $\beta$ using the following procedure. Alignments were trimmed for excess gaps and sequences were assigned to each species as described in Appendix 1. We then define the nucleotide diversity in the usual way as

$$\pi = \frac{1}{S(S-1)} \sum_{i \neq j \in I} \pi_{ij},$$ (7)

where $I$ is an index for sequences in the alignment, $S$ is the sample size, equal to the number of elements in $I$, and $\pi_{ij}$ is the fraction of nucleotide substitutions between $i$ and $j$, excluding gaps. To calculate synonymous ($\pi_S$) and non-synonymous diversities ($\pi_N$) we replace $\pi_{ij}$ with the fraction of synonymous ($\pi_{ij}^S$) and non-synonymous substitutions ($\pi_{ij}^N$), respectively. We calculated $\pi_{ij}^S$ and $\pi_{ij}^N$ using two different methods: the fraction of fourfold degenerate (4D) and onefold degenerate (1D) sites, respectively, at which $i$ and $j$ differ and using the Nei–Gojobori method (**Nei and Gojobori, 1986**). We found that both methods gave similar results and used the one that was most convenient for each analysis.

To calculate the divergence between species, we first determined the proportion of sites that were substituted as follows:

$$p = \frac{1}{S_\alpha S_\beta} \sum_{i \in I_\alpha, j \in I_\beta} \pi_{ij},$$ (8)

where $I_\alpha$ and $I_\beta$ are the indices of sequences from $\alpha$ and $\beta$ in the alignment, and $S_\alpha$ and $S_\beta$ are number of sequences from each species, respectively. The fraction of pairwise substitutions $\pi_{ij}$ is the same as above. We then use the Jukes–Cantor correction (**Jukes and Cantor, 1969**) to correct for the effect of multiple substitutions at a single site

$$d = -\frac{3}{4} \ln \left( 1 - \frac{4p}{3} \right).$$ (9)

The synonymous ($d_S$) and non-synonymous divergences ($d_N$) were calculated in a similar way by replacing $\pi_{ij}$ with $\pi_{ij}^S$ and $\pi_{ij}^N$ in **Equation 8**, respectively. Finally, to calculate the diversity within each species separately, we take the sum in **Equation 7** over $I_\alpha$ or $I_\beta$ only.

### Diversity heterogeneity along core genome

There are two plausible explanations for the heterogeneity within $\alpha$ found in **Figure 3** (main text). The low-diversity genes could be due to local sweeps similar to the ones that formed the divergence troughs shown in **Figure 5C** in the main text. Under this scenario, the high-diversity genes would represent the typical diversity within $\alpha$ that has accumulated since its common ancestor. Since the high-diversity genes are as diverse as typical $\beta$ genes, this scenario would imply $\alpha$ and $\beta$ each had common ancestors at similar times in the past. Alternatively, the low-diversity genes could represent the typical diversity within $\alpha$, with the high-diversity genes being the result of hybridization with other species.

To distinguish between these two possibilities, we developed a method to separate low diversity from high-diversity genes within $\alpha$. Specifically, we calculated the fraction $f_l^S = P_l^S/L_l^S$ of 4D sites at locus $l$ that were polymorphic within the $\alpha$ alignment. Here, $P_l^S$ is the number of synonymous polymorphisms and $L_l^S$ the number synonymous sites at that gene. By using 4D sites, we naturally control for differences in gene conservation. If mutation rates across loci do not vary too much, we

expect the distribution of $f_l^S$ across genes to be given by a Poisson distribution. The mean of this Poisson distribution can be fitted from the fraction of sites that are not polymorphic

$$P(k = 0|\mu) = e^{-\mu} = 1 - \frac{\sum_l P_l^S}{\sum_l L_l^S}, \tag{10}$$

from which we get the following estimate for the mean:

$$\hat{\mu} = -\ln\left[1 - \frac{\sum_l P_l^S}{\sum_l L_l^S}\right]. \tag{11}$$

When we compared this prediction to the data for α, we found that the Poisson distribution gave an excellent fit for around half of the core genes (675 out of 1448) with $f_l^S \lesssim 0.05$. In the remaining core genes (773 out of 1448), $f_l^S$ was much larger than predicted from the Poisson (**Figure 3B** in main text). Note that the agreement in the inset of **Figure 3B** is based on a single fitting parameter. Based on these results, we chose a threshold of $f_l^S \leq 0.05$ to define the α *ancestral backbone*, with all other genes labeled as hybrid.

To validate our partition, calculate the synonymous diversity $\pi_S$ within each group. We found that our criterion produced a clear separation in the synonymous diversity, with low-diversity genes having a mean $\pi_S \approx 1.7 \times 10^{-3}$, compared to $\pi_S \approx 3.4 \times 10^{-2}$ for the high-diversity genes. As a result, we estimate that more than 95% of the genetic diversity within α is the result of hybridization. This result shows that the molecular clock estimate of the time since the most common ancestor based on $T_{MRCA} = \pi_S/(2\mu\tau)$, where $\mu$ is the mutation rate per division and $\tau$ the generation time, should be interpreted with caution. In the case of α the estimate using the average $\pi_S$ across the whole genome would be around an order of magnitude less than the true value, even without considering how mutation rates might change over such long time scales.

We performed a similar analysis for β but did not find any large peak near $f_l^S = 0$ (**Figure 3C** in main text). Instead, the distribution of $f_l^S$ was close to unimodal but with significantly fatter tails compared to the Poisson. Given the excellent fit to the Poisson for α in the region where SNPs are purely due to mutations, this suggests that other mechanisms play an important role in generating the diversity within β. The most parsimonious explanation is that the hybridization process we observed in α has covered most of the original β genome by this point. While it is not clear what the distribution of $f_l^S$ should be in this case, it is plausible that it need not be Poisson (**Sakoparnig et al., 2021**). Further evidence in favor of this hypothesis is the presence of SNP blocks across a large fraction of β core genes, as discussed in the main text and Appendix 6.

## Contribution of hybridization to genetic diversity

The large number of SNP blocks found throughout the genomes of both α and β shows that hybridization is an important source of genetic diversity for the *Synechococcus* population. However, because of the requirement for perfect linkage within SNP blocks, the total number of SNPs that are the result of hybridization is likely much higher than the one we infer from our algorithm. To obtain quantitative estimates of the total contribution from hybridization to the diversity, we use α as a control. As shown previously, only ≈5% of the total diversity in the high-diversity loci from α can be attributed to mutations. These loci contained 603,588 sites and 64,462 SNPs, of which 10,391 SNPs were found inside SNP blocks. Thus, only around 17% of the SNPs that are likely the result of hybridization, or hybrid SNPs, are included in the SNP blocks.

Within the β core genome, we have 1,591,476 sites and 146,906 SNPs, of which 14,819 were within SNP blocks. Assuming a similar ratio of hybrid SNPs to SNPs in SNP blocks as in α, we estimate that around 90,000 or approximately 60% of SNPs in β are the result of hybridization. This is consistent with β having evolved over longer time compared to α. However, our knowledge of how multi-locus SNP patterns change over the course of evolution is limited, so the estimates for β should be interpreted with caution. In particular, if larger segments continue to be broken up due to recombination, it is likely that the fraction of hybrid SNPs within blocks would decrease over time for a constant minimum block size $l_{min}$ (see Appendix 6). Thus, the overall contribution of hybridization to the β genetic diversity may be even higher.

## Geological constraints on microbial evolution

Combining genetic diversity analysis with evidence from the geological record of the Yellowstone caldera allows us to make inferences about the evolutionary history of the *Synechococcus* population. If we assume that most of the fixed variants between species are the result of mutation accumulation, then we can use the typical synonymous divergence between species to infer the time since the most recent common ancestor for the whole population. Typical bacterial mutation rates for natural populations are around $\mu \sim 10^{-9.5}$ per bp per generation (*Chen et al., 2022*). Using this rate as a baseline and assuming generation times of $\tau \sim 1 - 10$ days gives an estimate of $T_{MRCA} \sim 10^{6.5} - 10^{7.5}$ years (*Rosen et al., 2018*).

For our argument, the lower bound is the more important one since it determines whether we can exclude the single-colonizer hypothesis. A $T_{MRCA}$ below $10^{6.5}$ years would imply a shorter division time or higher mutation rate than our estimates. Our lower bound of 1 day for the division time is around the same as the fastest division rate observed in the laboratory. If the population is in steady state, as it appears to be, the division time sets the maximum generation time, which can be attained if the death rate is equally high. If the death rate is lower than this, other factors would limit cell division and the generation times would be longer. Taken together, we can be fairly confident that a generation time of 1 day is a hard lower bound.

The mutation rate is less certain since we do not have any reliable measurements and experimentally measured mutation rates might not be a good predictor of substitution rates over evolutionary times (*Rocha, 2018*). We can instead use other bacteria to estimate a range of values. In wild-type *E. coli*, mutation rates of $\approx 2 \cdot 10^{-10}$ per bp per generation have been measured in laboratory conditions (*Lee et al., 2012*), but mutator strains with rates that are up to 100 times higher are also known (*Sniegowski et al., 1997*). In natural populations, rates as high as $4 \cdot 10^{-9}$ have been inferred using longitudinal sequencing (*Moran et al., 2009*). If we use $10^{-8.5}$ mutations per bp per generation as an upper bound, we would infer a lower bound of $T_{MRCA} \sim 10^{5.5}$ years, which very narrowly overlaps with the time since the formation of the Yellowstone caldera. Thus, while it is not inconceivable that the three species could have diverged within the caldera, such a scenario would require all the relevant parameters to be at the limit of currently known values. In addition, such a scenario would require additional hypotheses to explain why the strains initially diverged into distinct species, with seemingly little genetic exchange between them (*Rosen et al., 2018*), and then only much later started to hybridize. Taken together, current evidence suggests the most likely scenario is that $\alpha$, $\beta$, and $\gamma$ diverged from a common ancestor before the formation of the caldera around 640,000 years ago.

## Upper bound on mixing time between springs

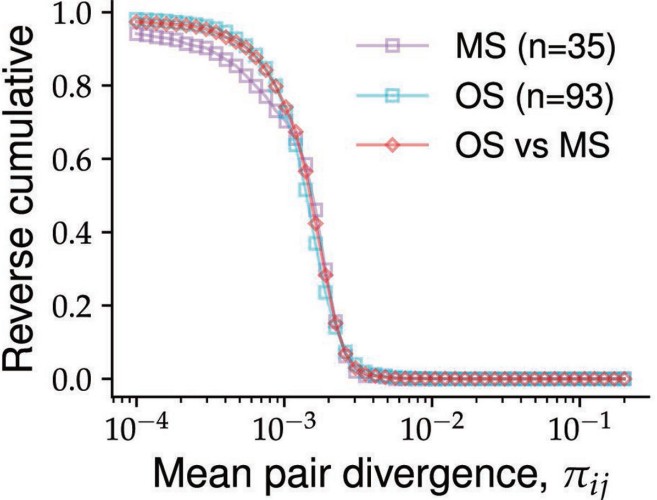

**Appendix 7—figure 1.** Non-hybrid regions of α have similar levels of diversity within and between springs. The distribution of pairwise synonymous diversity was calculated across non-hybrid genes from the alpha $\alpha$ population
*Appendix 7—figure 1 continued on next page*

*Appendix 7—figure 1 continued*
for different subgroups of cells. Square symbols show the distributions for pairs of cells from the same spring (purple for Mushroom Spring and cyan for Octopus Spring). The cyano triangles show the distribution for a random subsample of cells from the Octopus Spring of the same size as the Mushroom Spring sample ($n = 35$) for comparison. The red diamonds show the distribution for pairs of cells from different springs. Non-hybrid genes were determined using a 5% cutoff in the fraction of polymorphic synonymous sites (see text).

By using the diversity within $\alpha$ as a proxy for time, we can obtain estimates on the time scales of spatial mixing that influences the diversity within the population. To do this, we calculated the pairwise distances in the low-diversity regions of the $\alpha$ genome between pairs of cells in each spring separately and pairs of cells from different springs. We found that all three distributions were similar (*Appendix 7—figure 1*). The mean of the distribution for cells Mushroom Spring was slightly lower than those from the Octopus spring ($\pi_S = 8.6 \cdot 10^{-4}$ vs $\pi_S = 1.1 \cdot 10^{-3}$) and this difference could not be explained by differences in sample sizes between the two springs. However, both values were very close to the mean of cells from different springs ($\pi_S = 1.0 \cdot 10^{-3}$). This suggests that the time scale of spatial processes relevant for the long-term evolution of the population, $\tau_m$ is much shorter time scales than $T_{MRCA}$ for $\alpha$. Taking $T_{MRCA} \sim 3 \cdot 10^3$ years as a reference point sets an upper bound of $\tau_m \ll 3 \cdot 10^3$ years.

## Proposed scenario for *Synechococcus* evolution

We propose the following scenario that is consistent with our previous discussion. First, we argued that a small number (3–6, but possibly more) of *Synechococcus* populations that had been evolving separately for $\sim 10^6 - 10^7$ years independently colonized the Yellowstone caldera after its formation 640,000 years ago. After colonization, they started to hybridize, leading to a gradual mixing of their genomes (*Figure 8*). This can be most clearly seen in the $\alpha$ species, where the diversity in core genes that do not show signs of hybridization is more than an order of magnitude less than those that do. Moreover, genes that have undergone hybridization account for the majority of diversity within $\alpha$. Within $\beta$, we do not observe such heterogeneity, but we find SNP blocks are even more prevalent than in $\alpha$ (*Appendix 6—table 1*). The simplest explanation for this pattern is that $\beta$ has been undergoing hybridization for longer, and by now, all of its core genome has been mixed with other species. Consistent with this explanation, we find that only a small fraction of SNP blocks within $\beta$ can be traced back to $\alpha$ (*Appendix 6—table 1*). Most of the $\beta$ SNP blocks cannot be traced to either $\alpha$ or $\gamma$ and are likely the result of a combination of hybridization with species that are not currently within the Lower Geyser Basin, have abundances lower than our detection threshold, or have gone extinct.

## Appendix 8

### Estimate of recombination parameters

There are several important parameters that determine the impact of recombination and hybridization on the evolution of the population. First are the rates of recombination and mutation in the population. We denote the population recombination rate by $\rho = 2TR$ and the population mutation rate by $\theta = 2T\mu$, where $T$ is the average time since the most common ancestor, $R$ is the recombination rate per site, and μ is the mutation rate per site. Note that in the panmitic Wright–Fisher model, $T$ in units of generation time is equal to the population size $N$, but this is not generally true (*Neher and Hallatschek, 2013*; *Weissman and Hallatschek, 2014*; *Birzu et al., 2018*; *Birzu et al., 2021*). A second important parameter is typical length of recombined segments, which we denote by $L_r$. Finally, there is the typical divergence of recombined segments. In this Appendix, we use the nucleotide distance $p$ or the synonymous distance $p_S$, as defined in Appendix 7. We estimated these parameters for the hybridization process, which we argued previously is the dominant source of diversity in the population.

Before describing the analysis, it is important to note that while some of the simple statistics discussed here may be captured by simple neutral models, the underlying dynamics may be far more complex. Indeed, the analysis in the main text shows extensive evidence of selective sweeps affecting the genetic diversity across a broad range of genomic length scales, contrary to simple neutral theory. Moreover, while our analysis shows that—over time scales implied by our sequencing depth—the *Synechococcus* population is effectively well-mixed within the caldera, the actual population is obviously not. *Synechococcus* cells live in a dense microbial mat spread across thousands of springs separated by distances up to ~$10^2$ km (*Ward et al., 1998*). As a result, we expect the long-term evolution of the population to be described by *effective* parameters, which may have little relation to recombination rates and population sizes directly measurable in the field or in the laboratory (*Neher and Hallatschek, 2013*; *Weissman and Hallatschek, 2014*; *Birzu et al., 2018*; *Birzu et al., 2021*). Understanding the relationship between the measurable parameters describing recombination, selection, and spatial dynamics of individual cells and the effective parameters at the population scale is an important and interesting topic for future work.

Many different methods have been proposed to estimate these or other closely related parameters (*Croucher et al., 2015*; *Didelot and Wilson, 2015*; *Lin and Kussell, 2019*). All of the methods we are aware of are based on models that make various assumptions that may not be valid for any given population, such as ours. In our case, an accurate estimate of these parameters requires a quantitative model of the hybridization process and is beyond the scope of the paper. Here, we instead used the results of our previous analysis to get rough estimates of these parameters and compared them to two other commonly used methods.

We considered each species separately, starting with $\alpha$. For most evolutionary models, $\theta$ can be estimated from the nucleotide diversity π (*Neher and Hallatschek, 2013*). As in the main text, we used only synonymous sites to control for differences in the strength of purifying selection or the accumulation of non-synonymous mutations during rapid adaptation. Based on our previous analysis, we estimated the population synonymous mutation rate $\theta_S \approx 2 \cdot 10^{-3}$ (Appendix 7). Because of the clear distinction between the ancestral backbone and hybrid genes in $\alpha$, we expect this estimate to be reasonably accurate. A rough estimate of $\rho$ can be obtained from the linkage disequilibrium curves (*Figure 2* in main text). Our previous analysis showed that $\rho \approx 0.03$ and $\rho \approx 0.12$ gave good fits for $\alpha$ and $\beta$, respectively. Taking the ratio, we found a ratio between recombination and mutations of $\rho/\theta_S \approx 15$—much higher than previous estimates.

Estimating the mutation rate in $\beta$ is more difficult because there are no genes unaffected by hybridization. We instead used the fraction of SNPs found in SNP blocks to estimate the diversity purely due to mutations. In Appendix 7, we estimated that around 40% of SNPs are the result of mutations. Multiplying with the synonymous diversity gave $\theta_S \approx 0.02$. Using this estimate, we found $\rho/\theta_S \approx 6$, which is lower than in $\alpha$ but still significantly higher than values that have been reported previously.

The other recombination parameters can also be inferred from our analysis. As we showed previously, the typical divergence of hybrid segments is given by the synonymous distance between species $p_S \approx 0.4$. We estimated the average size of recombined segments from the linkage curves.

The recombination rate between two sites on the genome increases linearly with the separation $x$ for distances shorter $L_r$. In the standard neutral model, this leads to the scaling $\sigma_d^2(x) \sim 1/x$ at intermediate distances (see Appendix 4). For $x > L_r$, the recombination rate saturates to the genomewide recombination rate $\sim \rho L_r$, which results in deviations of the linkage decrease from the predicted $\sigma_d^2(x) \sim 1/x$ scaling. In $\alpha$, we observe a deviation around $x \sim 500$ bp. Note that we verified that this plateau cannot be explained by the smaller finite sample size in $\alpha$. A similar argument gives a shortened segment length $L_r \sim 300$ bp in beta. While we cannot directly test that the plateau is not due to sample size, the small effect of subsampling we observed earlier suggests it is an unlikely explanation (*Appendix 4—figure 2*). Note that in both species, our estimated genomewide recombination rate $\rho L_r$ is much greater than one, consistent with the genomewide linkage being close to the unlinked control, as discussed in the main text.

## Comparison to other methods

We compared our analysis to standard methods. ClonalFrameML and other related software are perhaps the most widely used methods to infer bacterial recombination (*Didelot and Wilson, 2015*). Such methods assume that recombination rates are rare enough that they can be described by a Poisson process on an asexual tree, similar to mutations. We refer to the assumption as the *clonal frame hypothesis*. ClonalFrameML uses the inferred phylogenetic tree from pairwise distances to identify discrete recombination events and assign them to branches on the tree. The recombination rate, genetic divergence, and length of recombined segments are then inferred from these events.

Note that the clonal frame hypothesis is inconsistent with the rapid decrease in linkage we observed (*Figure 2* in main text). Nevertheless, it is useful to see how the method performs on our data.

We used a subset of high-coverage SAGs to create whole-genome alignments for $\alpha$ and $\beta$, which we then analyzed with ClonalFrameML. We first selected SAGs more than 75% complete, which resulted in a dataset of 2 $\alpha$ and 12 $\beta$ SAGs (*Appendix 1—figure 1*). For each species, we chose the genes that mapped to OS-A or OS-B' and created concatenated alignments of the entire genome in the order of the references. Missing genes or sequences were replaced by gaps. We constructed the phylogenetic tree using FastTree and ran ClonalFrameML on the alignments and trees with default parameters.

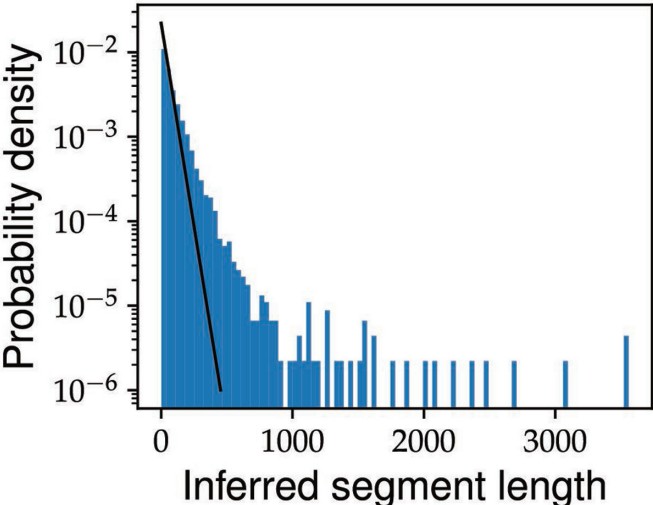

**Appendix 8—figure 1.** ClonalFrameML does not converge to a self-consistent set of recombination events in high-coverage $\beta$ SAGs. Whole-genome alignments for 12 high-coverage $\beta$ SAGs were generated as described in the main text. The phylogenetic tree for this subset was inferred using FastTree with default parameters and used as input for ClonalFrameML. The histogram shows the distribution of the length of recombined segments inferred by ClonalFrameML. The solid black line shows the maximum-likelihood exponential distribution used to fit the inferred segments. Note the heavy tail of the inferred distribution which is inconsistent with the exponential fit.

For both species, ClonalFrameML inferred a ratio of recombination to mutation rates $\rho/\theta \approx 1$ ($R/\theta$ in *Didelot and Wilson, 2015*) and a fraction of nucleotide differences for imported segments of $p = 0.07 - 0.09$ ($\nu$ in *Didelot and Wilson, 2015*). Note that both $\theta$ and $p$ reported by ClonalFrameML refer to nucleotide rather than synonymous differences. The values of $p$ are around half of the distance between species $p \approx 0.15$ and are inconsistent with our metagenome analysis which showed no genomes present in OS or MS at these divergences. The main difference between the species was in the average length of recombined segments, which was almost 10 times higher in $\alpha$ ($L_r = 365$) compared to $\beta$ ($L_r = 45$).

A detailed comparison of the results of ClonalFrameML with our analysis is beyond the scope of this paper. Nevertheless, we found several inconsistencies in the assignment of recombined segments by ClonalFrameML. Because of the lack of data in $\alpha$, we focused this analysis on $\beta$. First, we found that the distribution of recombined segment lengths was inconsistent with the exponential distribution assumed by ClonalFrameML and had a mean that was almost double the inferred $\delta = 45$ (*Appendix 8—figure 1*). Second, manual validation of a small sample of randomly chosen recombined segments revealed identical patterns present in other cells that were not assigned to the recombination event (data not shown). We suspect that these misassignments are due to the inconsistency of the observed segments with the asexual tree.

The failure of ClonalFrameML to fit the data should not be surprising. As we showed, the low levels of linkage at whole-genome scales in $\alpha$ and $\beta$ are inconsistent with the existence of a clonal frame assumed by the model. In addition, manual validation of some of the recombined segments detected revealed spurious segments caused by gaps in coverage due to the incomplete nature of the SAGs. These results caution against the naive application of methods such as ClonalFrameML to SAG datasets in general, but also to other populations for which the clonal frame hypothesis may not hold.

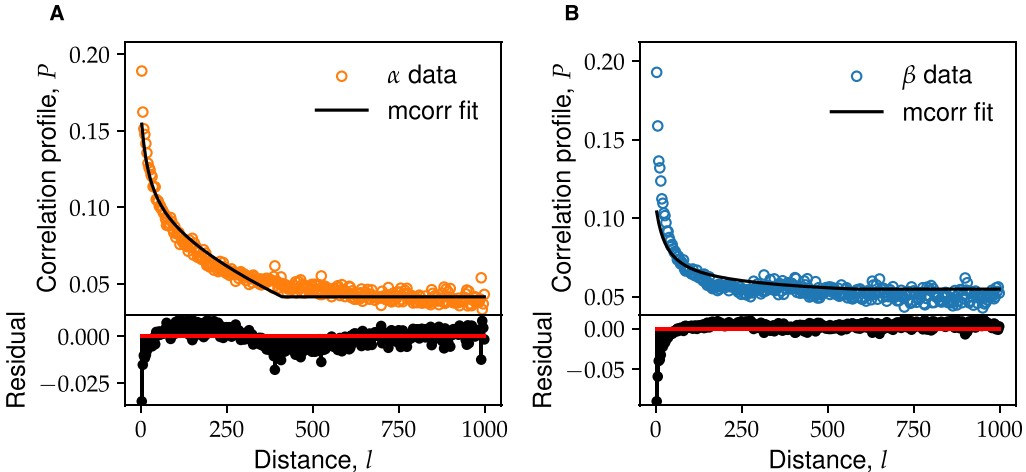

**Appendix 8—figure 2.** SNP correlation profiles in α agree with mcorr fits but show significant discrepancies at short distances in β.

An alternative method for fitting the clonal frame model was proposed recently by *Lin and Kussell, 2017*; *Lin and Kussell, 2019*. Instead of identifying recombination events directly, they use the correlations between SNPs to infer the recombination parameters statistically. Specifically, they model a local (sample) population that evolves according to a neutral model with recombination. They assume that the population recombines with a larger and more diverse 'pool' that evolves according to the same model but with different parameters. Under this model and assuming the recombination to mutation rate ratios are the same in the sample and pool populations, they show that the conditional probability that two sequences differ at a site given that they differ at a second site $l$ bp away, $P(l)$, depends only on population mutation rate ($\theta$ in our notation), the population recombination rate ($\rho$) and the average length of recombined segments ($L_r$). By fitting the analytical expression for $P(l)$ from the model to the data, all of the model parameters can be inferred.

**Appendix 8—table 1.** Comparison between recombination parameters obtained from our analysis, mcorr (**Lin and Kussell, 2019**), and ClonalFrameML (**Didelot and Wilson, 2015**).

For each species, the four columns show the estimated ratio of recombination and mutation rates $\rho/\theta$, the average length of recombined segments $L_r$, the fraction of nucleotide differences between host and recombined segments, $p$ and the ancestral nucleotide diversity $\theta$ estimated by the different methods. All quantities were converted to the notation used in the text as follows. The mcorr columns are the best fit values of the parameters $\phi_s/\theta_s, \bar{f}, d_p$, and $\theta_s$ from **Lin and Kussell, 2019**. The ClonalFrameML columns are the maximum likelihood values of the parameters $R/\theta, \delta, \nu$, and $\theta$ from **Didelot and Wilson, 2015**. Note that our analysis and mcorr only used synonymous sites, while ClonalFrameML used all sites. The diversity parameters ($\rho/\theta$, $p$, and $\theta$) from ClonalFrameML that are not directly comparable to the other two analyses are marked by asterisks.

|  | Species | $\rho/\theta$ | $L_r$ | $p$ | $\theta$ |
|---|---|---|---|---|---|
| Our analysis | α | ~15 | ≳500 | 0.4 | $2 \cdot 10^{-3}$ |
|  | β | ~6 | ≳300 | 0.4 | $\sim2 \cdot 10^{-2}$ |
| mcorr | α | 0.38 | 411 | 0.11 | $3 \cdot 10^{-3}$ |
|  | β | 0.45 | 573 | 0.06 | $1.5 \cdot 10^{-2}$ |
| ClonalFrameML | α | 0.88* | 365 | 0.07* | $3 \cdot 10^{-4*}$ |
|  | β | 1.5* | 45 | 0.09* | $3 \cdot 10^{-4*}$ |

We generated alignments of the high-coverage $\alpha$ and $\beta$ SAGs using Mauve (**Darling et al., 2004**) and followed **Lin and Kussell, 2019** to fit $P(l)$. Note that unlike in ClonalFrameML, we used only synonymous sites for the mcorr analysis, so the divergences cannot be directly compared between the two methods. We found the model fit the data for $\alpha$ reasonably well up to two distances around 300 bp, with the exception of the first two points. However, the model was not able to capture the slower decay in correlations at longer genomic distances (see also Appendix 4). In $\beta$, the fit at short distances was considerably poorer (**Appendix 8—figure 2**). In particular, correlations between nearby SNPs were ~2× higher in the data compared to the model. The increased correlations at short distances could be due to genetic hitchhiking during gene sweeps or could be a sign of subtle forms of population structure within $\beta$. Distinguishing between these scenarios is an interesting topic for future work.

Compared to ClonalFrameML, the recombined fragment lengths inferred by mcorr were larger (especially for $\beta$) and were reasonably close to our estimate. In addition, the mcorr estimate of the diversity due to mutations ($\theta$ in our notation) agreed remarkably well with our estimates in both species. Similar to ClonalFrameML, the synonymous divergence of hybrid segments was between the divergence between species and the diversity within species and was inconsistent with our metagenome analysis. Finally, the ratio of recombination and mutation rates was similar to the one inferred from ClonalFrameML and more than an order of magnitude smaller than our estimates. These results are summarized in **Appendix 8—table 1**. We leave a more careful analysis of the differences as a topic for future work.

## Appendix 9

## Variation in hybridization across samples

In this section, we describe how we quantified associations between genomic features and populations from different samples. Our primary focus was on differences in hybridization patterns, so we analyzed the variations in hybrid gene and SNP block haplotype frequencies across different samples and locations. Given the small number of samples in our study, we limited the number of statistical tests we performed in order to avoid false positives. For each of the two main species, $\alpha$ and $\beta$, we tested for the effect of sample location and sample composition separately, as described below.

## Variation in SNP block haplotype frequencies

We considered two distinct hypotheses that could lead to associations between block haplotypes and environmental conditions from which the sample was taken. First, we considered if the community composition was associated with specific haplotypes by comparing samples in which a single species was present to samples which contained a mixture of $\alpha$ and $\beta$. For $\alpha$, we chose the 48 cells from pure-$\alpha$ sample OS2005 as the test group and the 45 $\alpha$ cells from MS2004, MS2006, and OS2009 as the control. For $\beta$, we chose the 44 cells from the pure-$\beta$ sample MS2005 as the test group and the 46 $\beta$ cells from MS2004 and MS2006 as the control. We denote these two tests as '$\alpha$-composition' and '$\beta$-composition' tests, respectively. Second, we considered the hypothesis that sample location was associated with specific haplotypes. For this test, we divided all cells from each species into two groups labeled MS and OS, based on the location from which the sample was taken. These tests are denoted by '$\alpha$-location' and '$\beta$-location', respectively.

We used the following permutation test to find block haplotypes that were significantly associated with each environmental variable. We first removed all blocks in which the minority haplotype was below an arbitrary cutoff frequency, set at 5 sequences. This was done in order to avoid tests which were statistically underpowered. We then calculated the two-sided p-value for the association for each block individually using Fisher's exact test.

**Appendix 9—table 1.** Result of tests for associations between SNP block haplotypes and two environmental variables related to the samples.
The rows show the results of the different tests described in the main text. The p-value $p$ for each block was determined using Fisher's exact test and minimum p-value from the randomized control $p_{rand}^{min}$ was used to correct for false-discovery rates (see main text for details). The first four columns show the total number of blocks in each category. The last column shows the largest value of the $p_{rand}^{min}/p^{max}$ (most significant association) for each test.

| Test | Tota blocks | Blocks tested | Significant ($p_{rand}^{min}/p > 1$) | Highly significant ($p_{rand}^{min}/p > 10$) | Largest $p_{rand}^{min}/p$ |
|---|---|---|---|---|---|
| α-Composition | 1000 | 602 | 13 | 1 | 12.0 |
| α-Location | 1000 | 724 | 6 | 1 | 522 |
| β-Composition | 1483 | 848 | 0 | 0 | 0.8 |
| β-Location | 1483 | 1306 | 15 | 1 | 32.9 |

Because the sampling depth for each block is variable due to variations in coverage, standard methods that correct for the false-discovery rate are not well-suited in our case. We use the following procedure to determine significant associations. We randomized the composition of the test and control cell groups for each test, while keeping the total number within each group fixed. We then recalculated the p-values using the same procedure as before and recorded the lowest p-value across all blocks $p_{rand}^{min}$. We used the ratio $p_{rand}^{min}/p$ between the minimum p-value in the randomized control and the p-value in the test group as a measure of significance of the association. We arbitrarily set a threshold value of $p_{rand}^{min}/p_t = 10$, above which associations were deemed highly significant.

The results of these tests are summarized in *Appendix 9—table 1*.

Overall, we found very few blocks that were associated with either the sample composition or location. This is consistent with the results from the main text (*Figure 7D*), which showed that the block haplotypes are close to unlinked even on distances comparable to the whole genome.

The largest number of associations was with the sample composition in the $\alpha$ species, consistent with the previously mentioned results, which showed a small but statistically significant increase in linkage within the $\alpha$ species compared to the $\beta$. Nevertheless, the overall fraction of blocks whose haplotypes were strongly associated with the sample composition was ≈1.2% and still small. We therefore concluded that neither the composition of the population within samples nor the location has a large effect on the hybridization patterns revealed by the SNP blocks. A complete table with the results of these tests can be found in the supplementary data tables.

## Variation in hybrid gene transfers

We also examined variations in the rate of transfers of whole-gene alleles across species. The lower overall number of such events compared to the SNP blocks makes statistical tests for associations much more likely to yield false positives in this case. As a result, we limited the analysis to a qualitative comparison of the number of different types of transfers across samples shown in *Appendix 9—figure 1*. We did not observe any systematic variations with either sample location or sample composition for either species.

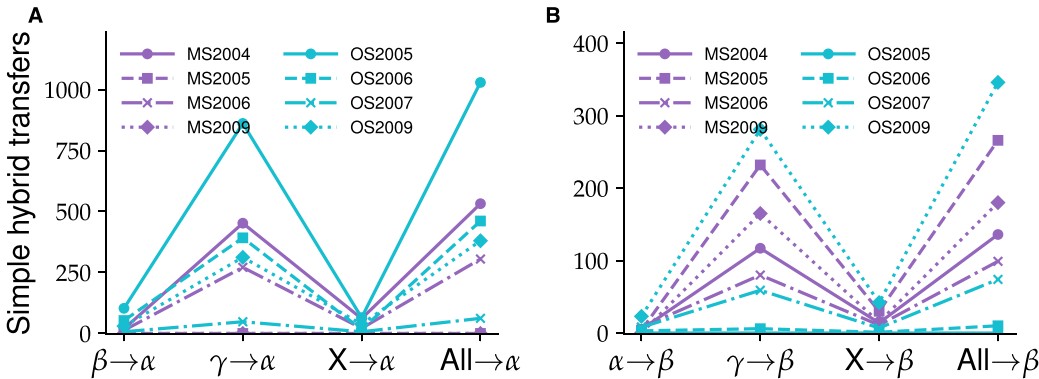

**Appendix 9—figure 1.** Numbers and types of gene transfers between species are similar across samples. The number of hybrid gene transfers from each donor found in α (**A**) and β (**B**) cells across all eight samples in the data. The labels $A$, $B$, and $C$, denote α, β, and γ, respectively, while $X$ denotes other *Synechococcus* clusters.

## Appendix 10

### Data quality control

Beyond the processing steps outlined in Appendix 1, we also explicitly verified that the hybridization patterns we observe cannot be explained by sequencing and genome assembly artifacts. Specifically, we tested if misassembly of reads from distinct alleles of the same gene could explain the presence of hybrid sequences. We considered two main sources of such reads: cross-contamination from neighboring reaction wells during whole-genome amplification (WGA) and the presence of paralogs horizontally transferred from other species.

Distinguishing between hybridization and artifacts is hard because both can lead to similar signals. Here, the structure of data provides us natural controls. First, $\alpha$ has genes with low diversity and little sign of hybridization and high-diversity genes with extensive hybridization. We can use former to establish a baseline for genome coverage and check for deviations in the later. If the hybridized genes are caused by misassemblies of actual between $\alpha$ genes and contamination from $\beta$ or between different alleles of the same gene within $\alpha$ cells, we should see an increase in the coverage at these loci. Second, the variability in species composition across samples provides controls for cross-contamination between reaction wells. Specifically, OS2005 contained 48 cells, all classified as $\alpha$, while MS2004, MS2006, and OS2009 together contained 45 $\alpha$ cells amplified on plates with mixtures of $\alpha$ and $\beta$ cells in similar proportions (*Appendix 10—figure 1*). We will refer to the former as the *control SAGs sample* and the latter as the *test SAGs sample*. We used these two groups as a measure of the contribution of cross-contamination from neighboring $\beta$ cells to the observed hybridization in $\alpha$.

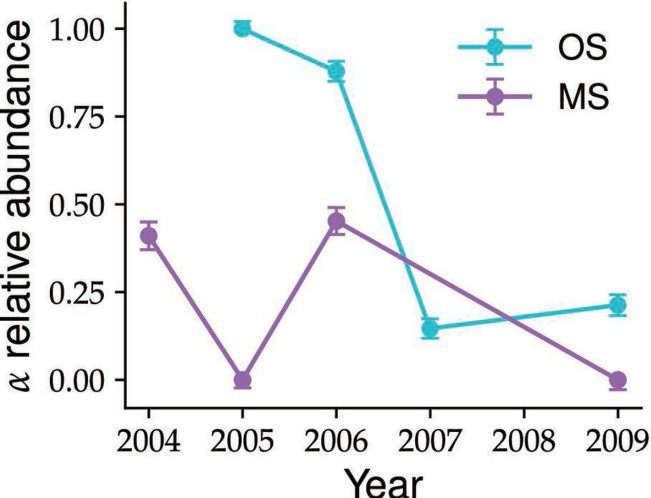

**Appendix 10—figure 1.** Species composition is highly variable across samples.

We first divided the core $\alpha$ OGs into test and control groups as follows. We first divided OGs into low- and high-diversity groups based on the probability that synonymous sites are segregating in the sample using a cutoff at 0.05. From the low-diversity OGs, we chose the subset that were grouped into $\alpha$ and $\beta$, or $\alpha$, $\beta$, and $\gamma$ clusters during the pagenome construction (approximately 300 each out of a total of ~800). These OGs were nearly clonal in the $\alpha$ sample and showed little to no evidence of hybridization in our entire data. These OGs will form the *control OG group*. From the high-diversity OGs, we selected those where all sequences were grouped together in a single mixed-species cluster (~300 out of a total of ~600). These include the cases with the most extensive hybridization and thus formed the *test OG group*.

We validated that core gene coverage can be described by a random process in which each gene has a fixed probability $p$ of being present in the assembly. We fitted the baseline probability $p$ using the low-diversity genes from the OS2005 $\alpha$ cells. Many of these genes were clonal within the entire $\alpha$ sample and showed no evidence of hybridization with the other species in our sample. They thus provide a good baseline for the expected coverage in the absence of contamination. While

the coverage among different SAGs was highly variable (*Appendix 10—figure 2A*), we found that the distribution of coverages of different genes was well fitted by a Gaussian distribution with the mean and variance given by assuming each gene is an independent Bernoulli random variable with success probability $p \approx 0.29$. We tested the independence assumption by calculating the covariance between the presence of different genes (*Appendix 10—figure 2C*). Under our very simple model, any correlations between the coverage at different loci would be purely due to statistical fluctuations caused by finite sample sizes. The distribution of diagonal ($C_{aa}$) and off-diagonal ($C_{ab}$) components of the covariance matrix can then be calculated exactly given $p$. We compared the values obtained from the data with these predictions and found a very good agreement (*Appendix 10—figure 2D*).

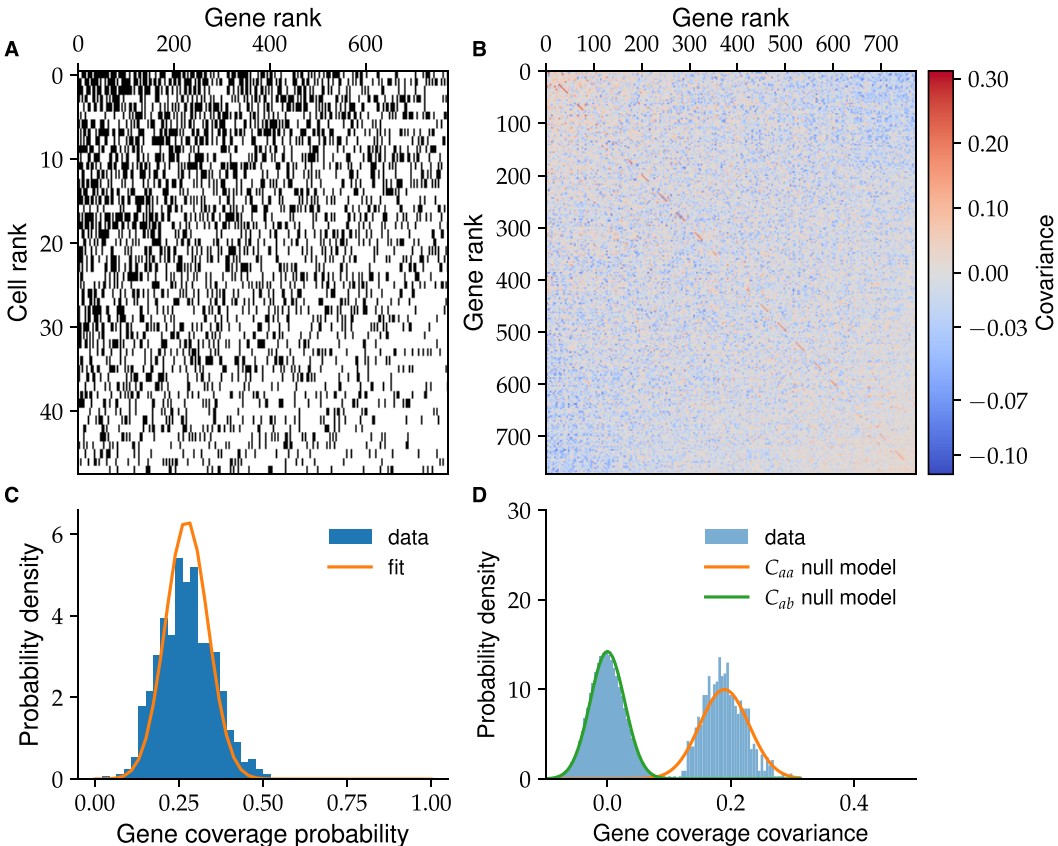

**Appendix 10—figure 2.** Genome coverage across core genes is well-explained by independent presence–absence model. (**A**) shows the presence-absence matrix $P_{ia}$ of low-diversity OGs in the control single-amplified genome (SAG) sample OS2005 (see main text for definitions). Rows and columns were ordered in decreasing order of the sums across the columns and rows, respectively. (**B**) shows the correlation matrix between OG presences defined as $C_{ab} = N^{-1} \sum_{i=1}^{N} \langle (P_{ia} - \overline{P_i})(P_{ib} - \overline{P_i}) \rangle$, where $\overline{P_i} = N^{-1} \sum_{a=1}^{L} P_{ia}$. Note the use of the mean abundance for each SAG rather than $p = (NL)^{-1} \sum_{i,a} P_{ia}$ to account for the large variability in coverage between SAGs. (**C**) shows the distribution of OG presences $\overline{P_a} = N^{-1} \sum_{i=1}^{N} P_{ia}$. The solid orange line shows the Gaussian fit under the assumption that each gene is a Bernoulli random variable with success probability $p$. (**D**) shows the distribution of the entries in the covariance matrix from (**B**) together with the predictions for the diagonal (orange) and off-diagonal (green) components under the assumption that each gene is independent and present in the final assembly with probability $p$. Note the value of $p$ was obtained from panel (**C**), and there are no free parameters for the predicted curves.

Having validated our gene presence model, we can now use it to perform simple but quite powerful tests for different artifacts. Consider the extreme case where all of the hybrid genes we observe are the result of misassemblies of reads from two distinct alleles, one from $\alpha$ and one from either $\beta$ or $\gamma$, from the same reaction well. Note that for this argument, it is unimportant whether the alternative allele is due to contamination or genuine horizontal transfer that resulted in two distinct, unmixed copies of the gene. In either case, the probability of observing the gene in our

assembly would be $P_{obs} = p(2 - p) \approx 0.5$, which is significantly higher than for a true single-copy gene. We tested for such an increase by comparing the distribution of coverages among $\alpha$ cells in the OS2005 sample of high-diversity genes that formed a single mixed cluster across all sequences in our data (see Appendix 1) and low-diversity genes with distinct $\alpha$ and $\beta$ or $\alpha$, $\beta$, and $\gamma$ clusters. We found the two distributions were very similar to each other (**Appendix 10—figure 3A**), with a small but statistically significant increase in the mean coverage at low-diversity genes with distinct species clusters ($p_{control} \approx 0.3$ vs $p_{test} \approx 0.28$). The difference is in the opposite direction from the one expected from the misassembly hypothesis, but it corresponds to less than an extra copy of a gene on average and is unlikely to have any consequence for our conclusions.

Note that the misassembly hypothesis above would also predict a specific pattern for the sequences within $\alpha$, with a large fraction $2(1 - p)/(2 - p) \approx 0.83$ having either the pure $\alpha$ or the pure $\beta$ alleles, and $p/(2 - p) \approx 0.17$ containing a mixture. We do not find evidence of such patterns in our data. More importantly, the 0.17 fraction is an upper bound on the maximum frequency of mixtures that can be explained through misassembly, as any lower amounts of contamination would lead to fewer chimeric sequences. Note that both of the above tests take advantage of the fact that the typical genome coverage is low.

Using a similar method to the one described above, we can also check for contributions from cross-contamination between reaction wells on the formation of chimeric sequences. Here, we compared the coverage distribution of mixed-species orthogroups in the mixed $\alpha - \beta$ samples described previously with the control group from the all $\alpha$ sample OS2005. If cross-contamination with $\beta$ was a significant contributor to the presence of mixed-species orthogroups, one would expect a large difference between the two distributions. We did observe a large systematic difference between the two groups in the overall coverage $p$ across all genes (data not shown), which was likely due to batch effects in the amplification and sequencing. We therefore compared the mean-centered distributions between the two groups and found no difference between them and the fit to the independent presence-absence model from before (**Appendix 10—figure 3B**).

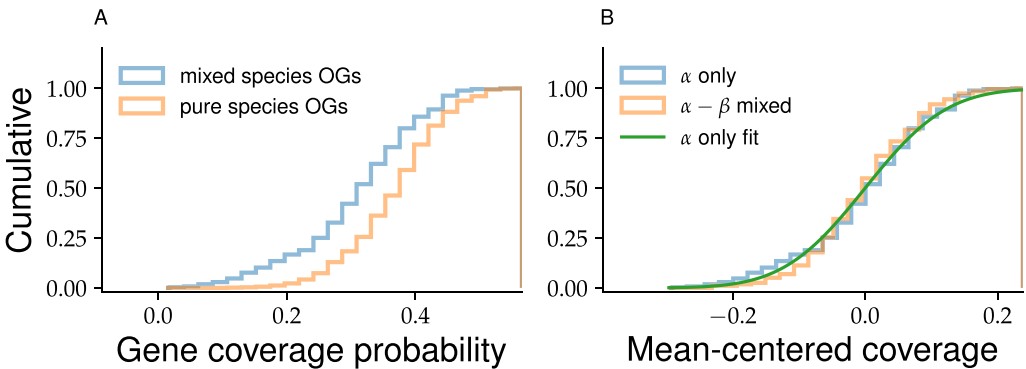

**Appendix 10—figure 3.** Variability in core gene presence is inconsistent with observed hybridization being dominated by artifacts. (**A**) compares the presence distributions among the control single-amplified genome (SAG) sample of the test OG group (blue) with the control group (orange), as defined in the main text. (**B**) compares the presence distributions of the test OG group in the control (blue) and test (orange) SAG samples. The green line shows the fit from **Appendix 10—figure 2C**. All distributions were centered at zero to control for batch effects (see main text).

Beyond the indirect presence of hybridization given by the presence of mixed-species orthogroups, we also tested for any effect of cross-contamination on the presence of linkage blocks. We applied the same algorithm described in Appendix 6 to the subsample of 48 $\alpha$ cells from OS2005 and compared the results with those from random samples of $\alpha$ cells from samples with mixtures of $\alpha$ and $\beta$ cells. We found that the number of blocks detected in OS2005 was similar to the bootstrapped distribution from mixed $\alpha - \beta$ samples (**Appendix 10—figure 4A**). We also found the distribution of block lengths and divergences between block haplotypes to be similar as well (**Appendix 10—figure 4BC**).

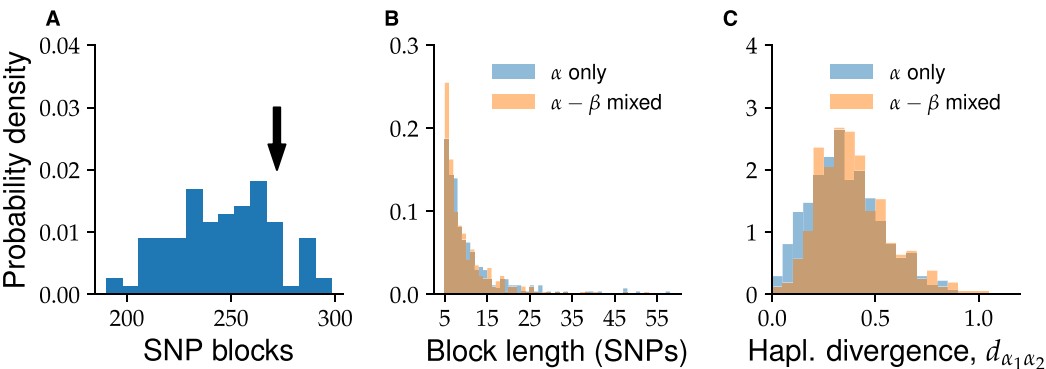

**Appendix 10—figure 4.** Linkage blocks are robust to large variations in species composition across samples. (**A**) Bootstrapped distribution of the number of linkage blocks detected in alignments across fourfold degenerate sites. The distribution was obtained by randomly choosing groups of 48 single-amplified genomes (SAGs) from samples containing mixtures of α and β SAGs (MS2004, MS2006, OS2006, OS2007, and OS2009). The arrow shows the number of blocks detected using the 48 SAGs from OS2005, which only contained α SAGs. Comparison of the distributions of block lengths (**B**) and divergence between the two haplotypes from each block (**C**) is shown in the other two panels. OS2005 (shown in blue) is compared to a randomly chosen sample from panel A (shown in orange).

Based on the evidence presented above, we concluded that cross-contamination between reaction wells and genome assembly artifacts does not contribute significantly to the presence of mixed-species orthogroups or linkage blocks. While we cannot definitively exclude that some sequences we observe are the result of amplification or assembly errors, the evidence above shows that they are rare enough to not have a significant effect on our main conclusions.

