## [Editor Report · eLife Assessment]

This study provides **important** insights into bacterial genome evolution by analyzing single-cell genome sequences of cyanobacteria from Yellowstone hot springs. Using **compelling** evidence, the authors demonstrate that both homologous recombination within species and frequent hybridization across species are major drivers of genome diversification. Despite the challenges that are inherent to sparse and fragmented single-cell data, the analyses are thorough, carefully controlled, and supported by multiple complementary approaches, making the conclusions highly robust. This work represents a significant advance in our understanding of microbial evolution in natural environments.

---

## [Referee Report · Reviewer #1 (Public review)]

Summary:

What are the overarching principles by which prokaryotic genomes evolve? This fundamental question motivates the investigations in this excellent piece of work. While it is still very common in this field to simply assume that prokaryotic genome evolution can be described by a standard model from mathematical population genetics, and fit the genomic data to such a model, a smaller group of researchers rightly insists that we should not have such preconceived ideas and instead try to carefully look at what the genomic data tell us about how prokaryotic genomes evolve. This is the approach taken by the authors of this work. Lacking a tight theoretical framework, the challenge of such approaches is to device analysis methods that are robust to all our uncertainties about what the underlying evolutionary dynamics might be.

The authors here focus on a collection of ~300 single-cell genomes from a relatively well-isolated habitat with a relatively simple species composition, i.e. cyanobacteria living in hot springs in Yellowstone National Park. They convincingly demonstrate that the relative simplicity of this habitat increases our ability to interpret what the genomic data tells us about the evolutionary dynamics.

Using a very thorough and multi-faceted analysis of these data, the authors convincingly show that there are three main species of Synechococcus cyanobacteria living in this habitat, and that apart from very frequent recombination within each species (which is in line with insights from other recent studies) there is also a remarkably frequent occurrence of hybridization events between the different species, and with as of yet unindentified other genomes. Moreover, these hybridization events drive much of the diversity within each species. The authors also show convincing evidence that many of these hybridization events are not neutral but are driven by natural selection.

Strengths:

The great strength of this paper is that, by not making any preconceived assumptions about what the evolutionary dynamics is expected to look like, but instead devicing careful analysis methods to tease apart what the data tells us about what has happened in the evolution in these genomes, highly novel and unexpected results are obtained, i.e. the major role of hybridization across the 3 main species living in this habitat.

The analysis is very thorough and reading the detailed descriptions in the appendices it is clear that these authors took a lot of care in using these methods and avoiding the pitfalls that unfortunately affect many other studies in this research area.

The picture of the evolutionary dynamics of these three Synechococcus species that emerges from this analysis is quite novel and surprising. I think this study is a major stepping stone toward development of more realistic quantitative theories of genome evolution in prokaryotes.

The analysis methods that the authors employ are also partially quite novel and will no doubt by very valuable for analysis of many other datasets.

Weaknesses:

The main text is tight and concise, but this sort of hides the very large amount of careful complementary analyses that went into the conclusions presented in the main text. The appendices are quite well written but they are substantial, so that really understanding the paper is not an easy read. However, I do not really think the authors can be faulted for this. The topic is complex and a lot of care is required to make sure conclusions are valid.

A very interesting observation is that a lot of hybridization events (i.e. about half) originate from species other than the alpha, beta, and gamma Synechococcus species from which the genomes that are analyzed here derive. For this to occur, these other species must presumably also be living in the same habitat and must be relatively abundant. But if they are, why are they not being captured by the sampling? I did not see a clear explanation for this very common occurrence of hybridization events from outside of these Synechococcus species. The authors raise the possibility that these other species used to live in these hot springs but are now extinct or that the occur in other pools. I guess this is possible but I still find it puzzling and wonder if these donors could have been filtered out at some step of the experimental and/or analysis procedures.

---

## [Referee Report · Reviewer #2 (Public review)]

Summary.

Birzu et al. describe two sympatric hotspring cyanobacterial species ("alpha" and "beta") and infer recombination across the genome, including inter-species recombination events (hybridization) based on single-cell genome sequencing. The evidence for hybridization is strong and the authors took care to control for artefacts such as contamination during sequencing library preparation. Despite hybridization, the species remain genetically distinct from each other. The authors also present evidence for selective sweeps of genes across both species - a phenomenon which is widely observed for antibiotic resistance genes in pathogens, but rarely documented in environmental bacteria.

Strengths.

This manuscript describes some of the most thorough and convincing evidence to date of recombination happening within and between co-habitating bacteria in nature. Their single-cell sequencing approach allows them to sample the genetic diversity from two dominant species. Although single-cell genome sequences are incomplete, they contain much more information about genetic linkage than typical short-read shotgun metagenomes, enabling a reliable analysis of recombination. The authors also go to great lengths to quality-filter the single-cell sequencing data and to exclude contamination and read mismapping as major drivers of the signal of recombination. This is a fascinating dataset with intricate analyses showing the great extent of between-species hybridization that is possible in nature.

Weaknesses.

This revised version is much improved, with a much clearer flow and organisation within both the main text and supplement. The remaining weaknesses that I note below are certainly not critical, but are simply useful context for the reader to keep in mind.

My main concern is that the evidence for selection on the hybridized genes is incomplete and statements about the 'overwhelming evidence for the crucial role played by selection' (lines 334-5) are a bit overstated. What fraction of the hybridization events were driven by positive selection? The breakdown of hard (15%) vs soft (85%) sweeps is given, out of 153 (as sidenote, it is not clear if this is 153 genes or events, troughs, etc.). But how many of the hybridization events (or genes) have evidence for a selective sweep relative to those that do not? I recognize that this may be a hard question to answer, because it may be statistically easier to identify a hybridization event that rises to high frequency due to positive selection from a neutral event that remains rare. Even a rough estimate would be useful; would it be something like 153 out of the number of core genes tested (~700)?

Regardless, I think that Figure 6 (A and B) could benefit from comparison to a neutral model, including hybridization but no selection to see if a similar pattern (notably, higher synonymous diversity in alpha troughs compared to the backbone) could arise due to hybridization alone without selection.

An implicit assumption in microbiology is often that cross-species recombination events are driven by selection. The authors recognize that "diversity troughs resulted from selective sweeps [...] likely overcame mechanistic barriers to recombination, genetic incompatibilities, and ecological differences" (lines 335-7) and thus would not be retained unless they had some strong adaptive value to offset these costs. There are surprisingly few tests of the hypothesis that cross-species recombination events tend to be driven by selection. An analysis of Streptococcus spp. genomes showed that between-species recombination events tended to be accompanied by positive selection, whereas most within-species events were not (Shapiro et al. Trends in Microbiology 2009; reanalysis of data from Lefebure & Stanhope, Genome Biology 2007). There are probably other examples out there, but the authors could highlight that they provide rare data to support a common expectation.

---

## [Author Response]

The following is the authors’ response to the original reviews

**Public Reviews:**

**Reviewer #1 (Public Review):**
Summary:What are the overarching principles by which prokaryotic genomes evolve? This fundamental question motivates the investigations in this excellent piece of work. While it is still very common in this field to simply assume that prokaryotic genome evolution can be described by a standard model from mathematical population genetics, and fit the genomic data to such a model, a smaller group of researchers rightly insists that we should not have such preconceived ideas and instead try to carefully look at what the genomic data tell us about how prokaryotic genomes evolve. This is the approach taken by the authors of this work. Lacking a tight theoretical framework, the challenge of such approaches is to devise analysis methods that are robust to all our uncertainties about what the underlying evolutionary dynamics might be.The authors here focus on a collection of ~300 single-cell genomes from a relatively well-isolated habitat with relatively simple species composition, i.e. cyanobacteria living in hotsprings in Yellowstone National Park, and convincingly demonstrate that the relative simplicity of this habitat increases our ability to interpret what the genomic data tells us about the evolutionary dynamics.Using a very thorough and multi-faceted analysis of these data, the authors convincingly show that there are three main species of Synechococcus cyanobacteria living in this habitat, and that apart from very frequent recombination within each species (which is in line with insights from other recent studies) there is also a remarkably frequent occurrence of hybridization events between the different species, and with as of yet unidentified other genomes. Moreover, these hybridization events drive much of the diversity within each species. The authors also show convincing evidence that these hybridization events are not neutral but are driven by selected by natural selection.Strengths:The great strength of this paper is that, by not making any preconceived assumptions about what the evolutionary dynamics is expected to look like, but instead devising careful analysis methods to tease apart what the data tells us about what has happened in the evolution in these genomes, highly novel and unexpected results are obtained, i.e. the major role of hybridization across the 3 main species living in this habitat.The analysis is very thorough and reading the detailed supplementary material it is clear that these authors took a lot of care in devising these methods and avoiding the pitfalls that unfortunately affect many other studies in this research area.The picture of the evolutionary dynamics of these three Synechococcus species that emerge from this analysis is highly novel and surprising. I think this study is a major stepping stone toward the development of more realistic quantitative theories of genome evolution in prokaryotes.The analysis methods that the authors employ are also partially novel and will no doubt be very valuable for analysis of many other datasets.

We thank the reviewer for their appreciation of our work.

Weaknesses:I feel the main weakness of this paper is that the presentation is structured such that it is extremely difficult to read. I feel readers have essentially no chance to understand the main text without first fully reading the 50-page supplement with methods and 31 supplementary materials. I think this will unfortunately strongly narrow the audience for this paper and below in the recommendations for the authors I make some suggestions as to how this might be improved.A very interesting observation is that a lot of hybridization events (i.e. about half) originate from species other than the alpha, beta, and gamma Synechococcus species from which the genomes that are analyzed here derive. For this to occur, these other species must presumably also be living in the same habitat and must be relatively abundant. But if they are, why are they not being captured by the sampling? I did not see a clear explanation for this very common occurrence of hybridization events from outside of these Synechococcus species. The authors raise the possibility that these other species used to live in these hot springs but are now extinct. I'm not sure how plausible this is and wonder if there would be some way to find support for this in the data (e.g that one does not observe recent events of import from one of these unknown other species). This was one major finding that I believe went without a clear interpretation.

We agree with the reviewer that the extent of hybridization with other species is surprising. While we do feel that our metagenome data provide convincing evidence that “X” species are not present in MS or OS, we cannot currently rule out the presence of X in other springs. In the revision we explicitly mention the alternative hypothesis (Lines 239-242).

The core entities in the paper are groups of orthologous genes that show clear evidence of hybridization. It is thus very frustating that exactly the methods for identifying and classifying these hybridization events were really difficult to understand (sections I and V of the supplement). Even after several readings, I was unsure of exactly how orthogroups were classified, i.e. what the difference between M and X clusters is, what a `simple hybrid' corresponds to (as opposed to complex hybrids?), what precisely the definitions of singlet and non-singlet hybrids are, etcetera. It also seems that some numbers reported in the main text do not match what is shown in the supplement. For example, the main text talks about "around 80 genes with more than three clusters (SM, Sec. V; fig. S17).", but there is no group with around 80 genes shown in Fig S17! And similarly, it says "We found several dozen (100 in α and 84 in β) simple hybrid loci" and I also cannot match those numbers to what is shown in the supplement. I am convinced that what the authors did probably made sense. But as a reader, it is frustrating that when one tries to understand the results in detail, it is very difficult to understand what exactly is going on. I mention this example in detail because the hybrid classification is the core of this paper, but I had similar problems in other sections.

We thank the reviewer for pointing out these issues with our original presentation. In the revision, we have redone most of the analysis to simplify the methods and check the consistency of the results. We did not find any qualitative differences in our results after reanalysis, but some of the numbers for different hybridization patterns have changed. The most notable difference is an increase in the number of alpha-gamma simple hybrids and a corresponding decrease in mixed-species clusters (now labeled mosaic hybrids). These transfers are difficult to assign because we only have access to a single gamma genome. We have added a short explanation of this point in Lines 219-222.

To improve the presentation, we significantly expanded the “Results” section to better explain our analysis and the different steps we take. We included two additional figures (Figs. 3 and 4) that illustrate the different types of hybrids and the heterogeneity in the diversity of alpha which is discussed in the main text and is important for interpreting our results. We also included two additional figures (Figs. 2 and 6) that were previously in the Appendix but were mentioned in the main text. We believe these changes should address most of the issues raised by the reviewer and hopefully make the manuscript easier to read.

Although I generally was quite convinced by the methods and it was clear that the authors were doing a very thorough job, there were some instances where I did not understand the analysis. For example, the way orthogroups were built is very much along the lines used by many in the field (i.e. orthoMCL on the graph of pairwise matchings, building phylogenies of connected components of the graph, splitting the phylogenies along long branches). But then to subdivide orthogroups into clusters of different species, the authors did not use the phylogenetic tree already built but instead used an ad hoc pairwise hierarchical average linkage clustering algorithm.

The reviewer is correct that there is an unexplained discrepancy between the clustering methods we used at different steps in our pipeline. We followed previous work by using phylogenetic distances for the initial clustering of orthogroups. On these scales we expect hybridization to play a minor role and phylogenetic distances to correlate reasonably well with evolutionary divergence. However, because of the extensive hybridization we observed, the use of phylogenetic models for species clustering is more difficult to justify. We therefore chose to simply use pairwise nucleotide distances, which make fewer assumptions about the underlying evolutionary processes and should be more robust. We have briefly explained our reasoning and the details of our clustering method in the revision (Lines 182-190).

**Reviewer #2 (Public Review):**
Summary:Birzu et al. describe two sympatric hotspring cyanobacterial species ("alpha" and "beta") and infer recombination across the genome, including inter-species recombination events (hybridization) based on single-cell genome sequencing. The evidence for hybridization is strong and the authors took care to control for artefacts such as contamination during sequencing library preparation. Despite hybridization, the species remain genetically distinct from each other. The authors also present evidence for selective sweeps of genes across both species - a phenomenon which is widely observed for antibiotic resistance genes in pathogens, but rarely documented in environmental bacteria.Strengths:This manuscript describes some of the most thorough and convincing evidence to date of recombination happening within and between cohabitating bacteria in nature. Their single-cell sequencing approach allows them to sample the genetic diversity from two dominant species. Although single-cell genome sequences are incomplete, they contain much more information about genetic linkage than typical short-read shotgun metagenomes, enabling a reliable analysis of recombination. The authors also go to great lengths to quality-filter the single-cell sequencing data and to exclude contamination and read mismapping as major drivers of the signal of recombination.

We thank the reviewer for their appreciation of our work.

Weaknesses:Despite the very thorough and extensive analyses, many of the methods are bespoke and rely on reasonable but often arbitrary cutoffs (e.g. for defining gene sequence clusters etc.). Much of this is warranted, given the unique challenges of working with single-cell genome sequences, which are often quite fragmented and incomplete (30-70% of the genome covered). I think the challenges of working with this single-cell data should be addressed up-front in the main text, which would help justify the choices made for the analysis.

We have significantly expanded the “Results” section to better justify and explain the choices we made during our analysis. We hope these changes address the reviewer’s concerns and make the manuscript more accessible to readers.

The conclusions could also be strengthened by an analysis restricted to only a subset of the highest quality (>70% complete) genomes. Even if this results in a much smaller sample size, it could enable more standard phylogenetic methods to be applied, which could give meaningful support to the conclusions even if applied to just ~10 genomes or so from each species. By building phylogenetic trees, recombination events could be supported using bootstraps, which would add confidence to the gene sequence clustering-based analyses which rely on arbitrary cutoffs without explicit measures of support.

It seems to us that the reviewer’s suggestion presupposes that the recombination events we find can be described as discrete events on an asexual phylogeny, similar to how rare mutations are treated in standard phylogenetic inference. Popular tools, such as ClonalFrame and its offshoots, have attempted to identify individual recombination events starting from these assumptions. But the main conclusion of both our linkage and SNP block analysis is that the ClonalFrame assumptions do not hold for our data. Under a clonal frame, the SNP blocks we observe should be perfectly linked, similar to mutations on an asexual tree. But our results in Fig. 7D show the opposite. Part of the issue may have been that in our original presentation, we only briefly discuss the results of our linkage analysis and refer readers to the Appendix for more details. To fix this issue we have added an extra figure (Fig. 2), showing rapid linkage decrease in both species and that at long distances the linkage values are essentially identical to the unlinked case, similar to sexual populations. We hope that this change will help clarify this point.

The manuscript closes without a cartoon (Figure 4) which outlines the broad evolutionary scenario supported by the data and analysis. I agree with the overall picture, but I do think that some of the temporal ordering of events, especially the timing of recombination events could be better supported by data. In particular, is there evidence that inter-species recombination events are increasing or decreasing over time? Are they currently at steady-state? This would help clarify whether a newly arrived species into the caldera experiences an initial burst of accepting DNA from already-present species (perhaps involving locally adaptive alleles), or whether recombination events are relatively constant over time.

The reviewer raises some very interesting questions about the dynamics of recombination in the population, which we hope to pursue in future work. We have added this as an open question in the Discussion (Lines 365-382).

These questions could be answered by counting recombination events that occur deeper or more recently in a phylogenetic tree.

The reviewer here seems to presuppose that recombination is rare enough that a phylogenetic tree can reliably be inferred, which is contrary to our linkage analysis (see the response to an earlier comment). Perhaps the reviewer missed this point in our original manuscript since it was discussed primarily in the Appendix. See also our response to a previous comment by the reviewer.

The cartoon also shows a 'purple' species that is initially present, then donates some DNA to the 'blue' species before going extinct. In this model, 'purple' DNA should also be donated to the more recently arrived 'orange' species, in proportion to its frequency in the 'blue' genome. This is a relatively subtle detail, but it could be tested in the real data, and this may actually help discern the order of the inferred recombination events.

We have included an extra figure in the main text (Fig. 6) that addresses the question of timing of events. A quantitative test of our cartoon model along the lines the reviewer suggested would certainly be worthwhile and we hope to do that in future work.

The abstract also makes a bold claim that is not well-supported by the data: "This widespread mixing is contrary to the prevailing view that ecological barriers can maintain cohesive bacterial species..." In fact, the two species are cohesive in the sense that they are identifiable based on clustering of genome-wide genetic diversity (as shown in Fig 1A). I agree that the mixing is 'widespread' in the sense that it occurs across the genome (as shown in Figure 2A) but it is clearly not sufficient to erode species boundaries. So I believe the data is consistent with a Biological Species Concept (sensu Bobay & Ochman, Genome Biology & Evolution 2017) that remains 'fuzzy' - such that there are still inter-species recombination events, just not sufficient to erode the cohesion of genomic clusters. Therefore, I think the data supports the emerging picture of most bacteria abiding by some version of a BSC, and is not particularly 'contrary' to the prevailing view.

We have revised the phrase mentioned by the reviewer to “prevent genetic mixture between bacterial species,” which more accurately represents our conclusions.

The final Results paragraph begins by posing a question about epistatic interactions, but fails to provide a definitive answer to the extent of epistasis in these genomes. Quantifying epistatic effects in bacterial genomes is certainly of interest, but might be beyond the scope of this paper. This could be a Discussion point rather than an underdeveloped section of the Results.

We agree with the reviewer that an exhaustive analysis of epistasis in the population is beyond the scope of the manuscript. Our original intention was to answer whether SNP blocks we discovered showed evidence of strong linkage, as might be expected if only a small number of strains are present in the population. In light of the previous comments by the reviewer regarding the consistency with the clonal frame hypothesis, we believe this is especially relevant for our results. Moreover, the results we found‑especially for the beta population‑were quite conclusive: SNP block linkages in beta are indistinguishable from an unlinked model. To avoid misdirecting the reader about the significance of our results, we have revised the relevant paragraph (Lines 316-319).

**Recommendations For The Authors:**

**Reviewer #1 (Recommendations For The Authors):**
Although I am entirely convinced of the validity of the results, methodology, and interpretations presented in this work, I must say I found the paper very hard to read. And I think I am really quite familiar with these kinds of approaches. I fear that for people other than experts on these kinds of comparative genomic analyses, this paper will be almost impossible to read. With the aim of expanding the audience for this compelling work, I think the authors might want to consider ways to improve the presentation.At the end of a long project, the obtained results typically form a web of mutual interconnections and dependencies and one of the key challenges in presenting the results in a paper is having to untangle this web of connected results and analysis into a linear ordered narrative so that, at any point in the narrative, understanding the next point only depends on previous points in the narrative. I frankly feel that this paper fails at this.The paper reads to me as if one author put together the supplement by essentially writing a report of all the analyses that were done together with supplementary figures summarizing all those analyses, and that another author then wrote the main text by using the materials in the supplement almost in the way a cook uses ingredients for a dish. Almost every other sentence in the main text refers to results in the (31!) supplementary figures and can only be understood by reading the appropriate corresponding sections in the supplementary materials. I found it essentially impossible to read the main text without having first read the entire 50-page supplement.I think the paper could be hugely improved by trying to restructure the presentation so as to make it more linear. The main text can be expanded to include a summary of the crucial methods and analysis results from the supplement needed to understand the narrative in the main text. For example, as it currently stands it is really challenging to understand what is shown in figures 2 and 3 of the main text without having to first read a very substantial part of the supplement. Figure 3, even after having read the relevant sections in the supplement, took me quite a while to understand and almost felt like a puzzle to decypher. Rethinking which parts of the supplement are really necessary would also help. Finally, it would also help if the terminology was kept as simple, transparent, and consistent as possible.I understand that my suggestion to thoroughly reorganize the presentation may feel like a big hassle, but I am afraid that in its current form, these important results are essentially rendered inaccessible to all but a small group of experts in this area. This paper deserves a wider readership.

We thank the reviewer for these valuable suggestions. In the revision, we have significantly expanded and restructured the “Results” section to make the presentation more linear, as the reviewer suggested (see our reply to the public comment by the reviewer for details). We hope these changes will make the manuscript easier to read.

**Reviewer #2 (Recommendations For The Authors):**
I found this paper challenging to follow since the main text was so condensed and the supplementary material so extensive. Given that eLife does not impose strong limits on the length of the main text, I suggest moving some key sections from the supplement into the main text to make it easier for the reader to follow rather than flipping back and forth. Adding to the confusion, supplementary figures were referenced out of order in the main text (e.g. S23 is referenced before S1). Please check the numbering and ensure figures are mentioned in the main text in the correct order.

We thank the reviewer for their feedback on the presentation of the results. In response to similar comments from Reviewer #1, we have significantly expanded and restructured the “Results” section to make it easier to read (see also our responses to Reviewer #1).

Page 2: The term 'coevolution' is typically reserved for two species that mutually impose selective pressures on one another (e.g. predator-prey interactions; see Janzen, Evolution 1980). In the context of these two cyanobacterial species, it's not clear that this is the case so I would simply refer to them 'cohabitating' or being sympatric in the same environment.

It is true that the term "coevolution” has become associated with predator-prey interactions, as the reviewer said. However, we feel that in our case “coevolution” fairly accurately describes the continual hybridization over long time scales we observe. We have therefore chosen to keep the term.

Page 3: The authors mention that the gamma SAG is ~70% complete, which turns out to be quite high. It would be useful to mention early in the Results the mean/median completeness across SAGs, and how this leads to some challenges in analysing the data. Some of the material from the Supplement could be moved into the Results here.

We have added a short note on the completeness in the Results (Lines 153-154). We have also added an extra figure in Appendix 1 with the completeness of all the SAGs for interested readers.

I was left puzzled by the sentence: "Alternatively, high rates of recombination could generate different genotypes within each genome cluster that are adapted to different temperatures, with the relative frequencies of each cluster being only a correlated and not a causal driver of temperature adaptation." This is suggesting that individual genes or alleles, rather than entire genomes, could be adapted to temperature. But figure 1B seems to imply that the entire genome is adapted to different temperatures. Anyway, this does not seem to be a key point and could probably be removed (or clarified if the authors deem this an important point, which I failed to understand).

We have revised this section to clarify the alternative hypothesis mentioned by the reviewer (Lines 100-103).

Page 4. 'Several dozen' hybrid genes were found, but please also specify how many genes were tested. In general, it would be good to briefly outline the sample size (SAGs or genes) considered for each analysis.

We have added the total numbers of genes we analyzed at each step of our analysis.

'Mosaic hybrid loci' are mentioned alongside the issue of poor alignment. Presumably, the mosaic hybrid loci are first filtered to remove the poor alignments? This should be specified, and please mention how many loci are retained before/after this filter.

We thank the reviewer for highlighting this important point. In the revision, we have implemented a more aggressive filtering of genes with poor alignments. We have added an extra paragraph to Appendix 1 (step 5 in the pipeline analysis) briefly explaining the issue.

Page 5. "By contrast, the diversity of mosaic loci was typical of other loci within beta, suggesting most of the beta genome has undergone hybridization." Please point to the data (figure) to support this statement.

We have restructured our discussion of the different hybrid loci so this comment is no longer relevant. In case the reviewer is interested, the synonymous diversity within beta was 0.047, while in mosaic hybrids it was 0.064.

Page 6. "The largest diversity trough contained 28 genes." Since this trough is discussed in detail and seems to be of interest, it would be nice to illustrate it, perhaps as an inset in Figure 2 or as a separate figure. If I understood correctly, this trough includes genes (in a nitrogen-fixation pathway) that are present in all genomes, but are exchanged by homologous recombination. So I don't think it's correct to say that the "ancestors acquired the ability to fix nitrogen." Rather, the different alleles of these same genes were present in the ancestor. So perhaps there was a selective sweep involving alleles in this region that provided adaptation to local nitrogen sources or concentrations, but not a gain of new genes. Perhaps I misunderstood, in which case clarification would be appreciated.

The reviewer raises an interesting possibility. We agree that it is in principle possible that the ancestor contained the nitrogen fixation genes and the selective sweep simply replaced the ancestral alleles. In this particular case, there is additional evidence that the entire pathway was acquired around roughly the same time from gene order. The gene order between alpha and beta is almost entirely different, with only a few segments containing more than 2-3 genes in the same order, as shown by Bhaya et al. 2007 and confirmed by additional unpublished analysis of the SAGs. One of the few exceptions is the nitrogen fixation pathway, which has essentially the same gene order over more than 20 kbp. Thus, if the ancestor of both alpha and beta contained the nitrogen-fixation pathway, we would expect these genes to be scatter across the genome. We have revised the sentences in question to clarify this point (Lines 260-271).

Page 6. Last paragraph on epistasis references Fig 3C, but I believe it should be Fig 3D.

Fixed.

Page 7. Figure 3 legend. "Note that alpha-2 is identical to gamma here." I believe it should be beta, not gamma.

The reviewer is correct. We have fixed this error.

Page 8. What is the evidence for "at least six independent colonizers"? I could not find the data supporting this claim.

The statement mentioned by the reviewer was based on the maximum number of species clusters we identified in different core genes. However, during the revision, we found that only a handful of genes contained five or more clusters. We did find several tens of genes with four clusters. In addition, Rosen et al. (2018) also found additional 16S clusters at low frequency in the same springs. Based on these results we conservatively estimate that at least four independent strains colonized the caldera, but the number could be much greater. We have revised the text in question accordingly (Lines 336-339) and added Fig. 2 in Appendix 1 to support the conclusion.

Page 9. Line 200: "acting to homogenize the population." It should be specified that the population is only homogenized at these introgressed loci, not genome-wide. Otherwise, the genome-wide species clusters seen in Fig 1 would not be maintained.

It is true that the selective sweeps that lead to diversity throughs only homogenize the introgressed loci. But other hybrid segments could also rise to high frequency in the population during the sweep through hitchhiking. The fact that we observe SNP blocks generated through secondary recombination events of introgressed segments throughout the genome supports this view. While we do not fully understand the dynamics of this process currently, we do feel that the current evidence supports the statement that mixing is occurring throughout the genome and not just at a few loci so we have kept the original statement.

The final sentence (lines 221-222) is vague and uninformative. On the one hand, "investigating whether hybridization plays a major role" is what the current manuscript has already done - depending on what is meant by 'major' (how much of the genome? Or whether there are ecological implications?). It is also not clear what is meant by a predictive theory and 'possible evolutionary scenarios. This should be elaborated upon, otherwise, it is not clear what the authors mean. Otherwise, this sentence could be cut.

We thank the reviewer for their feedback. One possible source of confusion could be that in this sentence we were referring to detecting hybridization in other communities. We have changed “these communities” to “other communities” to make this clearer.

Supplement.Broadly speaking, I appreciate the thorough and careful analysis of the single cell data. On the other hand, it is hard to evaluate whether these custom analyses are doing what is intended in many cases. Would it be possible to consider an analysis using more established methods, e.g. taking a subset of genomes with 'good' completeness and using Panaroo to find the core and accessory genome, then ClonalFrameML or Gubbins to infer a phylogeny and recombination events? Such analyses could probably be applied to a subset of the sample with relatively complete genomes. I don't want to suggest an overly time-consuming analysis, but the authors could consider what would be feasible.

We have added a comparison between our analysis and that from two other methods, including ClonalFrameML mentioned by the author. One important point that we feel might have been lost in the first version is that our linkage results imply that recombination is not rare such that it can be mapped onto an asexual tree as assumed by ClonalFrameML. Note that this is not simply due to technical limitations due to incomplete coverage and is instead a consequence of the evolutionary dynamics of the population. Consistent with this, we found several inconsistencies in how recombination events were assigned by ClonalFrameML. We have summarized these conclusions in Appendix 7 of the revised manuscript.

Page 8. Line 190. What is meant by 'minimal compositional bias'?

We mean that the sample is not biased towards strains that grow in the lab. We have revised the sentence to clarify.

Page 25. Figure S14 is not referenced in the text.

We have added part of this figure to the main text since it illustrates one of our main results, namely that sites at long genomic distances are essentially unlinked.

Page 26. The 'unlinked controls' (line 530) are very useful, but it would be even more informative to see if these controls also show the same decline in linkage with distance in the genome as observed in the real data. In particular, it would be good to know if the observed rapid decline in linkage with distance in the low-diversity regions is also observed in controls. Currently, it is unclear if this observation might be due to higher uncertainty in inferring linkage in low-diversity regions, which by definition have less polymorphism to include in the linkage calculation.

We thank the reviewer for the suggestion. After further consideration, we have decided to remove the subsection on linkage decrease in the low-diversity regions. We feel such detailed quantitative analysis would be better suited for a more technical paper, which we hope to do at a later time.

Page 26. There are some sections with missing identifiers (Sec ??).

Fixed.

Page 27. The information about the typical breadth of SAG coverage (~30%) would be better to include earlier in the Supplement, and also mentioned in the main text so the reader can more easily understand the nature of the dataset.

We have added an extra figure with the SAG coverages to Appendix 1.

Page 29. Any sensitivity analysis around the S = 0.9 value? Even if arbitrary, could the authors provide justification why they think this value is reasonable?

We have significantly revised this section in response to earlier comments by one of the reviewers. We hope that this would clarify the details of our methods to interested readers. To answer the reviewer’s specific question, we chose this heuristic after examining the fraction of cells of each species in different species clusters. For the clusters assigned to alpha and beta, we found a sharp peak near one and that a cutoff of 0.9 captured most clusters while still being high enough to inconsistent with a mixed cluster.

Page 30. I could not see where Fig. S17 was mentioned in the text. Also, how are 'simple hybrid genes' defined?

We have removed this figure in the revision. The definition of the different types of hybrid genes have been added to the main text in response to a comment from the other reviewer.

Page 36. It is hard to see that divergence is 'high' relative to what reference. Would it be possible to include the expected value (from ref. 12) in the plot, or at least explicitly mentioned in the text?

We have added the mean synonymous and non-synonymous divergences between alpha and beta to the figures for reference.

Page 38. Line 770 "would be comparable to that of beta." This is not necessarily the case since beta could have a different time to its most recent common ancestor. It could have a different time to the last bottleneck or selective sweep, etc.

We thank the reviewer for pointing out this misleading statement. Our point here was that in the first scenario the TMRCA of alpha and beta would be similar since the diversity in the high-diversity alpha genes is similar to beta. We have clarified this statement in the revision.

Page 39. Line 793. The use of the term 'genomic backbone' implies the presence of a clonal frame, which is not what the data seems to support. Perhaps another term such as 'genetic diversity' would more appropriately capture the intended meaning here.

We agree with the reviewer that the low-diversity regions may not be asexual. We used “genomic backbone” to distinguish from the “clonal frame,” which is usually used to mean that the backbone is asexual. We have added a note in the revision to clarify this point.

Page 39. Lines 802-805. I found this explanation hard to follow. Could the logic be clarified?

We simply meant that although the beta distribution is unimodal, it is not consistent with a simple Poisson distribution, unlike in alpha. We have added an extra sentence to clarify this.